**Bulk and Molecular-Level Characterization of Laboratory-Aged Biomass Burning Organic Aerosol from Oak Leaf and Heartwood Fuels**

Claire F. Fortenberry[1], Michael J. Walker[1], Yaping Zhang[1], Dhruv Mitroo[1,a], William H. Brune[2], Brent J. Williams[1]

[1] Department of Energy, Environmental, and Chemical Engineering, Washington University in Saint Louis, Saint Louis, MO 63130, USA

[2] Department of Meteorology and Atmospheric Science, Pennsylvania State University, State College, PA 16801, USA

[a] Now at the Department of Atmospheric Sciences, Rosenstiel School of Marine and Atmospheric Sciences, University of Miami, Miami, FL 33149, USA

*Correspondence to:* Brent J. Williams (brentw@wustl.edu)

**Abstract.**

The chemical complexity of biomass burning organic aerosol (BBOA) greatly increases with photochemical aging in the atmosphere, necessitating controlled laboratory studies to inform field observations. In these experiments, BBOA from American white oak (*Quercus alba*) leaf and heartwood samples was generated in a custom-built emissions and combustion chamber and photochemically aged in a Potential Aerosol Mass (PAM) flow reactor. A Thermal desorption Aerosol Gas chromatograph (TAG) was used in parallel with a high-resolution time-of-flight Aerosol Mass Spectrometer (AMS) to analyze BBOA chemical composition at different levels of photochemical aging. Individual compounds were identified and integrated to obtain relative decay rates for key molecules. A recently-developed chromatogram binning positive matrix factorization (PMF) technique was used to obtain mass spectral profiles for factors in TAG BBOA chromatograms, improving analysis efficiency and providing a more complete determination of unresolved complex mixture (UCM) components. Additionally, the recently characterized TAG decomposition window was used to track molecular fragments created by the decomposition of thermally labile BBOA during sample desorption. We demonstrate that although most primary (freshly-emitted) BBOA compounds deplete with photochemical aging, certain components eluting within the TAG thermal decomposition window are instead enhanced. Specifically, the increasing trend in the decomposition $m/z$ 44 signal ($CO_2^+$) indicates formation of secondary organic aerosol (SOA) in the PAM reactor. Sources of $m/z$ 60 ($C_2H_4O_2^+$), typically attributed to freshly-emitted BBOA in AMS field measurements, were also investigated. From the TAG chemical speciation and decomposition window data, we observed a decrease in $m/z$ 60 with photochemical aging due to the decay of anhydrosugars (including levoglucosan) and other compounds, as well as an increase in $m/z$ 60 due to the formation of thermally labile organic acids within the PAM reactor, which decompose during TAG sample desorption. When aging both types of BBOA (leaf and heartwood), the AMS data exhibit a combination of these two contributing effects, causing limited change to the overall $m/z$ 60 signal. Our observations demonstrate the importance of chemically-speciated data in fully understanding bulk aerosol measurements provided by the AMS in both laboratory and field studies.

## 1 Introduction

Atmospheric particulate matter (PM) negatively affects human health (e.g., Kampa and Castanas, 2008), impedes visibility (e.g., Appel et al., 1985), and impacts the global energy balance through direct radiative forcing or by acting as cloud condensation nuclei (e.g., Kanakidou et al., 2005). Organic aerosol (OA) particles compose 20-90% of submicron PM ($PM_1$) and may consist of thousands of distinct organic compounds (Goldstein and Galbally, 2007; Ng et al., 2010; Zhang et al., 2007). Given the multitude of organic compounds in the atmosphere and the numerous chemical reactions they can experience during atmospheric processing (e.g., Goldstein and Galbally, 2007; Kroll et al., 2009), laboratory studies are needed to fully understand the chemical composition and oxidative evolution of source-specific primary OA (POA, aerosol emitted directly into the atmosphere) and secondary OA (SOA, formed from gas-phase material that partition into the particle phase following photooxidation).

Biomass burning organic aerosol (BBOA) may contribute up to 90% of global combustion OA and 75% of combustion POA (Bond et al., 2004; May et al., 2013). Previous BBOA molecular speciation studies over the past several decades have focused on the chemical composition of primary emissions (e.g., Fine et al., 2002; Oros and Simoneit, 1999; Rogge et al., 1998; Simoneit et al., 2000). Recently, improved understanding of SOA formation in BBOA plumes has motivated the use of oxidation chambers in laboratory BBOA experiments (e.g., Cubison et al., 2011; Grieshop et al., 2009; Ortega et al., 2013). Some of these BBOA photooxidation studies have demonstrated that OA production can exceed decay under certain conditions due to oxidation and phase partitioning of gas-phase semivolatile and intermediately volatile compounds (SVOCs and IVOCs, respectively; Grieshop et al., 2009). Other field measurements show minimal OA enhancement with aging of primary biomass plumes (Capes et al., 2008). During the third Fire Lab at Missoula Experiment (FLAME-3) campaign (2013), OA enhancements following photooxidation varied widely depending on the biomass source; although BBOA from some sources doubled in mass after photochemical aging, other types of BBOA were depleted (Ortega et al., 2013). The variation in OA enhancement observed by Ortega et al. in the FLAME-3 study suggests that the amount of SOA from biomass emissions depends on the fuel type, illustrating the need for source-specific oxidation studies to investigate reactions and products leading to SOA formation.

Previous BBOA oxidation studies (e.g., Grieshop et al., 2009; Ortega et al., 2013) have utilized a High-Resolution Time-Of-Flight Aerosol Mass Spectrometer (HR-ToF-AMS, Aerodyne, Inc., Billerica, MA). The Aerosol Mass Spectrometer (AMS) obtains chemical information on bulk aerosol including total mass concentrations and high-resolution ion signals, allowing for determination of bulk aerosol chemical composition (Canagaratna et al., 2007; DeCarlo et al., 2006). Hydrogen-to-carbon ratios (H:C) and oxygen-to-carbon ratios (O:C) can also be calculated using high-resolution AMS data, which are incorporated into estimations of an average carbon oxidation state ($\overline{OS_C} \approx 2 \times H{:}C - O{:}C$; Kroll et al., 2011). Although the AMS provides real-time measurements of ensemble-averaged properties for submicron non-refractory aerosol, it does not achieve molecular speciation and thus cannot be used to identify individual compounds present in OA. Typical AMS BBOA studies use $m/z$ 60 ($C_2H_4O_2^+$) and $m/z$ 44 ($CO_2^+$) signals to quantify primary and aged emissions, respectively (e.g., Cubison et al., 2011; Ng et al., 2010, 2011). Levoglucosan, a cellulose decomposition product often used as a molecular tracer for freshly-emitted BBOA (e.g.,

Simoneit et al., 1999, 2004), is frequently considered to be a primary contributor to $m/z$ 60 in AMS laboratory and field studies (e.g., Lee et al., 2010; Ng et al., 2011b). However, although levoglucosan has traditionally been understood to remain stable over relevant timescales (Fraser and Lakshmanan, 2000; Locker, 1988; Simoneit et al., 1999), multiple laboratory studies suggest that hydroxyl radical (OH)-driven levoglucosan decay occurs at a timescale similar to transport and deposition timescales (Hennigan et al., 2010; Hoffmann et al., 2010; Lai et al., 2014). Additionally, recent measurements demonstrate that $m/z$ 60 abundances may remain above background levels with sufficient atmospheric processing, suggesting that not all $m/z$ 60 originates from BBOA (Cubison et al., 2011; Ortega et al., 2013). These two considerations highlight the need for *in situ* molecular speciation measurements to complement bulk aerosol chemical data supplied by the AMS.

The Thermal desorption Aerosol Gas chromatograph (TAG) pairs automated aerosol collection and thermal desorption with gas chromatography-mass spectrometry (GC-MS), providing molecular-level speciation with *in situ* analysis and hourly time resolution (Williams et al., 2006). The TAG has been used in field studies to identify molecular tracers in ambient air and to link different chemical profiles to unique sources (e.g., Kreisberg et al., 2009; Lambe et al., 2009; Williams et al., 2007, 2010, 2014; Worton et al., 2011; Zhang et al., 2014, 2016). The TAG is capable of providing speciated compound measurements for approximately 20% of total organic aerosol mass on average, depending on the type of aerosol collected (Williams et al., 2006). Although the TAG reliably detects a high fraction (up to 100%) of hydrocarbon OA mass, which is typical of POA, the analyzed fraction of oxidized OA mass is often much lower (Williams et al., 2010, 2016; Zhang et al., 2014). This discrepancy is attributed to low mass throughput of oxidized species through the 30-meter non-polar GC capillary column (Williams et al., 2006, 2016) and presents a disadvantage for TAG analysis of oxidized components typical of SOA.

Recent advances have expanded the TAG's analytical capability. Traditional gas chromatography (GC) utilizes a solvent delay to prevent detector damage from large solvent or water signals. In the TAG, much of the solvent can be purged prior to sample injection, and the solvent delay is no longer applied. The lack of a solvent delay allows volatile components and aerosol thermal decomposition products to reach the detector during thermal sample desorption (5-15 minutes of TAG GC program) from the TAG collection cell to the GC column. The mass-spectral signal within this period, called the thermal decomposition window, typically features an air signal (e.g., $m/z$ 32 for $O_2^+$, $m/z$ 40 for $Ar^+$, $m/z$ 44 for $CO_2^+$), but can also contain ions characteristic of decomposing nitrates ($m/z$ 30 for $NO^+$, $m/z$ 46 for $NO_2^+$), sulfates ($m/z$ 48 for $SO^+$, $m/z$ 64 for $SO_2^+$), and organics ($m/z$ 44 for $CO_2^+$). These ion signals were shown to correlate with corresponding AMS ions for ambient data collected during the Saint Louis Air Quality Regional Study in 2013 (Williams et al., 2016). However, the TAG thermal decomposition window has only recently been used to analyze ambient data, and more laboratory studies are needed to explore the thermal decomposition products of OA from unique sources.

In this work, we present results from laboratory studies aimed at characterizing BBOA chemical composition using both the TAG compound window (minutes 20-55 of the chromatogram; Figure 1) and the TAG thermal decomposition window (minutes 6-16 of the chromatogram; Figure 1) in parallel with an AMS. A custom-built emissions and combustion chamber was used to generate BBOA, and a Potential Aerosol Mass (PAM) oxidative flow reactor (OFR),

which can mimic up to 16 days of atmospheric aging with residence times on the order of 100 seconds (Kang et al., 2007; Lambe et al., 2011), was used to oxidize laboratory-generated BBOA plumes at different levels of accelerated photochemistry. Our experiments addressed three primary objectives. First, the chemical composition of laboratory-generated BBOA was explored to identify molecular tracers from the leaf and heartwood of the American white oak (*Quercus alba*). Recently-developed chromatogram-binning Positive Matrix Factorization (PMF) techniques (Zhang et al., 2014, 2016) were applied to the TAG compound window to determine the prevalence of different compound classes and functionalities unique to BBOA from each fuel type. Trends in compounds and compound classes with oxidation were evaluated using both individual compound integrations and chromatogram binning PMF results. Second, the TAG thermal decomposition window was used to investigate how the chemical composition of thermally decomposing BBOA varies with PAM aging. Concurrent AMS measurements were taken to complement TAG decomposition window data, providing $\overline{OS_C}$ estimations and high-resolution ion signals for bulk BBOA samples. These AMS parameters were used to inform interpretation of TAG decomposition ion signals, particularly the variation of TAG decomposition *m/z* 44 and *m/z* 60 signals with extent of oxidation in the PAM chamber. Chromatogram-binning PMF techniques (Zhang et al., 2014, 2016) were also applied to the decomposition window to investigate the presence and covariance of key ion signals. Finally, trends in TAG and AMS *m/z* 60 signals with PAM aging were explored to evaluate the utility of *m/z* 60 as a tracer for freshly-emitted BBOA. We present evidence that, depending on biomass source and atmospheric conditions, a significant fraction of AMS *m/z* 60, which is typically used to track primary BBOA in the atmosphere, may be attributed to aged OA mass.

## 2 Materials and Methods

### 2.1 Emissions and Combustion Chamber

A flow diagram of the experimental setup and a diagram of the custom-built emissions and combustion chamber are given in Supplemental Information (Figures S1 and S2, respectively). A complete description of the emissions and combustion chamber is available elsewhere (Mellott, 2012). The chamber is a rectangular 1.48 m$^3$ chamber made of 0.25-inch-thick tempered glass panels secured by aluminum framing (80/20, Inc., Columbia City, IN). The chamber is divided into two compartments, separated by an aluminum baffle with a central hole 3 cm in diameter. In the first compartment, biomass samples are resistively heated in proportional-integral-derivative- (PID) controlled stainless-steel cups installed along the chamber floor. The second compartment serves as a mixing chamber from which primary gases and particles are sampled at 10 L min$^{-1}$. Air was treated with a HEPA filter (Pall Corporation, Port Washington, NY) and a hydrocarbon trap (Model BHT-4, Agilent Technologies, Santa Clara, CA), then supplied to the heating compartment of the chamber to promote mixing. Both compartments are extensively vented between experiments to clear the chamber of gases and particles.

### 2.2 Devolatilization and Combustion Experiments

White oak (*Q. alba*) heartwood and leaves were chosen for these studies because of their abundance in the oak-hickory forests of Missouri and the southeastern United States. Although comparing different tree species is also of interest, two different plant fractions of the same species are studied here to investigate different types of wildfire or controlled

combustion processes, some of which may only impact leaf litter-fall and others would have wood available as a fuel. The white oak biomass samples used in this study were collected at the Tyson Research Center in Eureka, MO, located approximately 20 miles outside of St. Louis, MO. An oak trunk segment was taken from the site, and heartwood

samples were collected by drilling into the center of the trunk segment. Oak leaves were clipped from a single branch that was taken directly from a live tree. The leaf samples were air-dried for at least one week and milled into fine pieces using a tobacco grinder prior to running the experiment. All biomass was stored at room temperature (20-25°C), and moisture content was not controlled for either fuel type.

Samples of oak heartwood or leaf were pre-weighed (0.2-0.5 g), placed in the emissions chamber cup, and spread

evenly across the bottom rim. The cup was heated for 3.5 minutes, with typical ignition temperatures of 300°C. In this work, we use the term "devolatilization" to describe the non-combustive release of emissions from biomass fuels at elevated temperatures. During the heat pulse, the biomass sample was first devolatilized, with smoldering embers observed in the final minute of the heat ramp. No flaming combustion occurred during any of the emissions experiments.

To ensure the TAG and AMS collected particles within a similar size range, primary emissions were passed through a $PM_1$ cyclone (Thermo Fischer Scientific, Waltham, MA) operated at 16.7 L min$^{-1}$ to remove particles too large to be sampled by the AMS (DeCarlo et al., 2006). Because dilution drives partitioning of SVOCs and IVOCs from the particle phase into the gas phase in BBOA plumes (Grieshop et al., 2009; Ortega et al., 2013), dilution was minimized in the system during devolatilization and combustion experiments. Dilution air, purified using separate zero air

generators (Model 737, Aadco Instruments, Cleves, OH), was supplied before the $PM_1$ cyclone (6.7 L min$^{-1}$) and after the PAM chamber (4 L min$^{-1}$) to provide sufficient flow to the cyclone and to all instruments (Figure S1), giving a net dilution ratio of approximately 5 for all experiments.

### 2.3 PAM Reactor Operation

Particulate and gas-phase emissions were treated together in the PAM flow reactor. A detailed description of the PAM

reactor is given elsewhere (Kang et. al. 2007, Lambe et. al. 2011). The reactor consists of a 13 L cylindrical aluminum chamber coated internally with Iridite 14-2 (MacDermid, Inc., Waterbury, CT), a chromate conversion film designed to decrease charge buildup and thereby inhibit losses of charged particles to the walls of the reactor. Within the PAM chamber, low-pressure mercury lamps emit light at two wavelengths (185 nm and 254 nm) in the UV range, and different OH concentrations are produced by adjusting the intensity of the UV irradiation (Kang et al., 2007). Ozone

($O_3$) is produced externally by irradiating 0.4 L min$^{-1}$ of pure $O_2$ with mercury lamps ($\lambda$ = 185 nm; BHK, Inc., Ontario, CA) to produce 4 ppm externally added $O_3$. Water vapor is introduced into the PAM reactor with 4.6 L min$^{-1}$ of humidified $N_2$. A total flow rate of 10 L min$^{-1}$ was maintained throughout the experiments, giving an average residence time of 78 seconds within the reactor. To achieve consistent OH formation, the relative humidity (RH) inside of the reactor was kept at 30.0% ± 3.7% (one standard deviation), measured with a relative humidity and temperature probe

with manufacturer-specified accuracy of 1.5% (Vaisala, Inc., Woburn, MA). The reactor water concentration, and therefore RH, was altered by controlling $N_2$ flow through a Nafion membrane humidifier (Perma Pure LLC,

Lakewood, NJ). The role of water concentration in OH formation is discussed in detail in Supplemental Information (Method: PAM Calibrations and Equivalent Aging Estimations, Figure S3).

OH exposures ($OH_{exp}$) within the PAM reactor were calculated using the offline sulfur dioxide ($SO_2$) calibration method described in previous work (Kang et al., 2007). During reactor calibration, $SO_2$ concentrations (Airgas, Inc., Radnor, PA) were measured with an $SO_2$ monitor (Model 43i-TLE analyzer, Thermo Fischer Scientific, Waltham, MA) at varied UV lamp intensities; similarly, $O_3$ was measured downstream of the PAM reactor by UV photometry (Model 49i, Thermo Fischer Scientific, Waltham, MA). Equivalent atmospheric aging times from the $SO_2$ calibrations were calculated assuming an average atmospheric OH concentration of $1.5 \times 10^6$ molec $cm^{-3}$ (Mao et al., 2009) and are provided as the upper limit on the equivalent aging time ranges obtained for the system (Table 1). PAM reactor calibration details and results are provided in Supplemental Information (Methods: PAM Calibrations and Equivalent Aging Estimations). For both heartwood and leaf fuels, experiments were performed at two level of photochemical aging in addition to a baseline without OH exposure. Henceforward, the different photochemical aging conditions will be denoted by the corresponding equivalent aging time ranges (Table 1).

Previous PAM reactor studies have demonstrated that high concentrations of volatile organic compounds (VOCs) can suppress OH reactivity (Li et al., 2015; Peng et al., 2015). This suppression occurs because VOCs drive rapid conversion of OH to $HO_2$, and recycling of $HO_2$ back to OH can be slow without addition of sufficient $O_3$ (Peng et al., 2015, 2016). External OH reactivity ($OHR_{ext}$, $s^{-1}$) is defined as the sum of the products of concentrations of externally reacting species ($C_i$ for a compound $i$) and corresponding OH reaction rate constants ($k_i$; Peng et al., 2016):

$$OHR_{ext} = \sum k_i C_i \tag{1}$$

This metric is used to describe the potential for interfering gases to react with OH and suppress heterogeneous oxidation. The external production of $O_3$ featured in our system is expected to reduce OH suppression by introducing additional $O_3$ to promote recycling of $HO_2$ back to OH (Peng et al., 2015).

Due to a lack of gas-phase measurements, $OHR_{ext}$ values were not calculated during TAG and AMS collections. However, supplementary experiments were conducted to approximate $OHR_{ext}$ by repeating the fuel burning procedure and measuring resulting CO emissions with a CO monitor (Peak Laboratories, Mountain View, CA). During these experiments, emissions were sampled alternately through the PAM chamber, set to approximately 3 days of equivalent aging according to the most recent offline $SO_2$ calibration, and a bypass line. We observed little difference in CO $OHR_{ext}$ between PAM-aged emissions (maximum $OHR_{ext}$ = 0.558 $s^{-1}$) and bypassed emissions (maximum $OHR_{ext}$ = 0.516 $s^{-1}$). Additionally, we estimated total $OHR_{ext}$ by scaling trace gas emission factors (EFs) from previous laboratory-generated oak biomass combustion VOC measurements (Burling et al., 2010) to our measured CO concentrations. Using this method, we approximate a total $OHR_{ext}$ of 2.2 $s^{-1}$. This $OHR_{ext}$ value is assumed for subsequent $OH_{exp}$ and equivalent aging estimations. A detailed description of the experimental methods, as well as a discussion of the limitations of this $OHR_{ext}$ estimation approach, is available in Supplemental Information (Methods: PAM Calibrations and Equivalent Aging Estimations, "Estimation of External OH Reactivity ($OHR_{ext}$)"). Averaged CO concentrations for aged and unaged leaf BBOA are provided in Figure S4.

Based on an RH of 30%, a typical internally-produced output $O_3$ range of 0.3-1.7 ppm (measured during reactor calibrations), and an $OHR_{ext}$ of 2.2 s$^{-1}$, we estimated $OH_{exp}$ ranges for each PAM UV light setting using the Oxidation Flow Reactor Exposure Estimator version 2.3 developed by Peng et al., available for download at http://sites.google.com/site/pamwiki/hardware/estimation-equations (Peng et al., 2015, 2016). Results obtained using this spreadsheet are given in Supplemental Information (Table S1). The "condition type," which indicates whether VOC suppression is significant under the input conditions, was found to be "safer," indicating that chemical interferences from VOCs are minimal based on input measurements and assumptions.

Flow field simulations and chemical tracer tests have demonstrated that the PAM reactor used in this study is approximately well mixed if sufficient time (at least 15 minutes) is given prior to sample collection to establish a well-mixed and near steady-state concentration throughout the combustion chamber and PAM chamber (Mitroo, 2017; Mitroo et al., 2017). The TAG therefore consistently collected 30 minutes after the biomass heat pulse to minimize particle concentration gradients within the reactor.

Photobleaching of BBOA, particularly at 254 nm, has been reported in previous literature (e.g., Sumlin et al., 2017; Wong et al., 2017; Zhao et al., 2015) and therefore should be considered when estimating oxidative aging. With the spreadsheet provided by Peng et al., we estimate 254 and 185 nm exposure ratios (ratio of photon flux, photons cm$^{-2}$, to $OH_{exp}$; Peng et al., 2016) to be $1.2 \times 10^5$ cm s$^{-1}$ and $8.1 \times 10^2$ cm s$^{-1}$, respectively, at a measured internally-generated $O_3$ concentration of 1.7 ppm (at the highest PAM UV lamp intensity), a water mixing ratio of 1% (RH = 30%), and assuming a maximum $OHR_{ext}$ value of 1 (Peng et al., 2016). Using Figures 1 and 2 of Peng et al., 2016 to interpret these values, we find that at both 185 nm and 254 nm, photolysis rates are likely less than 10% for species of interest.

### 2.4 Instrumentation and Data Analysis

The TAG and the AMS were used to collect complementary chemical composition data. A Scanning Mobility Particle Sizer (SMPS; Model 3081 DMA, Model 3022A CPC, TSI, Inc., Shoreview, MN) was used to measure aerosol size distributions and volume concentrations.

The devolatilization and combustion experiments were performed in two distinct experimentation periods. In the first period, the procedure was done at each level of PAM oxidation using 0.2 g biomass. Triplicate experiments were done with the TAG and the SMPS during this period to ensure repeatability of the devolatilization and combustion cycle. In the second experimentation period, experiments were performed once more at each level of oxidation to obtain simultaneous TAG, SMPS, and AMS measurements. For these experiments, the devolatilization and combustion procedure was done with more biomass fuel (0.5 g) so the AMS could obtain sufficient signal.

### 2.4.1 Thermal Desorption Aerosol Gas Chromatograph (TAG)

A full description of the TAG system is provided in previous literature (Williams et al., 2006). Particles are collected via humidification and inertial impaction at a typical flow rate of 9.3 L min$^{-1}$, with a particle cutoff ($d_{p50}$) of approximately 70 nm (Williams et al., 2006). Following sample collection, the collection and thermal desorption (CTD) cell is heated to 310 ℃ at a typical rate of 50℃ min$^{-1}$ to thermally desorb the collected OA. The desorbed

sample is flushed through a heated transfer line over helium and transported to a gas chromatography column for separation and mass spectral detection. An Agilent 6890 GC (Agilent Technologies, Santa Clara, CA) with a 30m-long 0.25mm i.d. RTX5-MS non-polar fused silica capillary column (Restek Corporation, Bellefonte, PA) was used to achieve chromatographic separation. A 70 eV electron ionization quadrupole mass spectrometer (5973 MSD, Agilent Technologies, Santa Clara, CA), operated to scan between 29-450 $m/z$, provided mass spectral detection. TAG performance was evaluated regularly (once every 1-3 days) using 5 ng $C_{12}$-$C_{40}$ even alkane standard mixture (Sigma Aldrich, St. Louis, MO) manually injected onto the CTD cell and thermally desorbed onto the GC column via a helium carrier stream (Kreisberg et al., 2009).

The TAG system developed by Isaacman et al. features an online derivatization technique designed to improve analysis of oxidized species, including methoxyphenols, levoglucosan, and other compounds unique to BBOA (Isaacman et al., 2014). Although this technique presents multiple analytical advantages, it was developed for a metal filter collection cell and is not suitable for the impactor-style CTD cell used in these experiments. We chose to use the impactor-style CTD cell to allow analysis of the thermal decomposition window, since other collection cells purge this material when transferring to a secondary trap. Additionally, we were interested to identify new molecular marker compounds that could be associated with these source types. We therefore performed all experiments without sample derivatization prior to chromatographic analysis.

TAG data were collected during the first experimentation period using 0.2 g biomass in the heat pulse. For all the oak leaf and heartwood experiments, particles were collected on the TAG for four minutes, thirty minutes after the heat pulse was performed in the emissions chamber. The TAG collected two additional samples over the course of three hours to ensure that both the emissions chamber and the PAM reactor were clean prior to the subsequent devolatilization cycle.

In this work, the TAG compound and thermal decomposition time windows were analyzed as complementary sets of chemical data (Figure 1). As defined for this study, the thermal decomposition window occurs between minutes 6-16 of GC analysis, which coincides with the thermal desorption of sample from the CTD cell. The compound window consists of material eluting from minutes 20-55 of analysis following condensation of desorbed sample at the column head. This window contains information on OA components that have been successfully desorbed, transferred, and separated.

Prior to each experiment, a system blank chromatogram was obtained by sampling from the empty emissions chamber through the PAM reactor, with the PAM UV lamps set to the voltage corresponding to the subsequent equivalent aging time to be tested. A system blank was subtracted from each chromatogram prior to data processing to correct for both TAG system artifacts (e.g., air signal and column bleed) and sampling system (PAM reactor and emissions and combustion chamber) artifacts. Additionally, to isolate changes in aerosol chemical properties from changes in aerosol mass with photochemical aging, each blank-subtracted chromatogram was normalized to volume concentration by dividing the abundance at each scan by the maximum volume concentration ($nm^3$ $cm^{-3}$) obtained by the SMPS for each devolatilization cycle (Table S2 and Figure S5 in Supplemental Information). This blank subtraction and

normalization process was done for all total ion count (TIC) chromatograms and single-ion chromatograms (SICs) presented in this work.

### 2.4.2 TAG Positive Matrix Factorization

Positive matrix factorization (PMF) was performed on TAG chromatograms to identify source-specific major
compounds and compound classes present in the heartwood and leaf BBOA. TAG chromatograms were binned by
retention time according to the method outlined in previous work (Zhang et al., 2014, 2016). Prior to chromatogram
binning, each chromatogram was blank-subtracted to minimize the contribution of background noise in PMF
calculations. An instrument error of 10%, chosen based on a typical average TAG instrument error of 10% (Williams
et al., 2006), was assumed during PMF calculations.

The GC-resolved mass spectral PMF method for binned TAG data was developed to separate compounds in TAG
chromatograms into chemically similar factors, improving analysis efficiency (Zhang et al., 2014). With this method,
mass spectral data is supplied to the PMF model, and solutions are obtained using the PMF2 algorithm (Paatero, 1997).
Each resulting factor consists of a mass spectrum corresponding to a compound or class of compounds present in the
TAG chromatograms (Zhang et al., 2014). This PMF method was performed on the compound and decomposition
analytical windows separately for data obtained from both BBOA types. PMF output and solutions were evaluated
using custom-built pre- and post-processing analysis software in conjunction with the PMF Evaluation Tool (version
3.00A; Ulbrich et al., 2009) in Igor Pro version 6.38Beta01 (WaveMetrics, Inc.). Mass spectral identification of
different factors was aided by the NIST MS Search Program version 2.0, available for download at
http://chemdata.nist.gov/mass-spc/ms-search/.

The number of appropriate PMF factors was determined for each solution based on two considerations. First, in a
typical PMF analysis, the optimal number of factors in a solution is selected based on the objective function $Q$, which
is the sum of weighed squared residuals (Paatero, 1997). The $Q/Q_{exp}$ value, or the ratio of the actual objective function
to the expected objective function assuming normally distributed residuals, should ideally approach 1; too few factors
may result in a large $Q/Q_{exp}$, indicating that errors have been underestimated in PMF calculations (Ulbrich et al., 2009).
Additionally, if too many factors are specified, the solution may feature split factors, where information from a
compound or compound class is distributed across multiple factors. In this work, the number of factors presented for
each analysis was selected to minimize split factors while maximizing identifiable factors. Because of the TAG data's
high chromatographic resolution, low rotational ambiguity was assumed, and all calculations were performed with
$f_{peak} = 0$. This assumption is supported by previous work, where TAG data were not sensitive to $f_{peak}$ or starting point
(seeds) during PMF analysis (Williams et al., 2010).

### 2.4.3 AMS

The AMS data presented in this work were obtained using 0.5 g of biomass in the heat pulse instead of 0.2 g to ensure
the AMS received sufficient signal. The AMS was operated in V-mode throughout all experiments (DeCarlo et al.,
2006). AMS data were processed in Igor Pro version 6.38Beta01 using the SQUIRREL version 1.57 toolkit for unit

mass resolution analysis and the PIKA version 1.16 toolkit for high resolution analysis. Both AMS data analysis tools

are available for download at http://cires1.colorado.edu/jimenez-group/ToFAMSResources/ToFSoftware/index.html.

## 3 Results and Discussion

### 3.1 AMS Measurements

Average AMS mass spectra and van Krevelen plots are provided in Supplemental Information (Figures S6 and S8,

respectively). In addition, AMS measured concentrations of key species, including total organics, sulfate, and

potassium ($K^+$), are provided in Figure S7 and Table S3.

According to AMS mass spectra, the BBOA measured in these experiments is chemically consistent with BBOA from

similar oak fuel sources, though with key differences related to combustion conditions (Cubison et al., 2011; Ortega

et al., 2013; Reece et al., 2017; Weimer et al., 2008). Detailed analysis and contextualization of the AMS chemical

composition data is given in Supplemental Information (Section: AMS Chemical Characterization).

### 3.2 Individual Compound Analysis

The mass spectral dot product method proposed by Stein and Scott was used to determine chemical similarity between

each chromatogram and to evaluate inter-test variability. For each blank-subtracted TAG chromatogram, a summed

mass spectrum was obtained by summing all ions ($m/z$ 33-$m/z$ 450) across all scans (retention times) in the

chromatogram and converting the resulting mass spectral vector into a unit vector. To assess the similarity of two

mass spectra, the dot product of the mass spectral unit vectors was calculated; a dot product of 1 signifies a perfect

mass spectral match, and a dot product of 0 indicates a complete mismatch (Stein and Scott, 1994). Within a fuel type

and an oxidation condition, the dot product was assessed for two TAG chromatograms at a time for a total of 3 dot

product values. These values are given in Table S4.

For both leaf and heartwood BBOA, key molecules identified within the compound window of the TAG

chromatograms are given in Supplemental Information (Table S5). Corresponding molecular structures for the

compounds used in individual compound analysis are also provided (Figure S9). Identification certainty ("Certainty

of ID") was classified for each compound according to the following criteria: A) the compound was positively

identified based on external standard injections; B) the compound was identified based on a high match quality (MQ

> 75%) using available mass spectral libraries; C) the compound was identified based on a low-to-moderate match

quality (MQ < 75%) using available mass spectral libraries; and D) no adequate mass spectral library match was

available for the compound, so the compound structure was inferred by retention time and manually evaluating

possible fragmentation patterns. Identification method (D) was particularly relevant for long-chain aliphatic

compounds, including alkenes and even-carbon aldehydes. For these compounds, the parent ion was first determined,

then major ions were identified (e.g., in tetracosanal, $m/z$ 334 corresponds to $C_{24}H_{46}^+$ following loss of $H_2O$). The

feasibility of the identified structure was confirmed based on predicted vapor pressures and retention times from even

alkane standards.

Subcooled liquid vapor pressures at 25°C were predicted for each compound using the Advanced Chemistry Development (ACD/Labs) Software V11.02 (© 1994-2017 ACD/Labs), available for use on the SciFinder website (ACD/Labs, 2017).

### 3.2.1 Trends in Individual Compounds with Photochemical Aging

Leaf and heartwood BBOA chromatograms at three levels of photochemical aging are overlaid for comparison in Figure 2. Raw peak integration values with standard deviations are provided for each compound at each level of equivalent aging are also provided (Table S6). Each chromatogram constitutes an average of the triplicate blank-subtracted measurements, with each chromatogram normalized to the maximum total volume concentration measured during the experiment. For these plots, the averaged, normalized chromatograms at each level of aging were further normalized to the point of highest abundance in the unaged ("0 days") average chromatogram. In the leaf BBOA chromatograms (Figure 2a), many of the low volatility species eluting after minute 35 of the GC analysis are long-chain alkanes, alcohols, aldehydes, and terpenoids, compounds commonly found in the leaf's waxy exterior coating (Gulz and Boor, 1992). Based on even-numbered alkane standard injections, compounds eluting after minute 35 exhibit approximate saturation vapor pressures not exceeding that of docosane (approximately $2.73 \times 10^{-5}$ torr at 25ºC), which corresponds approximately to $\log_{10}(C^*) = 2.76$ (Table S5 in Supplemental Information; ACD/Labs, 2017).

To illustrate the relative rates of decay that each compound experiences in the PAM reactor, Figure 3a provides integrated abundances for nine compounds of interest. The integrated abundances were first normalized to appropriate volume concentrations, then to the corresponding abundances at no oxidation. Nearly all compounds identified after 35 minutes decrease in relative abundance with photochemical aging. Notably, we have identified an even-carbon aliphatic aldehyde series based on $[M-18]^+$ and $[M-28]^+$ (where M is the parent mass) peaks present in the mass spectra of each of the compounds (Watson and Sparkman, 2007). As the carbon number ($n_C$) increases, the aldehyde abundance decreases more readily with oxidation. To our knowledge, rate constants for the reaction of long-chain ($n_C \geq C_{20}+$) condensed-phase aliphatic aldehydes with OH have not been reported. However, previous studies on short-chain ($n_C \leq C_{14}$) condensed-phase aliphatic aldehydes demonstrate that OH reaction rate constants increase with increasing carbon chain length (D'Anna et al., 2001; Niki et al., 1978). Although aliphatic aldehydes, particularly $C_{26}$ and $C_{28}$ aldehydes, have been characterized as components of oak leaf waxes (Gulz and Boor, 1992), these aldehydes have not been reported as components of oak leaf BBOA and may therefore serve as novel tracer species in future field experiments. To confirm the presence of aldehydes in the leaf waxes, solvent extractions were performed on oak leaves and were manually injected onto the TAG CTD cell (Method: Oak Leaf Solvent Extractions and Figure S10 in Supplemental Information). Analysis of these extractions confirm that the aldehydes are present in the leaf wax prior to devolatilization and combustion.

Literature information available for hydrocarbon particle- and gas-phase OH kinetics indicates that the trends observed in leaf BBOA alkane and aldehyde abundances are consistent with heterogeneous OH oxidation. For example, Smith et al. report approximately 70% decay of squalane (a $C_{30}$ branched alkane) particles when exposed to an $OH_{exp}$ of 1.1 $\times 10^{12}$ molec cm$^{-3}$ s$^{-1}$ (approximately 10 days of equivalent aging; Smith et al., 2009), a figure approximately consistent

with the observed $C_{29}$ alkane decay of 75% at 6-10 days of equivalent aging. Additionally, based on parameters provided by Kwok and Atkinson, gas-phase OH reaction rate constants at 298K are estimated to be $2.5 \times 10^{-11}$, $2.7 \times 10^{-11}$, and $3.1 \times 10^{-11}$ cm³ molec⁻¹ s⁻¹ for $C_{23}$, $C_{25}$, and $C_{29}$ alkanes, respectively (Kwok and Atkinson, 1995). Taking these rate constants into account, if purely gas-phase chemistry is assumed, all three alkanes would react nearly 100% before 1-3 days of equivalent aging. A similar analysis on relevant aldehydes gave estimated gas-rate constants of $2.5 \times 10^{-11}$, $2.8 \times 10^{-11}$, and $3.0 \times 10^{-11}$ cm³ molec⁻¹ s⁻¹ for $C_{24}$, $C_{26}$, and $C_{28}$ aldehydes, respectively (Kwok and Atkinson, 1995), which in all cases would lead to complete depletion by 1-3 days of equivalent aging if gas-phase chemistry is assumed.

Compounds characteristic of heartwood primary BBOA are typically more volatile than those found in the leaf primary BBOA, eluting between minutes 28 and 35 of the GC analysis (Figure 2b). Based on even alkane standard injections, compounds eluting within this time window exhibit approximate vapor pressures within $4.52 \times 10^{-3}$-$2.73 \times 10^{-5}$ torr at 25ºC ($\log_{10}(C^*) \approx 4.85$-$2.76$; Table S5 in Supplemental Information; ACD/Labs, 2017). The compound with the highest abundance in unoxidized wood BBOA chromatograms is sinapaldehyde (4-hydroxy-3,5-dimethoxycinnamaldehyde), a phenolic compound derived from lignin. Of the compounds examined, sinapaldehyde decays most rapidly in the PAM reactor, with the normalized average integrated peak area decreasing by approximately 70% from 0 days to 1-3 days of equivalent aging (Figure 3b). Based on a rapid gas-phase OH reaction rate constant of $2.7 \times 10^{-12}$ cm³ molec⁻¹ s⁻¹, the observed sinapaldehyde decay is likely occurring in the particle phase. Other compounds, including methyl-β-D-glucopyranoside, galactoheptulose, and acetylgalactosamine, also exhibit decreases in abundance. Relative rates of decay for these and other wood BBOA tracers are given in Figure 3b.

Syringol (2,6-dimethoxy-phenol), syringaldehyde (4-hydroxy-3,5-dimethoxy-benzaldehyde), and vanillin (4-hydroxy-3-methoxy-benzaldehyde) increase in abundance from 0 days to 1-3 days of equivalent aging and are depleted with 6-10 days of equivalent aging. Since the average volume concentration for runs at 1-3 days of aging were larger than those at 0 days of aging by a factor of approximately 1.3 (Table S2 in Supplemental Information), the factor of ~2 increase in syringol and syringaldehyde integrated abundances could occur due to partitioning from the gas phase into the particle phase. To estimate phase partitioning for these compounds, particle-phase fractions for syringol, syringaldehyde, and vanillin ($\xi_i$) were calculated based on AMS total organic concentrations ($C_{OA}$, µg m⁻³; Table S3 in Supplemental Information) and effective saturation concentrations ($C_i^*$, µg m⁻³) using a basic partitioning equation (Donahue et al., 2006; Table S5 in Supplemental Information):

$$\xi_i = \left(1 + \frac{c_i^*}{c_{OA}}\right)^{-1} \tag{2}$$

Resulting particle-phase fractions are tabulated in Supplemental Information (Table S7). Based on these approximations, syringol, syringaldehyde, and vanillin are expected to partition primarily to the gas phase. For these compounds, the increase in abundances at low-mid levels of oxidation could therefore result from increased SOA formation driving these compounds into the particle phase. This observation is consistent with previous measurements where maximum SOA concentrations were observed at similar levels of $OH_{exp}$ for aerosol generated from oxidation of a single precursor (Lambe et al., 2012; Ortega et al., 2016).

Although phase partitioning may contribute to the trend in vanillin with photochemical aging, the nearly eight-fold increase in vanillin integrated abundance from 0 days to 1-3 days of aging could suggest an alternative formation mechanism driven by reactions occurring in the PAM reactor. One potential mechanism for the formation of aldehydes from larger lignin decomposition products involves the cleavage of the $C_\alpha$-$C_\beta$ unsaturated bond on the benzyl substituent following formation and fragmentation of a peroxide radical intermediate (Wong et al., 2010; Figure S11 in Supplemental Information). The presence of OH in the PAM reactor may drive a similar process, leading to increases in vanillin abundance at moderate $OH_{exp}$.

### 3.2.2 Compound Window PMF Analysis

GC-MS PMF results are provided for both leaf and wood BBOA chromatograms using data collected within the TAG compound window (Figures 4 and 5). $Q/Q_{exp}$ and residual plots are provided in Supplemental Information (Figures S12 and S13, respectively). The chromatograms are displayed as averages of binned data from triplicate measurements at each level of oxidation and are displayed in one trace; different equivalent aging times are demarcated with vertical lines along the x-axis. Corresponding mass spectra are identified and displayed with key ions labeled. High factor solutions ($\geq 15$) were used for compound window data to best deconvolve the large and complex mixture of compounds. However, in some cases, factor splitting resulted in the distribution of ions between two or more factors, made evident by similarities in retention times. Wherever possible, split factors were recombined by summing the binned chromatograms and the mass spectra and are labeled accordingly (e.g., "F10+F12" indicates that factor 10 and factor 12 have been recombined). In general, for the compound window, factor solutions were chosen to maximize the number of identifiable factors while minimizing the number of split factors.

A 15-factor solution was chosen to deconvolve leaf BBOA compound window chromatograms (Figure 4; additional information provided in Figures S12a and S13a). This solution provided enough factors to resolve the lowest-abundance components (e.g., F1), and increasing the number of factors past 15 led to greater factor splitting without providing additional insight into the chromatograms. Among the factors identifiable with this solution include quinic acid (Factor 2, F2), sugars and anhydrosugars (e.g., mannose; F3), alcohols and alkenes (F6), aldehydes (F10), terpenoids (e.g., friedelin; F11), and column bleed (F13+F14). Other factors (F1, F5+F7, F9+F12, F15) correspond to different classes of unresolved complex mixture (UCM) and have been tentatively identified by considering the closest matches in the NIST mass spectral database. Factor 4 (F4) is identified as a split factor, exhibiting mass spectral characteristics of multiple factors, including acids ($m/z$ 129) and anhydrosugars ($m/z$ 116). Factors 13 and 14 demonstrate contributions from both terpenoid-like UCM and column bleed and are therefore combined. The presence of alkylbenzenes (F8), dominated by $m/z$ 91 ($C_7H_7^+$) and $m/z$ 92 ($C_7H_8^+$), is noteworthy, as alkylbenzenes are typical of anthropogenic materials (e.g., detergent precursors produced from petroleum; Forman et al., 2014) and have not been reported as components of biomass. Since the leaves were not cleaned after they were collected, the alkylbenzenes could come from deposition of fuel combustion aerosol onto the leaves' surface prior to biomass sample collection. The presence of alkylbenzenes on the surface of the leaf was confirmed with TAG analysis of solvent-extracted leaf surface components (Figure S14), supporting the interpretation of deposition of anthropogenic compounds on the leaf's exterior.

An 18-factor solution was applied to deconvolve compounds in the wood BBOA chromatograms (Figure 5; additional information provided in Figures S12b and S13b). Notable factors correspond to levoglucosan (F1), guaiacol (F4), vanillin and guaiacyl compounds (F7), syringol (F8), syringaldehyde (F10), sinapaldehyde (F11), and column bleed (F18). Based on retention time and mass spectral characteristics (e.g., $m/z$ 77), factor 5 (F5) corresponds to aromatic species and is not matched to a single compound. Factor 6 (F6) is featured in multiple aromatic compounds, but is

also present in levoglucosan in very low abundances. Several types of UCM (F2, F3, F9, F12+F13+F14, F15, F16) were deconvolved and tentatively identified using the top matches from the mass spectral database. Factor 16 (F16) is predominated by siloxanes (e.g., $m/z$ 73, $m/z$ 281, $m/z$ 341), though some UCM has been split from other factors. Finally, factor 17 (F17) exhibits characteristics of multiple classes of compounds and is therefore identified as a split factor.

Nearly all factors obtained in the leaf BBOA compound window analysis decrease with photochemical aging, including quinic acid (F2), sugars and anhydrosugars (F3), alkanes and long-chain aliphatics (F6, F10, F15), alkylbenzenes (F8), terpenoid components (F11), and various classes of UCM (F1, F4, F5+F7, F9+F12). This trend agrees well with the individual compound analysis and further indicates that primary components undergo increased fragmentation at higher $OH_{exp}$. In the heartwood BBOA, some primary components decrease steadily with

photochemical aging, including sinapaldehyde (F11), aromatics (F5), and various classes of UCM (F12+F13+F14, F15, F17). Other factors, including guaiacol (F4), vanillin (F7), syringol (F8), and syringaldehyde, exhibit a strong increase in abundance at 1-3 days of aging followed by a decrease at 6-10 days of aging, possibly due to changes in partitioning as described previously. Levoglucosan (F1) also appears to increase slightly in abundance at 1-3 days of equivalent aging, though this is likely due to differences in aerosol mass produced between experiments. Results from

both types of BBOA show changes in column bleed (F13+14 and F18 for leaf and wood BBOA, respectively) from unaged chromatograms to 6-10 days of aging. Although the column bleed decreases with photochemical aging in both cases, this trend is due to differences in blank subtractions from run to run and is not related to changes in photochemical aging.

**3.3 TAG Thermal Decomposition Window**

The TAG thermal decomposition window has been used in previous work to assess contributions of inorganic (nitrates, sulfates, etc.) and organic species present in atmospheric aerosol (Williams et al., 2016). In this work, we provide evidence that the TAG thermal decomposition window can be used to evaluate the relative level of oxidation of bulk OA samples using the $m/z$ 44 ($CO_2^+$) ion. In addition, we demonstrate that other fragments within the decomposition window may give insight into the chemical composition of aged, thermally labile BBOA.

Replicable, quantitative TAG data were not obtained during experiments that used 0.5 g biomass, potentially due to a minor system leak. However, the TAG chromatograms that were obtained using 0.5 g biomass were chemically similar to the triplicate TAG chromatograms obtained using 0.2 g biomass, and we therefore compare all AMS data with TAG chromatograms collected using 0.2 g biomass in subsequent analysis. Chemical similarity between chromatograms was confirmed using the dot product mass spectral comparison method outlined by Stein and Scott (Stein and Scott,

1994). The dot product was determined for two chromatograms, one obtained with 0.5 g biomass and one obtained

with 0.2 g biomass, at each level of oxidation. The resulting dot products for both leaf and wood oak are all above 0.75 and are provided in Supplemental Information (Figure S15; Table S8).

### 3.3.1 *m/z* 44 as a Tracer for Aged OA

Figures 6a and 6b show *m/z* 44 TAG decomposition SICs for leaf and wood BBOA, respectively. Raw SICs, along with blanks, are provided in Figure S16. At each oxidation condition, SICs from the triplicate chromatograms were blank subtracted, normalized to maximum volume concentrations, and averaged to obtain the displayed trace. Within each plot, the chromatograms have been further normalized to the point of highest abundance within the unaged ("0 days") *m/z* 44 signal. The *m/z* 44 signals were also summed across the entire decomposition window following blank subtraction, normalization to appropriate volume concentrations, and triplicate averaging, and are provided as functions of equivalent aging time (± one standard deviation) in Figure 6c. The upward trend in the *m/z* 44 signal between minutes 6 and 10 of GC analysis coincides with the CTD temperature ramp from 45°C to 310°C, and is thus consistent with gradual increase in OA thermal decomposition as the temperature rises. The subsequent decrease in *m/z* 44 signal from minute 10-16 reflects the thermal decomposition of remaining material as the CTD cell is held at 310°C. For both types of BBOA, the decomposition *m/z* 44 integrated signal increases overall from 0 days to 6-10 days of equivalent aging, indicating an increase in OA material that can thermally decompose with increased PAM oxidation. This trend is consistent with relative increased decomposition of highly oxidized aerosol formed within the PAM reactor, as demonstrated in previous ambient aerosol observations (Williams et al., 2016). In the leaf BBOA chromatograms, the increase in integrated *m/z* 44 signal is most pronounced from 0 to 1-3 days of equivalent aging, while the heartwood BBOA data exhibits the most dramatic increase from 1-3 to 6-10 days. The variation in the shape of the decomposition *m/z* signal between the two types of biomass likely reflects differences in thermal lability between different types of OA.

AMS $\overline{OS_C}$ values calculated for both types of biomass range from -1.5 to -0.2 (Figure 7). In both types of BBOA, an increase in relative integrated TAG decomposition *m/z* 44 signal coincides with an increase in $\overline{OS_C}$ from 0 to 6-10 days of photochemical aging. A linear correlation between decomposition *m/z* 44 and AMS $\overline{OS_C}$ for wood BBOA ($r^2$ = 1) indicates that under these experimental conditions, the TAG thermal decomposition window has the potential to provide quantitative measurements of bulk OA oxidation levels. By contrast, leaf BBOA decomposition *m/z* 44 and AMS $\overline{OS_C}$ correlate poorly ($r^2 = 0.8$ for a linear fit). The non-linear trend in TAG decomposition *m/z* 44 for leaf BBOA may indicate a shift in the dominant oxidation mechanisms between moderate and high levels of OH within the PAM chamber; at the highest $OH_{exp}$, primary gas and/or particle-phase components may undergo increased fragmentation, leading to a net decrease in production of the aged OA that thermally decomposes during TAG analysis, along with an increase in highly volatile fragmentation products that are not captured by the TAG. However, the mechanisms behind this trend remain unclear and merit further investigation.

For each fuel type, AMS $f_{44}$ vs $f_{43}$ data have been plotted at each level of equivalent aging (Figure 8). To further explore the TAG's analytical capability in relation to AMS bulk chemical data, TAG integrated ion fractions ($f_{ion}$) are

also provided in these plots. These fractions are defined as the blank-subtracted integrated ion signal divided by the blank-subtracted integrated TIC signal. For example, for a chromatogram $i$, the TAG $f_{44}$ signal is defined as:

$$f_{44,i} = \frac{(A_{44})_i - (A_{44})_{blank}}{(A_{TIC})_i - (A_{TIC})_{blank}}$$

(3)

where $(A_{44})_i$ is the integrated $m/z$ 44 signal across all (i.e., TAG total chromatogram) or part (i.e., TAG compound window) of $i$, $(A_{44})_{blank}$ is the integrated $m/z$ 44 signal across a blank chromatogram, $(A_{TIC})_i$ is the integrated TIC across all or part of $i$, and $(A_{TIC})_{blank}$ is the integrated TIC across the same blank. For heartwood BBOA, although AMS $f_{44}$ increases and $f_{43}$ decreases with photochemical aging, both TAG $f_{44}$ and $f_{43}$ increase with increasing oxidation, particularly when the decomposition window is included in analysis (i.e., TAG total chromatogram). However, TAG fractions from the leaf BBOA data are more varied and do not exhibit a clear trend. In general, the TAG fractions tend to fall to the left of AMS $f_{44}$ vs $f_{43}$ data points, indicating that the TAG excels at throughput of less-oxygenated hydrocarbon OA and struggles with throughput of oxidized species in the compound window. However, the increase in TAG $f_{44}$ with inclusion of decomposition window material shows a clearer oxidation trend that is in greater agreement with the AMS oxidation trend. This interpretation relies on the assumption that the $m/z$ 43 and $m/z$ 44 signals obtained in the TAG decomposition window from sample thermal desorption at 310°C are similar in nature to those obtained when aerosol is flash vaporized at 600°C in the AMS.

### 3.3.2 Decomposition Window PMF Analysis

To aid identification of key thermal decomposition products, the binning deconvolution PMF method was applied to the TAG chromatogram decomposition window (Figures 9 and 10). Details of the PMF analyses are provided in Supplemental Information (Figures S12 and S13). Tentative identification of different factors was facilitated by the NIST mass spectral database, though standard injections are needed to adequately quantify the decomposition window signal and identify the factors with complete confidence. As with the compound window PMF results, chromatograms are displayed as triplicate averages of binned data at each level of oxidation and are demarcated by vertical lines across the x-axis. Key ions are labeled, and tentative identifications are provided above each mass spectrum.

For the leaf BBOA chromatograms, a 4-factor solution gave several distinguishable factors (Figure 9; additional information provided in Figures S12c and S13c), including the $m/z$ 44 ($CO_2^+$) signal previously identified as originating from thermal decomposition oxidized organics (F1). Factor 3 (F3), dominated by $m/z$ 78 (possibly $C_6H_6^+$) with smaller contributions from $m/z$ 39 ($C_3H_3^+$) and $m/z$ 51 ($C_4H_3^+$), could indicate decomposing aromatics. Factor 2 (F2) matches with nitrogenated compounds in the mass spectral database, and the co-elution of $m/z$ 43 (possibly $C_2H_3O^+$) and $m/z$ 79 (possibly $C_4H_3N_2O^+$) could signal the presence of nitrogenated oxidized organics. Finally, factor 4 (F4) is dominated by multiple fragments characteristic of less-oxidized or unsaturated organic material, including $m/z$ 55 ($C_4H_7^+$), $m/z$ 67 ($C_5H_7^+$), and $m/z$ 91 ($C_7H_7^+$); this factor may also include contributions from air ($m/z$ 40; $Ar^+$) and $m/z$ 79 split from factor 3.

A 5-factor solution was chosen for the wood BBOA chromatograms (Figure 10; additional information provided in Figures S12d and S13d). Factor 1 (F1) is dominated by $m/z$ 44, attributed to decomposing oxidized organics ($CO_2^+$).

Acetic acid was identifiable in factor 2 (F2) based on relative abundances of $m/z$ 43 ($C_2H_3O^+$), $m/z$ 45 ($CHO_2^+$), and $m/z$ 60 ($C_2H_4O_2^+$), suggesting that organic acids comprise part of the thermal decomposition OA. Factor 3 (F3) features $m/z$ 50 and $m/z$ 52 (possibly $CH_3{}^{35}Cl^+$ and $CH_3{}^{37}Cl^+$, respectively) in the 3:1 isotopic ratio characteristic of chlorine, indicating that the wood BBOA may contain chlorinated organics. Based on comparison of retention times, the large contribution of $m/z$ 44 to factor 3 may be due to splitting from factor 1. Factor 4 is dominated by ions characteristic of less-oxygenated or unsaturated organic material, including $m/z$ 55 ($C_4H_7^+$), $m/z$ 72 ($C_4H_8O^+$), and $m/z$ 84 ($C_5H_8O^+$). Lastly, factor 5 (F5) has been identified as furfural using the mass spectral database, which has been previously reported in gas-phase mass spectral measurements of biomass burning emissions (Stockwell et al., 2015).

Because of the lack of chemical resolution in the thermal decomposition window, trends in factors with oxidative aging remain challenging to interpret. Notably, the factors featuring $m/z$ 44 (F1 in both Figure 9 and 10) increase with photochemical aging, consistent with an increase in oxidized OA. In the heartwood BBOA, F2 (acetic acid) and F4 (less-oxidized organics) appear to peak at 1-3 days of equivalent aging, though the mechanisms driving this change remain uncertain. The PMF results obtained in this study will be used to develop appropriate standards for the TAG thermal decomposition window, allowing for more quantitative analysis and easier identification of mass spectral fragments in future field and laboratory work.

### 3.4 $m/z$ 60 as a Tracer for both Primary and Aged BBOA

The signal eluting between minutes 27 and 32 of GC analysis results from the co-elution of multiple compounds, including levoglucosan. Many of these co-eluting species exhibit $m/z$ 60 (dominated by the $C_2H_4O^+$ ion) as a major fragment in their mass spectra. These compounds are poorly resolved because the non-polar GC column is not designed to resolve such polar compounds. SICs at different levels of oxidation reveal that each compound within this retention time window reacts at a unique rate, allowing for the identification of different co-eluting species.

Heartwood and leaf BBOA $m/z$ 60 SICs at each level of oxidation are given in Figure 15, and relative abundances of key $m/z$ 60 fragmenting species in the TAG compound window are provided in Supplemental Information (Tables S9 and S10). In the unaged heartwood BBOA chromatograms, approximately 82% of the TAG compound window $m/z$ 60 signal has been identified as levoglucosan (retention time determined from authentic standards; Figure S17 in Supplemental Information), though other sugars and anhydrosugars exist in lower abundances. Although some levoglucosan (between 8.35% and 3.20%) is present in the leaf BBOA chromatograms, up to 60% of the TAG compound $m/z$ 60 signal comes from quinic acid, which elutes beginning at minute 29 (retention time determined from authentic standards; Figure S17). The differences in sources of $m/z$ 60 between types of biomass illustrate that the $m/z$ 60 signal in any given BBOA sample may be highly complex and dependent on the type of biomass burned. Additionally, the presence of $m/z$ 60 is likely dependent on the combustion characteristics, as combustion processes can influence the emission and phase of different compounds.

In the leaf and heartwood BBOA, an increase in the $m/z$ 60 signal was observed in the decomposition window from 0 to 6-10 days of equivalent aging (Figure 12). Deconvolution PMF results demonstrate that the $m/z$ 60 decomposition signal co-elutes with $m/z$ 43 and $m/z$ 45 signals, which likely correspond to $C_2H_3O^+$ and $CHO_2^+$, respectively, and is

distinct from the mass spectrum of levoglucosan (Figure S18 in Supplemental Information). The co-elution of these three fragments and their relative integrated abundances provides evidence that organic acids constitute a portion of the decomposing OA. Further, the increase in the *m/z* 60 integrated signal suggests that these acids are formed during oxidative reactions occurring in the PAM chamber, either through heterogeneous oxidation of primary BBOA or condensation of oxidized SOA material.

Relative rates of decay for TAG integrated *m/z* 60 fragmenting species are given in Figure 12. For leaf BBOA (Figure 12a), these compounds include levoglucosan, quinic acid, mannose, and octadecanoic acid, and for heartwood BBOA (Figure 12b), these include levoglucosan, methyl-β-D-glucopyranoside, galactoheptulose, n-acetyl-d-galactosamine, and 1,6-anhydro-α-d-galactofuranose. The TAG decomposition window *m/z* 60 signal, total TAG compound window *m/z* 60 signal, and AMS $f_{60}$ (the ratio of *m/z* 60 to the total signal; Ng et al., 2011) are also included in Figure 12a and 12b for comparison. All values have been normalized to the signal obtained at 0 days of equivalent aging. The normalized abundances for TAG species were obtained by integrating each compound's *m/z* 60 signal at each level of oxidation, then dividing each peak area by the peak area obtained in the unaged chromatograms ("0 days"). As with TAG species, AMS $f_{60}$ has been normalized at each level of oxidation to the AMS $f_{60}$ obtained without photochemical aging.

Primary TAG species generally decrease in abundance with photochemical aging, though rates of decay vary depending on the compound. By contrast, in both heartwood and leaf BBOA, the TAG decomposition *m/z* 60 summed signal increases overall from zero to 6-10 days of equivalent aging, peaking at 1-3 days of aging. In the leaf BBOA, the AMS *m/z* 60 signal decreases by approximately 10% at 6-10 days of aging, while the AMS $f_{60}$ in the wood BBOA is reduced to 50% of its original value at the highest level of oxidation. These trends in AMS $f_{60}$ may reflect the combined effects of the oxidative decay of primary BBOA compounds, including sugars and anhydrosugars, and the formation of organic acids with functionalization reactions in the PAM chamber. Previous BBOA chemical characterization studies have identified organic acids as BBOA tracers (Falkovich et al., 2005; Lin et al., 2016; Mazzoleni et al., 2007), and Ortega et al. report that organic acids formed through OFR-driven oxidation may contribute to net AMS *m/z* 60 (Ortega et al., 2013).

Figure 12c displays experimental relative abundances as functions of equivalent aging time for various TAG and AMS markers observed during wood BBOA oxidation, along with levoglucosan decay rates calculated using $k_{LG}$ values obtained in previous studies (Hennigan et al., 2010; Kessler et al., 2010). In addition, AMS $f_{60}$ values obtained for PAM-aged turkey oak BBOA (*Q. laevis*) during the FLAME-3 campaign (Ortega et al., 2013) are overlaid for comparison; the values plotted correspond to $f_{60} = 0.028$ at $OH_{exp} = 0$ molec cm$^{-3}$ s and $f_{60} = 0.016$ at $OH_{exp} = 5.6 \times 10^{11}$ molec cm$^{-3}$ s (approximately 4 days of equivalent aging based on their PAM reactor calibration), with each point normalized to $f_{60} = 0.028$ (Ortega et al., 2013).

The OH-driven oxidation kinetics of levoglucosan in BBOA have been investigated in previous chamber oxidation studies. For example, Kessler et al. obtained a second order rate constant of $k_{LG} = (3.09 \pm 0.18) \times 10^{-13}$ cm$^3$ molec$^{-1}$ s$^{-1}$ from AMS measurements of OFR-oxidized levoglucosan particles (Kessler et al., 2010), while Hennigan et al.

obtained a rate constant of $k_{LG} = (1.1 \pm 0.5) \times 10^{-11}$ cm$^3$ molec$^{-1}$ s$^{-1}$ from smog chamber experiments (Hennigan et al.,

2010). Lai et al. obtained expressions for $k_{LG}$ as a function of relative humidity and temperature in their own smog chamber experiments; at 25°C and 30% relative humidity, $k_{LG} = 1.107 \times 10^{-11}$ cm$^3$ molec$^{-1}$ s$^{-1}$, a value in good agreement with Hennigan et al.'s results (Lai et al., 2014). Lai et al. attribute the discrepancy between Kessler et al.'s and Hennigan et al.'s calculated $k_{LG}$ to differences in both the levoglucosan detection method and experimental OH concentration ranges. First, while Hennigan et al. used offline filter collections to determine levoglucosan

concentrations, Kessler et al. took online measurements using an AMS and used $m/z$ 144 as the marker fragment for levoglucosan. Lai et al. suggest that because the parent ion of $m/z$ 162 was not used as the marker fragment in Kessler et al.'s AMS measurements, any potential effects from reaction products cannot be fully isolated, possibly leading to an underestimate of levoglucosan decay. However, our chromatographic methods are not subject to this mass spectral interference, and in the case of the heartwood BBOA, the TAG-measured levoglucosan decay matches the decay

predicted by Kessler et al. Additionally, Lai et al. suggest that their own results may differ from those obtained by Kessler et al. because they operated at much lower OH concentrations. During these experiments, OH concentrations ([OH]) ranged from $10^9 - 10^{10}$ molec cm$^{-3}$, closer to the operating conditions of Kessler et al. ([OH] = $10^9 - 2 \times 10^{11}$ molec cm$^{-3}$; Kessler et al., 2010) than Lai et al. ([OH] = $3.50 \times 10^7$ molec cm$^{-3}$; Lai et al., 2014).

Although levoglucosan decays rapidly in the leaf BBOA with increasing OH$_{exp}$, levoglucosan in the heartwood BBOA

is depleted more slowly. Levoglucosan is classified as semivolatile (at 25°C, $p_L° \sim 1.81 \times 10^{-7}$ torr; ACD/Labs, 2017) and is therefore expected to partition between the gas and particle phases. To approximate phase partitioning, particle-phase fractions for levoglucosan ($\xi_{LG}$) were calculated based on AMS total organic concentrations and effective saturation concentrations ($C_{LG}^*$, μg m$^{-3}$) using equation 2. The resulting values and relevant parameters are reported in Table S12. For each fuel, little variance is expected in levoglucosan particle-phase fraction between oxidation

conditions, so we conclude that phase partitioning is unlikely to be driving trends in levoglucosan abundances observed in these experiments. Based on the partitioning approximations, the leaf BBOA is expected to contain a higher percentage of levoglucosan in the particle phase than the heartwood BBOA ($91.1 \pm 1.65\%$ vs $77.8\% \pm 2.26\%$), though in both cases, gas-phase levoglucosan concentrations are likely to remain low. The prevalence of levoglucosan in the particle phase during photochemical aging is consistent with previous laboratory measurements of aged

levoglucosan particles (Kessler et al., 2010). Considering that heartwood BBOA exhibited lower total organic concentrations than the leaf BBOA, the slower depletion of levoglucosan in the heartwood samples is perhaps consistent with OH suppression effects, wherein OH experiences increased reactivity with gas-phase species at the particle surface.

The AMS $m/z$ 60 signal agrees well with the levoglucosan decay rate calculated using Kessler et al.'s $k_{LG}$, and

decreases with increasing OH$_{exp}$, though displays less overall decay compared to levoglucosan measured by the TAG. Our results demonstrate that although $m/z$ 60 may be an effective tracer for levoglucosan and primary BBOA under certain conditions, the formation of organic acids through photochemical aging may also impact AMS $m/z$ 60 and should be considered when using the AMS to track levoglucosan and primary BBOA in future studies. Furthermore, these results illustrate the utility of TAG data in interpreting AMS bulk OA measurements, as it gives both molecular

characterization as well as additional insight on the chemical makeup of the most aged OA through evaluation of thermal decomposition components.

**4 Conclusions and Atmospheric Implications**

The experimental methods presented in this work allow repeatable collection, oxidation, and molecular-level analysis of source-specific BBOA. The identification of molecular tracers unique to leaf and wood fuels can aid apportionment
of BBOA to different plant fractions. For example, based on our results, a BBOA plume exhibiting high concentrations of aliphatic leaf wax components may be attributed to canopy or leaf litter devolatilization and combustion, while a plume with high concentrations of levoglucosan and lignin decomposition products could be attributed to heartwood combustion. Additionally, our results suggest that certain molecular components present in freshly-emitted BBOA may persist after 3 days of equivalent aging and could even increase in abundance with atmospheric aging due to
reaction or gas-to-particle partitioning. The relative rates of OH-driven decay obtained from TAG measurements may thus inform future field observations where molecular speciation information is obtained for photochemically aged plumes.

The PMF deconvolution results support the identification and analysis of individual compounds present in heartwood and leaf BBOA. Because each chromatogram may contain hundreds of compounds, a general knowledge of the
680 compound classes characteristic of each BBOA type can greatly reduce individual compound analysis time and ensure that chromatograms are characterized as completely as possible. The results presented in this study therefore confirm that the chromatogram binning method coupled with PMF, as developed by Zhang et al. (Zhang et al., 2014, 2016), can aid molecular tracer analysis by elucidating different compound classes of interest present in BBOA. The compound window PMF results provide information on characteristic mass spectral signatures within leaf and wood
primary BBOA and may be compared to results obtained in future BBOA studies to more fully characterize how different compounds evolve with photochemical aging in the atmosphere.

Based on previous studies, combustion conditions are expected to significantly impact the chemical composition of both primary and secondary BBOA (Ortega et al., 2013; Reece et al., 2017; Weimer et al., 2008; see "AMS Chemical Characterization" in Supplemental Information). The resistive heating technique applied in these experiments allows
for the isolation of devolatilization (pre-combustion) and low-temperature (≤300°C) smoldering conditions, which is difficult to achieve in combustion chambers that require ignition of a flame. For example, Tian et al. designed a chamber that allows the user to control the relative contributions of smoldering and flaming combustion, though smoldering combustion is only achieved in this chamber following the introduction of a flame to the biomass fuel (Tian et al., 2015). The devolatilization and combustion procedure presented here is thus advantageous for
investigating aerosol from small masses of biomass fuel under tightly controlled conditions. However, these results alone are likely not representative of a real-world system, where smoldering combustion often occurs alongside flaming combustion. Our results may therefore serve to complement field measurements, where either smoldering or flaming combustion may dominate, as well as laboratory studies where combustion conditions are controlled.

Future work will focus on characterizing sources of bias to improve quantification of material in both the TAG compound and decomposition window. For example, particle matrix effects, whereby certain compounds exhibit enhanced or diminished recovery due to the presence of a particle matrix, have been reported to influence compound responses in previous work with the TAG and other thermal desorption GC systems, particularly for large molecular weight compounds (Lambe et al., 2009; Lavrich and Hays, 2007). Lambe et al. quantified this effect for the TAG by co-injecting a constant $C_{30}$ deuterated alkane standard with 0-60 µg motor oil and found that the presence of the motor oil matrix enhanced recovery of the standard by a factor of 2-3 (Lambe et al., 2009). In these experiments, the TAG collected estimated ranges of 6-16 µg particles for leaf BBOA and 22-36 µg particles for heartwood BBOA. Based on these mass ranges, we do not expect these matrix effects to contribute significantly to our results, especially for the lower molecular weight compounds. However, future work will incorporate an evaluation of matrix effects to minimize bias in TAG measurements. Although the TAG's OA analysis capability has historically been limited by poor mass throughput of highly oxygenated species, we demonstrate here that the TAG decomposition window can be used to gain a better understanding of the molecular composition of oxidized BBOA. Though the decomposition window does not provide chemical composition information with molecular resolution, the chromatogram binning PMF results allow identification of different co-eluting factors, many of which correspond to molecular fragments that could be used as source-specific BBOA tracers in future field studies.

The utility of the thermal decomposition window is limited by a lack of adequate analytical standards, particularly for organic components. Although ammonium sulfate and ammonium nitrate standards have been used to quantify sulfate and nitrate particles in previous work (Williams et al., 2016), the development of satisfactory standards for decomposing organics remains difficult for several reasons. Fragments eluting in the decomposition window may be tentatively identified using available mass spectral identification tools, though we often cannot infer the source of the fragments, since they are products of compound thermal decomposition rather than volatilization. Many of the compounds undergoing decomposition during sample desorption may therefore be too involatile for typical GC-MS analysis. Despite these challenges, analytical standards are currently under development to aid identification and interpretation of decomposition window results based on molecular functionality. For both types of BBOA, the *m/z* 44 signal in the TAG decomposition window increases with photochemical aging, confirming that this signal indicates the presence of thermally labile oxygenated OA. The increase in *m/z* 44 with oxidation in both the TAG decomposition window and the AMS mass spectra is consistent with results from previous studies (Williams et al., 2016). However, our observations suggest that the utility of decomposition *m/z* 44 as a quantifiable tracer for aged OA varies depending on OA type. For the heartwood BBOA, the TAG decomposition *m/z* 44 signal correlates well with AMS $\overline{OS_C}$, suggesting that for this type of BBOA, the decomposition *m/z* 44 abundance could be used to estimate the aerosol's oxidation state. By contrast, the correlation between TAG decomposition *m/z* 44 and AMS $\overline{OS_C}$ is not significant for PAM-aged oak leaf BBOA, perhaps because compounds formed with photochemical aging of leaf BBOA are less thermally labile and more resistant to thermal decomposition than those found in aged heartwood BBOA. In addition, without mass-based standard calibrations for the decomposition window, distinguishing between an increase in thermally labile mass (i.e. due to SOA formation) and a relative increase in thermally decomposing OA due to changes in chemical composition (i.e. due to heterogeneous oxidation and functionalization) remains challenging.

From the TAG data, we observe two competing effects driving the overall *m/z* 60 signal measured in the AMS. While many primary BBOA components exhibiting a characteristic *m/z* 60 fragment, including anhydrosugars like levoglucosan, were depleted with photochemical aging, an enhanced *m/z* 60 signal in the decomposition window indicates increased formation of organic acids in the PAM reactor. Both processes have been reported in previous

literature, though the oxidative depletion of primary BBOA is most typically thought to drive AMS *m/z* 60 trends in field and laboratory studies. Our data suggest that although AMS measurements provide useful chemical composition information on bulk OA, laboratory studies with molecular-level measurements are needed to complement AMS data and provide a more complete understanding of processes occurring in the atmosphere.

The mechanisms driving compositional changes in BBOA remain challenging to interpret. Although many compounds

observed in this study are clearly depleted through functionalization reactions, some species may be subjected to phase partitioning effects in addition to PAM-driven oxidation. In particular, the enhancement in TAG thermal decomposition *m/z* 44 and *m/z* 60 may occur due to formation of SOA through oxidation and condensation of low-volatility gases, heterogeneous functionalization of compounds in the particle phase, or a combination of these processes. Future studies will focus on investigating the role of phase partitioning in OA chemical composition within

BBOA plumes, with emphasis on the thermally labile material eluting in the TAG thermal decomposition window. In addition, different types of biomass will be tested to explore the dependence of phase partitioning and photochemical aging effects on fuel type, broadening the applicability of these techniques to future field measurements.

**Data Availability**

Data from this study are available upon request by contacting the corresponding author.

**Acknowledgements**

The material presented is based on work supported by the National Science Foundation (award no. 1437933). The authors would also like to acknowledge support from the International Center for Energy, Environment and Sustainability (INCEES) and the McDonnell Academy Global Energy and Environment Partnership (MAGEEP) at Washington University in St. Louis. The authors would also like to thank Audrey Dang, Benjamin Sumlin, and

760 Junseok Lee for assisting with supplementary CO measurements. Finally, the authors would like to thank Benjamin Sumlin for his insight during the editing process.

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

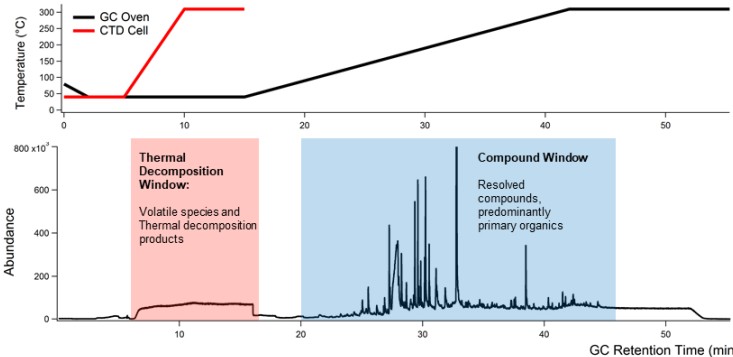

**Figure 1:** An example TAG chromatogram with GC oven and TAG collection and thermal desorption (CTD) cell temperature ramp programs.

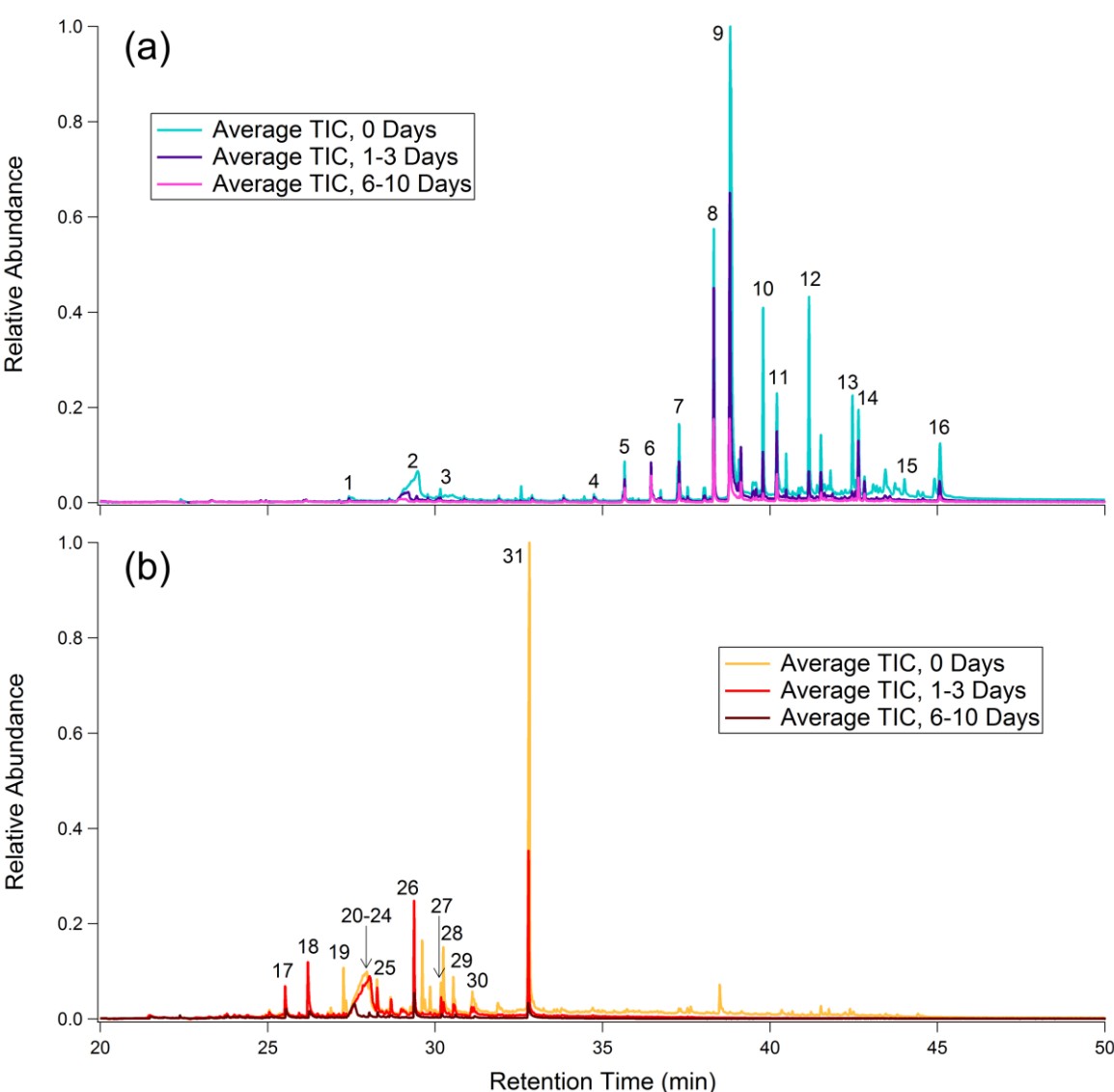

**Figure 2:** Chromatograms for **(a)** leaf BBOA and **(b)** heartwood BBOA at different levels of oxidation. Corresponding names and structures for numbered compounds are given in Table S5 and Figures S9.  For each plot, all traces are normalized to the point of highest abundance within the average unaged chromatogram.

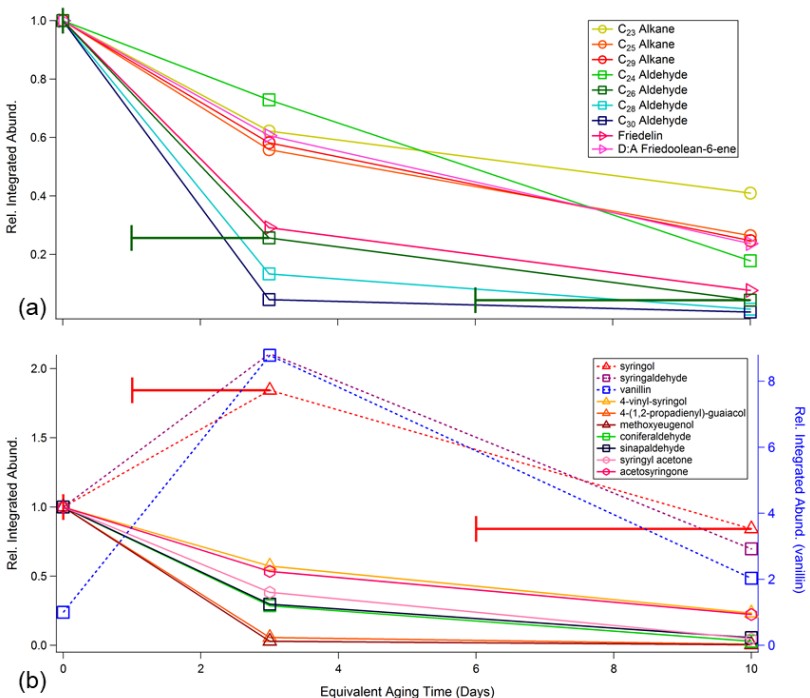

**Figure 3:** Relative changes in integrated abundance as a function of equivalent aging time (per $SO_2$ calibrations) for primary compounds identified in **(a)** oak leaf BBOA chromatograms, and **(b)** oak heartwood BBOA chromatograms. For each compound, the integrated abundances were first normalized to appropriate volume concentrations, then subsequently normalized to corresponding abundances at no oxidation ("0 days"). Compounds that decrease in abundance are indicated with solid lines, and compounds that deviate from this trend are given with dotted lines. Raw compound abundances are provided in Supplemental Information (Table S6). X-axis error bars denote equivalent aging time ranges calculated for this study and are applicable to all TAG data presented here, though they are only included on one compound per panel to preserve figure readability.

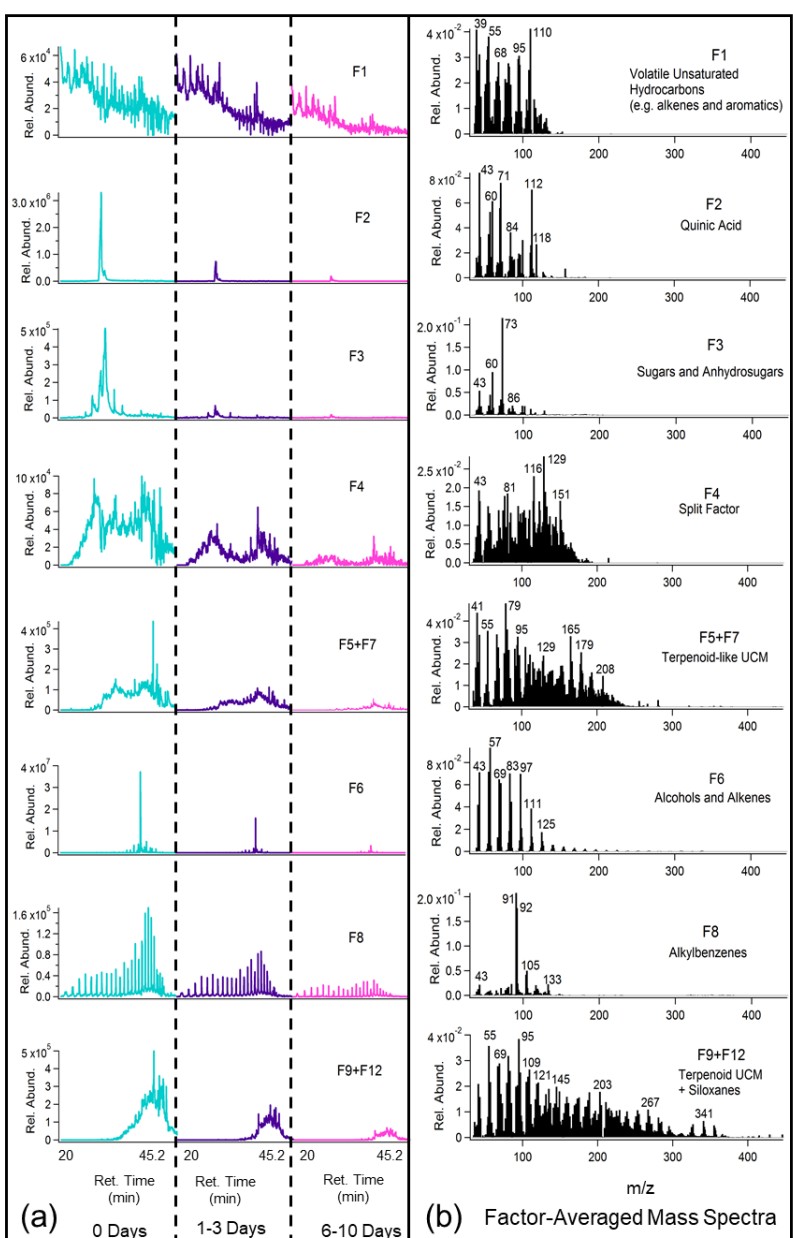

**1015**

**Figure 4:** Average binned chromatograms and mass spectra for factors 1-9+12 (F1-9+12) in PMF 15-factor solution on TAG oak leaf BBOA compound window data. Relevant plots obtained in PMF calculations are provided in Supplemental Information (Figures S12a and S13a). These chromatograms were obtained from PMF calculations by averaging binned data corresponding to triplicate chromatograms at each level of oxidation. The triplicate-averaged

**1020**   binned chromatograms at each equivalent aging time are displayed in one trace; different aging times are demarcated with vertical lines across the x-axis.

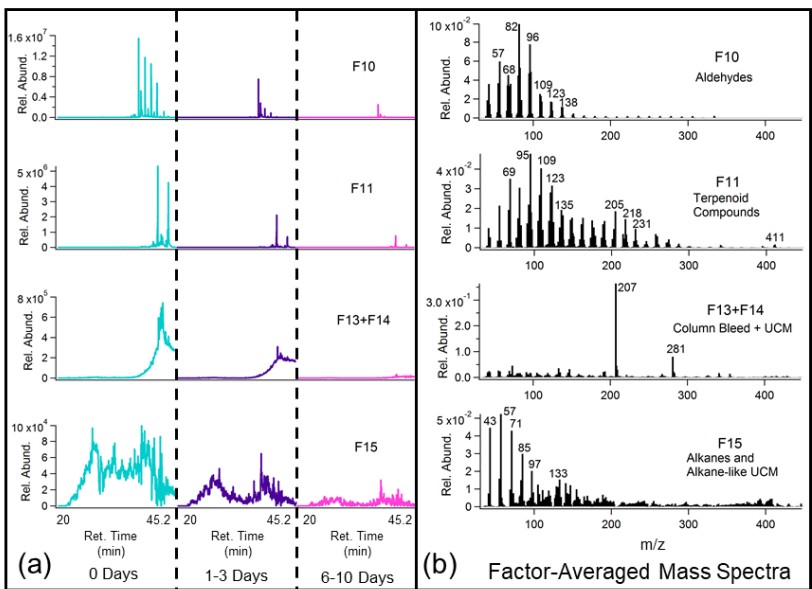

**Figure 4, cont'd:** Average binned chromatograms and mass spectra for factors 10-15 (F10-15) in PMF 15-factor

solution on TAG oak leaf BBOA compound window data.

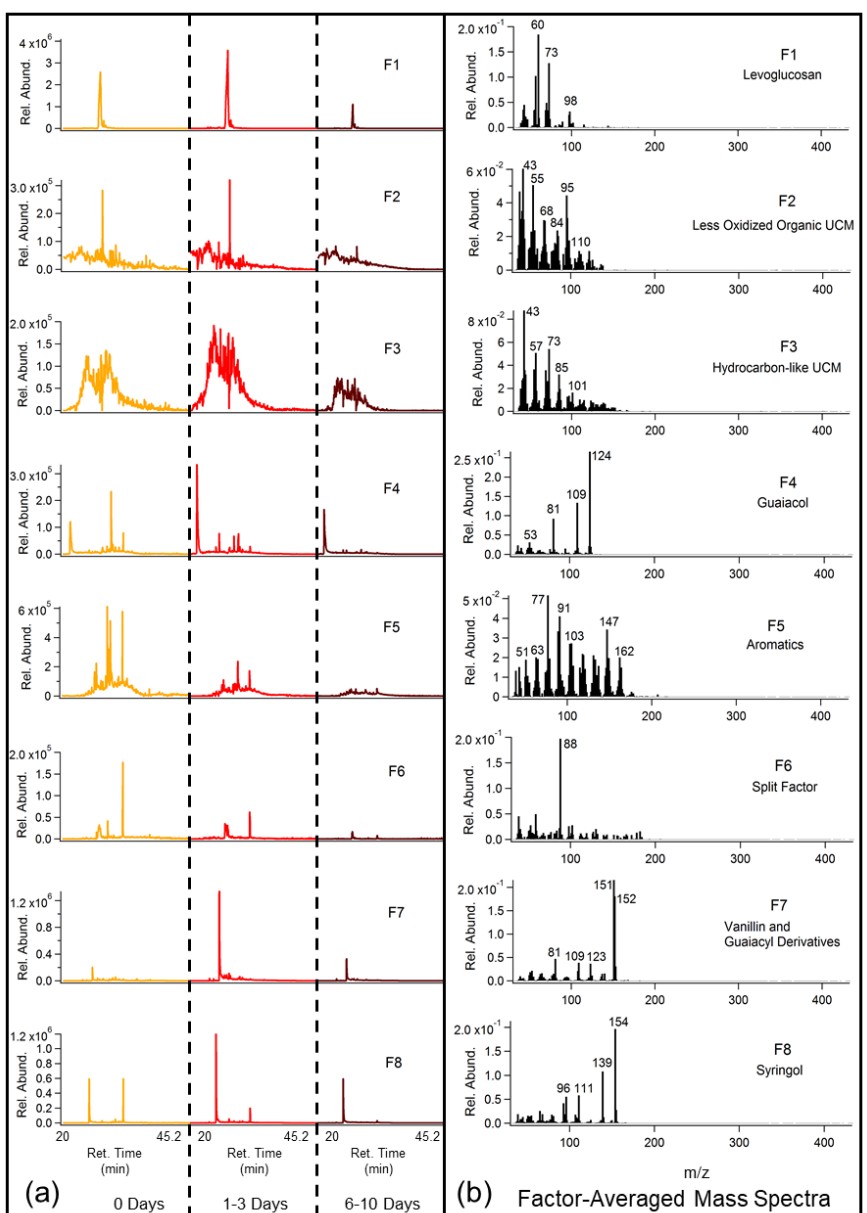

**Figure 5:** Average binned chromatograms and mass spectra for factors 1-8 (F1-8) in PMF 18-factor solution on TAG oak heartwood BBOA compound window data. Relevant plots obtained in PMF calculations are provided in Supplemental Information (Figures S12b and S13b). These chromatograms were obtained from PMF calculations by averaging binned data corresponding to triplicate chromatograms at each level of oxidation. The triplicate-averaged

binned chromatograms at each equivalent aging time are displayed in one trace; different aging times are demarcated with vertical lines across the x-axis.

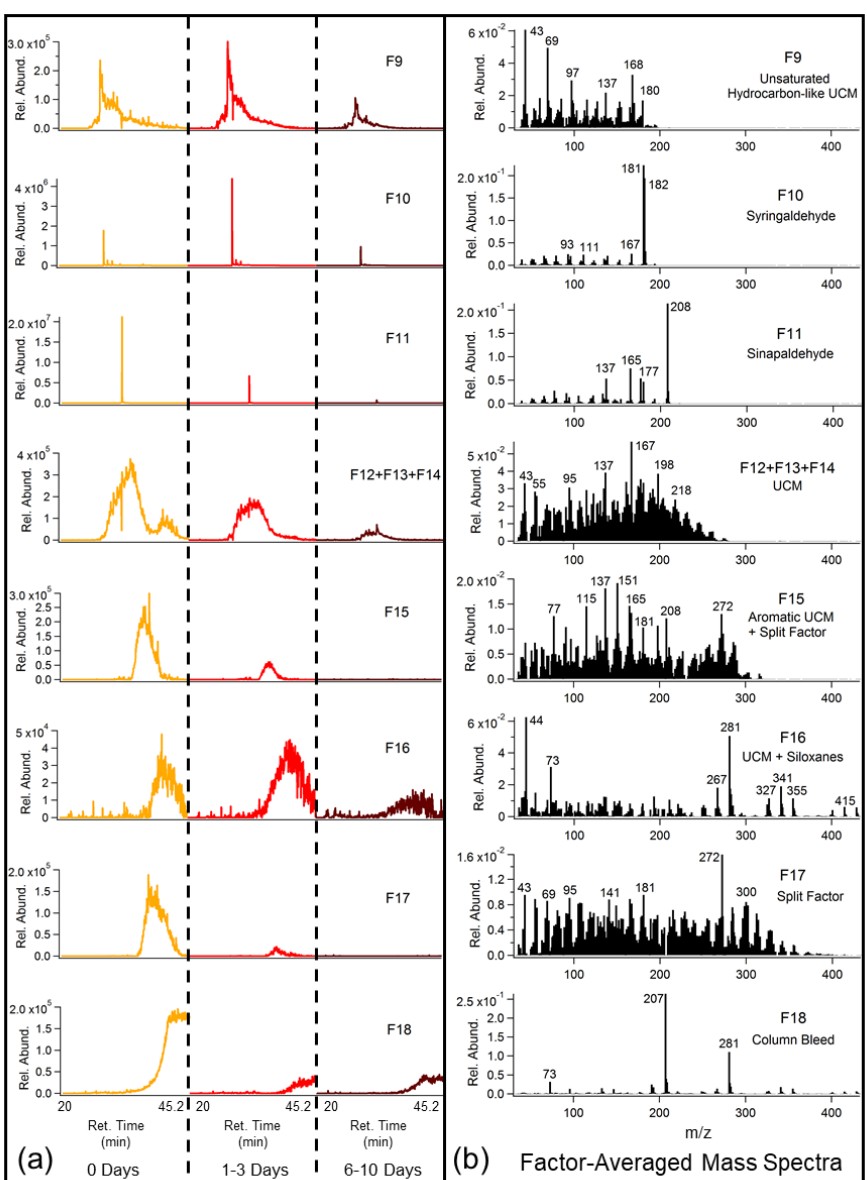

**Figure 5, cont'd:** Average binned chromatograms and mass spectra for factors 9-18 (F9-18) in PMF 18-factor solution on TAG oak heartwood BBOA compound window data.

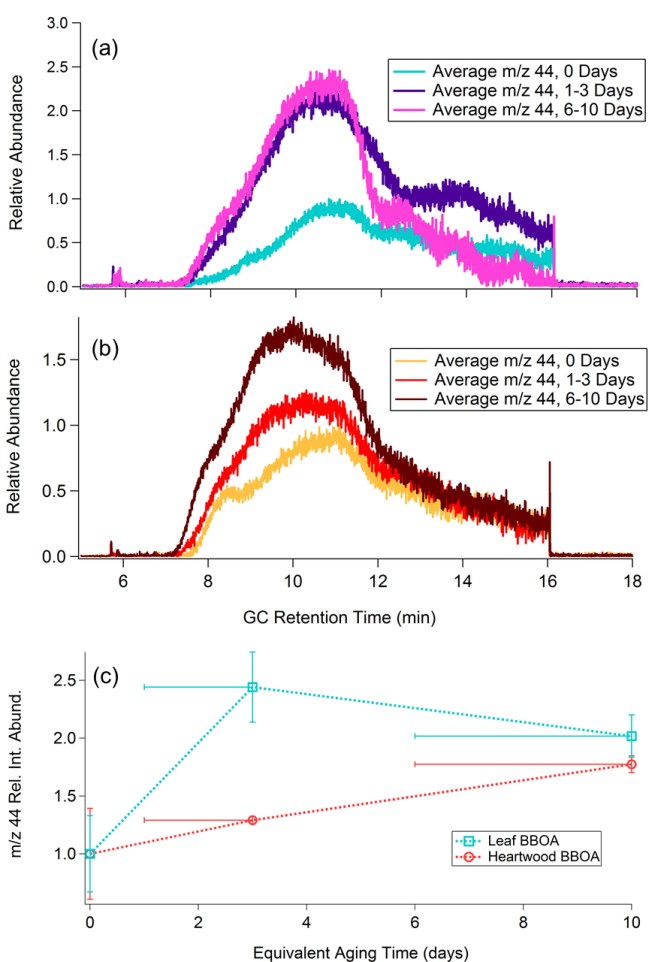

**Figure 6: (a)** Average *m/z* 44 single ion chromatograms (SICs) across distinct levels of photochemical aging for leaf BBOA, normalized to the point of highest abundance within the averaged unaged chromatogram ("0 days"). **(b)** Average *m/z* 44 single ion chromatograms (SICs) across different levels of photochemical aging for heartwood BBOA, normalized to the point of highest abundance within the averaged unaged chromatogram. **(c)** summed relative *m/z* 44

decomposition signal as a function of photochemical aging for both fuels (± one standard deviation). These values were obtained by averaging triplicate *m/z* 44 decomposition signals at each level of photochemical aging. For each fuel type, all summed abundances are normalized to the unaged *m/z* 44 signal ("0 days"). The x-axis error bars denote the equivalent aging time range and are applicable for all measurements obtained in this study.

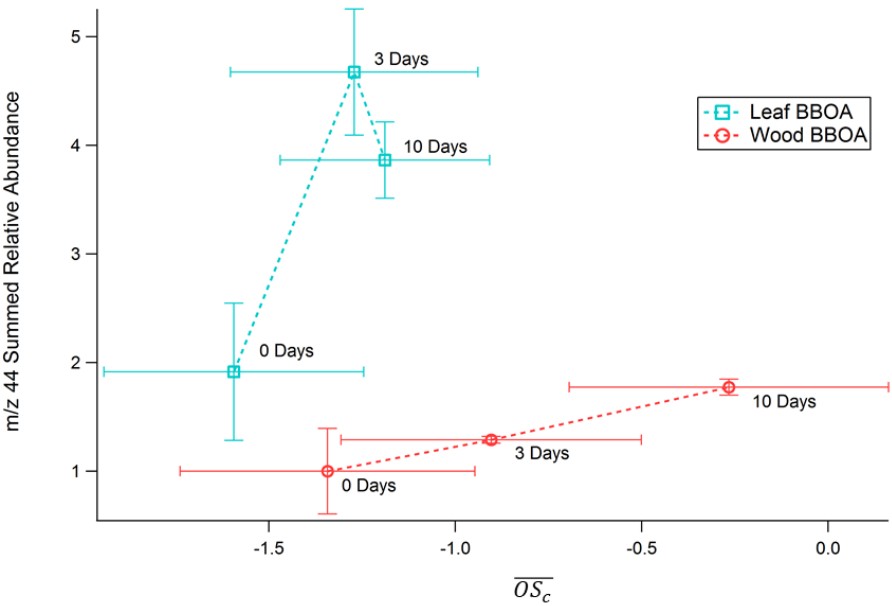

**Figure 7:** TAG decomposition *m/z* 44 integrated relative abundances for PAM-aged leaf and heartwood BBOA as functions of AMS $\overline{OS_c}$. Here, all TAG data have been normalized to the unaged ("0 days") wood BBOA integrated *m/z* 44 abundance.

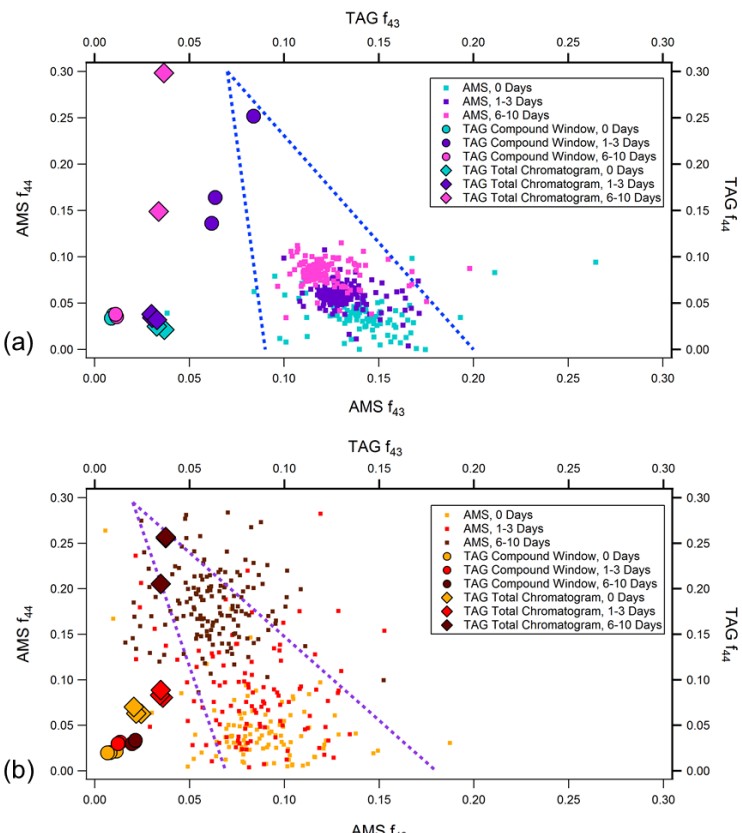

**Figure 8:** AMS and TAG $f_{44}$ vs $f_{43}$ at different levels of photochemical aging for **(a)** leaf and **(b)** heartwood BBOA. TAG $f_{44}$ and $f_{43}$ values were obtained using Eq. (3). To minimize noise, AMS data is plotted only for points where sufficient total organic concentrations were achieved, around the peak of the concentration profile. The triangles formed by the blue dotted lines provide visual guidelines for the evolution of OA chemical composition across $f_{44}$ vs $f_{43}$ space; the apex of the triangle indicates the direction of OA photochemical oxidation (Ng et al., 2010).

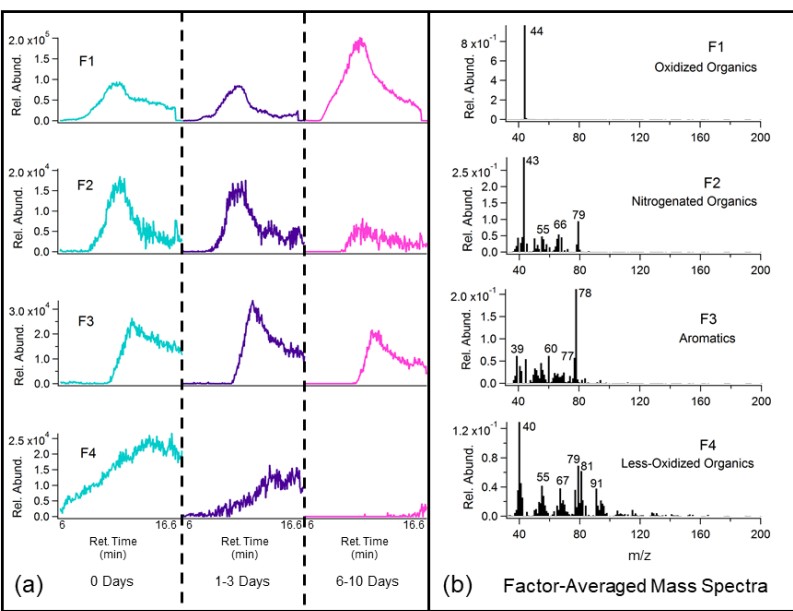

**Figure 9:** Average binned chromatograms and mass spectra for factors 1-4 (F1-4) in PMF 4-factor solution on TAG oak leaf BBOA decomposition window data. Relevant plots obtained in PMF calculations are provided in Supplemental Information (Figures S12c and S13c). These chromatograms were obtained from PMF calculations by averaging binned data corresponding to triplicate chromatograms at each level of oxidation. The triplicate-averaged binned chromatograms at each equivalent aging time are displayed in one trace; different aging times are demarcated with vertical lines across the x-axis.

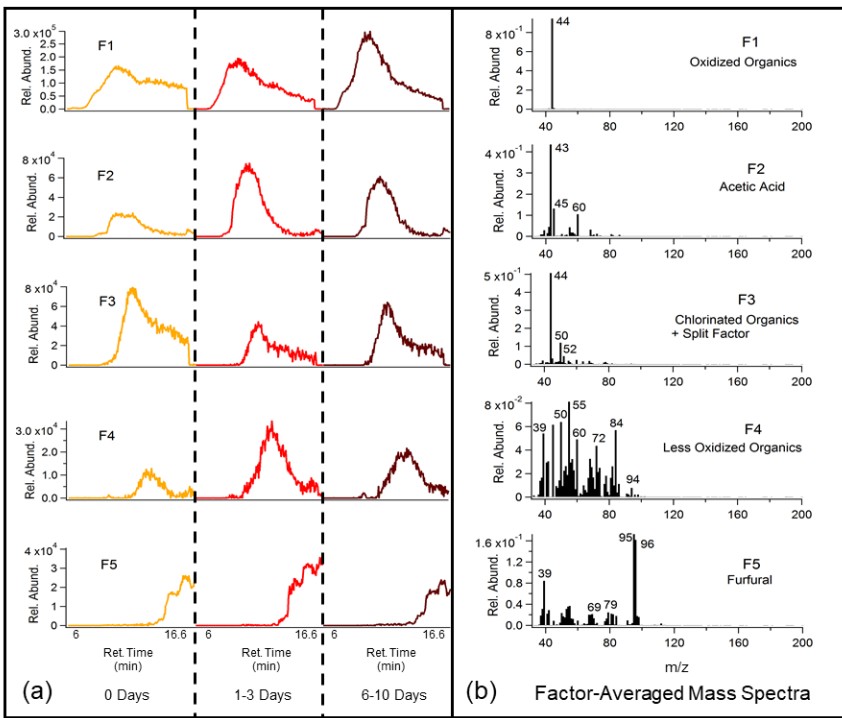

**Figure 10.** Average binned chromatograms and mass spectra for factors 1-5 (F1-5) in PMF 5-factor solution on TAG heartwood BBOA decomposition window data. Relevant plots obtained in PMF calculations are provided in Supplemental Information (Figures S12d and S13d). These chromatograms were obtained from PMF calculations by averaging binned data corresponding to triplicate chromatograms at each level of oxidation. The triplicate-averaged binned chromatograms at each equivalent aging time are displayed in one trace; different aging times are demarcated with vertical lines along the x-axis.

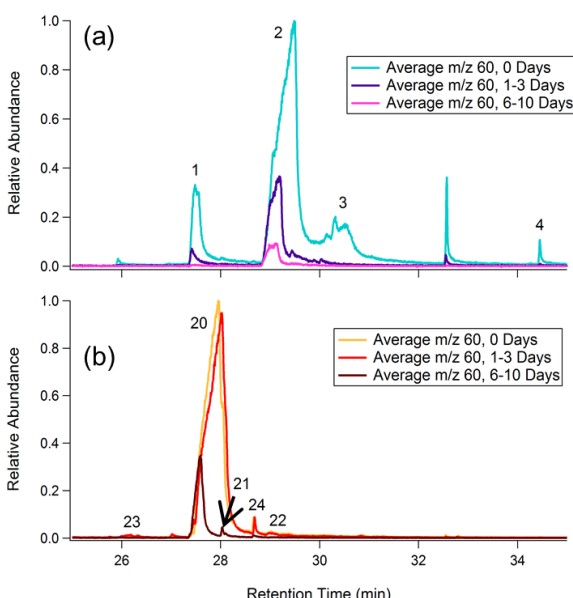

**Figure 11:** Average *m/z* 60 single ion chromatograms (SICs) across the compound window for **(a)** leaf BBOA; **(b)** heartwood BBOA. For each plot, all traces are normalized to the point of highest abundance within the average unaged chromatogram. Individual compounds are labeled according to identifications provided in Supplemental Information (Figures S9; Table S5).

1095

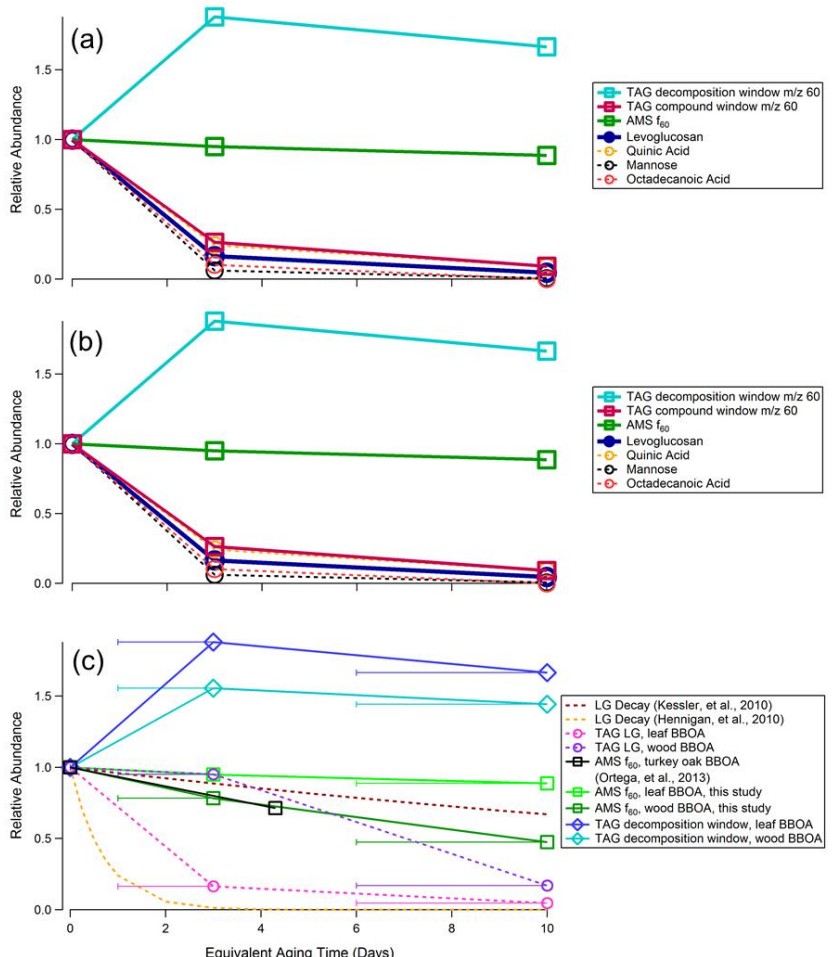

**Figure 12.** Relative changes in abundance for different *m/z* 60 fragmenting species in **(a)** leaf and **(b)** heartwood BBOA; **(c)** TAG and AMS *m/z* 60 species as a function of OH_exp. Levoglucosan (LG) decay rates were calculated using two different literature $k_{LG}$ values (Hennigan et al., 2010; Kessler et al., 2010) with an assumed typical outdoor OH concentration of $1.5 \times 10^{-6}$ molec cm$^{-3}$ (Mao et al., 2009). Additionally, normalized AMS $f_{60}$ values for turkey oak (*Quercus laevis*) BBOA obtained during the FLAME-3 campaign were adapted from Figure 10b in Ortega et al. (Ortega et al., 2013) and are included for comparison. The x-axis error bars denote the equivalent aging time range and are applicable for all measurements obtained in this study, though they are only included in panel (c) to preserve figure readability.

**Table 1.** Qualitative levels of PAM-reactor oxidation with corresponding OH exposure ($OH_{exp}$) estimations and equivalent aging times. The $OH_{exp}$ estimations were made using methods described in Supplemental Information (Methods: PAM Calibrations and Equivalent Aging Estimations).

| Qualitative Level of Oxidation | $OH_{exp}$ (molec cm$^{-3}$ s) | Equivalent Aging Time (days) |
|---|---|---|
| low-mid | $1.7\times10^{11}$-$4.4\times10^{11}$ | 1-3 |
| high | $7.7\times10^{11}$-$1.3\times10^{12}$ | 6-10 |