# Peer review of "Bulk and Molecular-Level Characterization of Laboratory-Aged Biomass Burning Organic Aerosol from Oak Leaf and Heartwood Fuels"

_Atmospheric Chemistry and Physics, 2017_

## Referee Comment (RC1) · Anonymous Referee #1 · 3 Aug 2017

Fortenberry et al. present a study that examines links between chemically speciated fresh and aged BBOA data to more bulk measurements of the AMS. They use PMF technique to pull out mass spectral trends in the speciated data and looked mass spectra from compounds that eluted during the thermal decomposition window of the TAG. Their results show 60 m/z, a traditionally used ion for the AMS to track biomass burning, depends on fuel type and aging of the aerosol particles. They also suggest that 44 m/z ion could be used as an estimate for aerosol particle's aging state. This manuscript is written clearly and contributes to the understanding of both the complexity of BBOA and interpreting results from the AMS. I recommend this paper be published in ACP with some minor revisions.

[Figure]

Minor Comments:

Line 38: parallel structure, change to "and impacts" or something along those lines

Line 39: awkward sentence starting with "Organic aerosol. . ." In addition, aerosol refers to both the particle and gas phase. When the authors mean particle phase, please change aerosol to aerosol particle.

Line 164: Traditionally, the radical dot of OH is left off, though the authors are technically correct. The dot just looks a bit strange when used, for example, in line 580.

Line 166: How well-mixed in the PAM reactor? Could large concentration gradients in aerosol particles affect the observed results?

Line 180: add fuels after leaf

Line 210: What is the rate at which the cells are heated to 310ËŽC? There is some discussion that heating rate will affect which compounds desorb vs. thermal decompose. The authors do illustrate the ramping time in Figure 1, but it would be helpful to have it written down in the text.

Line 222: combustion chamber were clean

Line 469: have specific standards been observed to decompose at these ions in this thermal decomposition window?

Line 534 (though may happen earlier): No comma after Kessler. This comma is not needed as Kessler and others (in English) requires no comma even when used in line. Please remove comma in previous+subsequent usages.

Figure 2 (a): The three types of green are very difficult to distinguish. I understand the authors were aiming for green=leaf and warm colors for heartwood, but greens all look the same. Maybe cool colors for leaf and warm colors for heartwood?

Figure 3 (b): Why are syringol, syringaldehyde, and vanillin dotted lines? Is it because

[Figure]

they are observed to increase with aging time? If so, please mention that in the caption.

Figure 8 (a): The greens are hard to distinguish.

Figure 10 (a): The green color gradient is not great and implies there was a near-continuous gradient of samples collected for that range of aging time. There were only three sampled times, so a gradient seems a bit misleading. Also, the colored right-hand and top axes are a bit confusing because the reader is trying to match the green points to the green axis instead of TAG to green. Also, the caption says the dotted lines are guidelines for where the points tend to be concentrated. This doesn't seem to be the case for the green AMS points. The blue dotted line for these points would look more like a rectangle covering the bottom half of the graph then a downward pointing cone. Is there greater meaning behind this cone?

Figure 15 (A): same comment about the green.

Even more minor comments for the SI:

Figure S4: the TAG collection 1 blue shaded region does not look to start at 30 minutes after start of heat pulse as the caption indicates.

Table S5-6: The raw SIC integration numbers have too many significant digits (and is difficult to read). Maybe consider limiting it to 2-3 with scientific notation.

---

## Referee Comment (RC2) · Anonymous Referee #2 · 12 Aug 2017

This paper describes experiments and analysis examining the composition of fresh and aged aerosol emissions from volatilization/combustion of oak wood and leaf matter using powerful and novel analytical methods. The paper describes repeated experiments heating the biomass in a combustion chamber and sampling it via a Potential Aerosol Mass (PAM) reactor into a Thermal desorption Aerosol Gas Chromatograph – Mass Spectrometer (TAG), High-Res AMS and a SMPS. Emissions were sampled unaged and under two different PAM aging conditions. TAG data were analyzed in a number of ways, including for specific eluted compounds, positive matrix factorization of chromatograms, and analysis of chemical fragments from thermal decomposition of particles. These analyses are compared to more 'standard' analyses of bulk aerosol

composition from the AMS. The suite of different approaches taken to analyze these data leads to a number of interesting and potentially important conclusions that will be of interest to the broad community interested in emissions from biomass burning and how they evolve in the atmosphere. For example, the emission of aliphatic aldehydes from leaf coating volatilization is nicely supported. The emission and evolution of components contributing mass at m/z60 in both TAG and AMS spectra receives special attention and evidence for contribution from components formed during oxidation to mass at m/z 60 is given. It does an especially nice job of spanning levels of chemical detail from compound-specific determination using the TAG, to PMF of tag chromatograms to point to compound classes, to the AMS measurements of bulk fragments.

Overall, this is a very nicely written and clear paper that makes an important contribution to understanding of a complex and important source of atmospheric aerosol. Therefore, I find it to be suitable for publication in ACP once my concerns are addressed. I have identified a number of points that, when addressed, will help the paper better fit into the existing literature on the topic.

Major points: While the analytical methods applied here are unique and provide strong insights, I have a concern about how these results can be compared to other 'forms' of biomass combustion, and so I think that more effort should be made to qualify/compare the types of emissions that were sampled. The emphasis in emission generation was clearly on repeatability and consistency, rather than on representativeness, which makes sense for these experiments. However, it would be helpful to put the OA studied here a bit more clearly in the context of 'biomass burning OA (BBOA)', which typically refers to ambient observations of biomass burning emissions emitted from a range of different fuels/types/combustion conditions. In this case, the nature of the 'combustion' that was the source of the sampled aerosols is someone unclear to me. Very small portions of biomass (0.2-0.5 g) were 'combusted' in the chamber, but it is not clear to me if flame was involved, or strictly smoldering, and so how the results might be compared to what might come from a fire. For example, flame typically produces

black carbon, was any generated here? The experiments are called 'devolatilization and combustion', but is there any way to classify this combustion more broadly or put it in the context of biomass burning more generally? If not, can anything be said about the representativeness of the emissions from this setup relative to other studies? The relative change in f60 for the study of Ortega et al. is shown in Fig. 16, but the chemical character of the OA is not compared to that measured in that or other studies, even of the same type of fuel. Several studies have shown that combustion phase/type can have a substantial effect on OA emission properties (Weimer et al. 2008; Reece et al. 2017) and as you noted, observations of SOA production in lab and field studies have been found to be highly variable and distinct. Therefore, to the extent that you can include information about your combustion and the basic characteristics of your emissions, it will enable comparison with existing measurements and analysis.

In a similar vein, one of the motivations discussed for the use of the PAM was to understand SOA production, but this is never discussed in the paper, though some evidence is presented in Table S2 that there is SOA production for the wood experiments but not for leaf experiments. These outcomes are of interest in the context of the variability in SOA production discussed above, but also because they may influence interpretation of the 'relative to unaged' presentation of data that is used in a number of figures (e.g. Fig. 3, 16). For example, are changes in fragments/compounds due to 'dilution' of primary OA by SOA, or strictly due to gas-phase or heterogeneous oxidation? This is mentioned in the paper's final paragraph, but it seems at least some further evidence/data could be presented.

The authors rightly point out that the operation of the PAM during experiments was not fully constrained by the SO2 calibration of integrated OH exposure, but then in the paper use quite tightly constrained values (3.4 and 9.8 days) of equivalent oxidation to describe the aging under the two operation conditions. The fact that there are repeated experiments and repeatable results is great (and difficult to do for biomass burning) and suggests that aging within an experiment tyep should be consistent. However,

your 'sensitivity' analysis (Table S1) shows that actual OH exposure estimates for your experiments may vary by a factor of 5 to 10 given the assumed range of external OH reactivities. Therefore, it seems strange to specify your aging conditions to such a precise degree. I would feel more comfortable if a range of days were reported or if you can find a way to estimate OH reactivity during your experiments (e.g. using published VOC profiles and a tracer ratio?) to better constrain this. At the very least, uncertainty in this value should be clearly stated when it is discussed (e.g. in the context of Table 1), so that the values given are not over-interpreted.

Specific points

Page 4; Line 138 - Initially it was unclear to me whether this heading referred to separate experiments or one type. As noted above, more of an effort should be made to describe/qualify the approach taken and how the resultant emissions compare to what might come from a fire. In addition, it would make sense to be clear and consistent when using 'BBOA' in the context of your experiments.

Page 5; Line 165 - Was level of external O3 injection always the same?

Page 6; Line 194-195 - As noted above, this uncertainty should be reflected in estimated atmospheric ages used throughout paper.

Page 7; Line 234-235 - Also related to combustion emission properties. Why are SMPS volumes used and not AMS OA concentrations? For example, if there is a contribution from BC, this will both effect the determination of OA mass by adding to volume, and also potentially affecting DMA sizing. This may not be an issue, but could at least compare AMS OA to SMPS volume?

Page 9; Line 298 - Need to be clear that this is referring to relative abundance - important if SOA production is 'diluting' primary species.

Page 9; Line 296 - I noted this included in Supplemental tables, but it might be helpful to translate to effective saturation concentration.

Page 9; Line 299 - Where possible (e.g. Fig 3),would be best to include error bars to show inter-test variability. You have done this in some places, but would be good to see it here.

Page 10; Line 323-324 - These don't seem to be fully depleted -seems to be 50-100% of relative abundance at the start?

Page 12; Line 412-413 - A useful comparison to quantify inter-test variability might be to do this calculation on repeated experiments at same loading. E.g., what are dot products between repeated tests at same conditions that are averaged together for other analyses?

Page 13; Line 425 - Isn't really clear if this is indicating an increase in the presence of material containing mz44 that can thermally decompose or an increase in thermal decomposition?

Page 13; Line 431 - As noted above, to be most useful, this should be placed in the context of other BBOA measured by AMS. How do these numbers compare to those measured in other studies - e.g. Ortega et al, 2013, Reece et al, 2017

Page 13; Line 433-436 - Significant figures not justified (or, really, linear regression advised) for 3 data points unless there is a very strong argument for there being a linear relationship

Page 13; Line 439 - It seems as or more plausible that fragmentation leads to move volatile species that aren't captured by the TAG?

Page 15; Line 504-505 - This is a good point, but here the distinction may be as much type of emission/combustion as type of biomass, as it appears that at least some OA is from volatilized leaf coating so is not 'burned' (for leaf e.g. Fig. S7)

Page 15; Line 511-513 - If possible, it would be helpful to quantify (even approximately) the relative amount of material contributing m/z 60 in the compound window vs decomposition window. I take it there is more in the former? The AMS will presumably see a

weighted average of the two?

Page 17; Line 563 - OH suppression will likely be dominated by what is in the gas phase and total OHR may be very different for two types (see above comment about uncertainty in actual OH exposure.

Page 18; Line 608-609 - No mention of relative enhancement of OA and so how much condensation versus oxidation drives changes in relative contribution from different components.

Minor points

Page 7; Line 219 - data were, not data was

Page 14; Line 461 - I think I know what 'triplicate averages' is meant to say, but can be said more clearly.

Page 15; Line 508 - I think 'distinct' would work better than 'unique'.

Page 15; Line 506-507 - Would be good to point to Fig. 16 here.

Page 17; Line 578-579 - Not sure if a species can be called a 'tracer' (for a primary source) if it is increasing w/ atmospheric processing. At the very least, it's not a tracer of a unique source.

References

Reece, S. M., Sinha, A., and Grieshop, A. P. (2017). "Primary and photochemically aged aerosol emissions from biomass cookstoves: chemical and physical characterization." Environmental Science & Technology. DOI: 10.1021/acs.est.7b01881

Weimer, S., Alfarra, M. R., Schreiber, D., Mohr, M., Prévôt, A. S. H., and Baltensperger, U. (2008). "Organic aerosol mass spectral signatures from wood-burning emissions: Influence of burning conditions and wood type." Journal of Geophysical Research: Atmospheres, 113(D10), D10304.

---

## Referee Comment (RC3) · Anonymous Referee #3 · 21 Aug 2017

Fortenberry et al. present chemical composition measurements of photochemically aged laboratory biomass burning organic aerosols (BBOA). BBOA was generated from the combustion of oak leaves and oak wood samples in a burn chamber, then exposed to OH radicals in a Potential Aerosol Mass oxidation flow reactor. Ensemble aerosol mass spectra were obtained with an AMS, and GC-MS samples were obtained with a TAG. The authors used factor analysis to identify characteristic groups of GC effluent signals that behaved differently as a function of OH exposure. In my opinion, the manuscript presents an interesting experiment and application of the measurement techniques that were used to mimic aging of BBOA surrogates. Publication in ACP may be appropriate after consideration of my comments below.

**General/Major Comments**

1. Given the goal of using TAG measurements to interpret ensemble/bulk techniques such as the AMS, and given the large number of oxygenated/polar compounds present in BBOA (and oxidized BBOA), it wasn't clear to me why the authors chose not to incorporate the online derivatization technique used in previous TAG measurements (Isaacman et al., 2014), which reports "complete derivatization of […] alkanoic acids, polyols, diacids, sugars, and multifunctional compounds." In principle, derivatization should offer the following advantages:

    a. improved recovery of methoxyphenols, levoglucosan, and other sugars and primary species that are measured in this work, along with potentially less significant matrix effects (see Comment #2).

    b. recovery of oxidation products formed following OH exposure in the PAM reactor (e.g. dicarboxylic acids) that were not resolved here.

    c. evaluation/supplementation of the thermal decomposition window because the TAG recovery and resolution of highly polar compounds is still low, as implied by discussion in L441-L454 (see Comment #16) and L493-L494.

    The authors should explain in the manuscript why they chose not to incorporate/adapt the TAG derivatization technique published by Isaacman et al.

2. The TAG recovery of the selected tracers is potentially influenced by BBOA matrix effects, which could be either positive or negative in magnitude. Using a different TD-GC/MS system, Lavrich and Hays et al. (2007) showed that the thermal extraction of large PAHs from a soot matrix was hindered. Using a TAG system, Lambe et al. (2010) showed that the recovery of a $C_{30}D_{62}$ alkane internal standard in a lubricating oil matrix increased by a factor of 2-3 as a function of matrix loading. Matrix effects may be even more significant for the polar analytes measured in BBOA (e.g. methoxyphenols and sugars). Without application of representative internal standards for at least a subset of experiments, in my opinion the authors cannot unambiguously rule out the contribution of matrix effects. For example, in Fig. 3b, the authors show an increase in the abundance of vanillin, syringol, and syringaldehyde when the OH exposure in the PAM reactor is increased to

3.4 days. In the manuscript, a plausible formation mechanism for vanillin was provided (Figure S8). The increase in vanillin, other methoxyphenols, and other tracers (including the integrated m/z = 44 SIC) that display similar behavior could also be due to higher concentrations of desorbed primary or secondary organic aerosols that adsorbed onto active sites in the TAG sample transfer path, e.g. the effect observed in Lambe et al. (2010). At the least, the discussion should be revised to acknowledge that the above scenarios can plausibly explain the observed trends regarding increase and decrease in abundance as a function of OH exposure. A more convincing response – which may prove the above hypotheses incorrect – would be to repeat one or two of the combustion experiments, while manually spiking each collected TAG sample with an appropriate set of isotopically labeled standards. For example, there are sugars that are readily available with a range of levels of deuterium-substitution, and I also found suppliers of vanillin-5-$d_1$ and isovanillin-2,5,6-$d_3$.

3. Aerosol loadings corresponding to primary BBOA, oxidized BBOA, and/or SOA formed from oxidation of VOCs/IVOCs in the PAM reactor are not presented. In my opinion this data should be added to provide information about (1) the magnitude of SOA formation and corresponding SOA-to-BBOA ratio (2) phase partitioning of the selected biomass burning tracers. For example:

   a. C23, C25, C29 alkane signals decrease ~60%, ~70%, and ~75% following 9.8 days aging time (Fig. 3a). At an OH exposure of ~1.1E12 molec/cm$^3$/sec (8.5 days), Smith et al. (2009) observed ~70% decay of squalane particles subjected to heterogenous oxidation by OH. Thus, the observed C23, C25, & C29 decay rates are broadly consistent with heterogenous oxidation in the condensed phase. On the other hand, if the same compounds were oxidized in the gas phase, the observed decay rates should be much faster because the reaction is no longer rate-limited by diffusion of OH to the particle surface. Applying estimated gas-phase OH rate constants of 2.9E-11, 3.2E-11, and 3.8E-11 cm$^3$/molec/sec, for C23, C25, C29 alkanes (Kwok and Atkinson, 1995) suggests that ~100% of the alkanes should be reacted at only 3.4 days' OH exposure if the reaction occurs in the gas phase. Information about the experimental partitioning of this tracers would provide context for interpreting the observed decay rates.

   b. Levoglucosan signal decreases ~80% following 9.8 days aging time (Fig. 16). The authors reference literature rate constants of 3.09E-13 cm$^3$/molec/sec and 1.1E-11 cm$^3$/molec/sec. The levoglucosan decay rate reported in this paper is somewhere in between the referenced literature values. Is it possible that some of the discrepancy is related to phase partitioning? This is alluded to near the end of the paper (L558-L571), but it wasn't clear to me why the authors didn't explore this further by calculating the levoglucosan phase partitioning in the oak leaf and

oak wood experiments and comparing to phase partitioning in the literature studies.

    c. Increased condensed-phase partitioning of vanillin and other methoxyphenols following potentially significant SOA formation after ~3.4 days aging time (~4.4e11 molec/cm$^3$*sec) might explain their increase in concentration from 0 to 3.4 days' oxidation. At this approximate OH exposure, the "peak" SOA yield from oxidation of a specific precursor has been observed in previous studies, e.g. Lambe et al. (2012), Ortega et al. (2016). Although vanillin is relatively volatile, without knowing the aerosol loadings and ensuing partitioning, one can hypothesize plausible scenarios to explain some or all of the effect observed in Figure 3b.

I encourage the authors to expand their discussion to analyze the observed tracer decay rates in the context of the expected phase partitioning. They already report calculated C*'s, which, together with the aerosol loadings provided by AMS, facilitate this discussion. While I don't view it as the authors' responsibility to resolve the discrepancy in reported levoglucosan decay rates, it would certainly increase the impact of the paper if a plausible explanation is possible (L533-L549).

4. Photobleaching of biomass burning particles has reported in previous literatures studies, e.g. Zhao et al., ACP, 2015; Wong et al., ES&T, 2017. The authors should discuss the potential role of 254 nm photolysis in these experiments, especially in regard to degradation of condensed-phase aromatic species that strongly absorb 254 nm radiation and react relatively slowly with OH due to diffusion limitations. Were control experiments conducted with 254 nm radiation (no 185 nm radiation) and no ozone addition to investigate whether photolysis induces changes in BBOA composition?

5. To the extent possible, I recommend that the authors make additional effort to simplify, consolidate, and streamline the results that are presented, so that the reader is not overwhelmed – especially with the PMF results (see Comment #21).

**Technical/Minor Comments**

6. **L139**: Out of curiosity, what factor(s) led to the use of oak leaves and oak wood as opposed to, for example, a soft wood fuel that might have generated a much different range of tracers? Please briefly explain why the chosen systems were studied.
7. **L163**: Clarify that the chromate coating increases the electrical conductivity of the chamber, which decreases charge buildup, and consequently loss of charged particles to the walls of the reactor.

8. **L164-L166**: State here the range of ozone mixing ratios that were added to the reactor via the ozone chamber, and the range of ozone mixing ratios that were generated inside the reactor via 185 nm irradiance of $O_2$.

9. **L172**: Here, and elsewhere, please be more precise with statements such as "The role of RH in OH· formation…". Changing $[H_2O]$ does change the rate of OH formation, and from the text, it appears that the authors did manipulate $[H_2O]$. Changing RH by itself, however – for example, changing the temperature inside the reactor – does not change the rate of OH production.

10. **L183-L195**: It wasn't clear why the authors didn't simply add $SO_2$ during a "representative" combustion experiment to conduct an online OH exposure calibration in the presence of (I)VOCs that might have suppressed OH. I would certainly encourage this, if practical, as this approach should introduce less uncertainty than attempting to apply the OH exposure estimator when the OH reactivity of the biomass smoke emissions is not known.

11. **L282**: I suggest replacing "determined" with "inferred" or similar.

12. **L309**: This wording is confusing. Were oak leaves placed in a solvent to extract compounds on the surface of the leaves, and was this extract then injected into the TAG CTD? If so, please rewrite the sentence to clarify. What solvent(s) were used?

13. **L316-L318**: It's true that sinapaldehyde signal decays more quickly than other tracers (e.g. alkanes), but ~70% decay over 3.4 days' aging is still slow in the context of gas phase oxidation rates – this corresponds to an effective rate constant of ~2.7E-12 $cm^3$/molec/sec, whereas, for example, the gas-phase OH rate constant of syringol is 8.5E-11 $cm^3$/molec/sec (Lauraguais et al., 2015).

14. **L387-L395**: This paragraph seems out of place here, I would consider paraphrasing and moving to Conclusions.

15. **L417**: What is the signal-to-noise ratio for the m/z = 44 decomposition SICs? I understand that the SIC's presented are background corrected – how large are gas-phase $CO_2^+$ backgrounds compared to the background + sample m/z = 44 SIC's? This might be useful information to add to the Supplement.

16. **L441-L454 and Figure 10**: Implicit in this discussion is the observation that TAG recovery of highly oxidized/oxygenated species is low (even with inclusion of the thermal decomposition window). One or two sentences should be added that states this explicitly. Another point that should be made is that this attempt at a direct $f_{43}$ and $f_{44}$ comparison assumes AMS flash vaporization at T = 600 deg C and TAG thermal decomposition at T < 310 deg C produce the same m/z = 43 and m/z = 44 ion signals. It's not clear to me that this assumption is justified, but at the least, this assumption should also be stated explicitly.

17. **L483-L489**: Consider also moving this to Conclusions.

18. **L539-L549**: In my opinion, Lai et al.'s explanation for discrepancy in levoglucosan oxidation kinetics requires two unlikely scenarios:

a. using mz144 rather than mz162 would bias $k_{LG}$ ~ 30x too low -- Fortenberry et al.'s measurements are not subject to mass spectra interference either, and their levoglucosan decay rate is much closer to Kessler et al. than than Hennigan/Lai et al. (L564). A calculated levoglucosan + OH rate constant of 2.21E-13 $cm^3$/molec/sec (Bai et al., 2013), which is based on a theoretical study, may help put the different results in context.

OR

b. oxidation kinetics of OH + levoglucosan (or other model organics) are not first-order with respect to OH. Previous studies suggest otherwise (e.g. Renbaum and Smith, 2011).

I don't think it benefits the discussion in this paper to cite someone else's (in my opinion) incomplete explanation. I would consider removing it.

**Figures**

19. **Figure 2 and related text**: It is hard to distinguish the multiple shades of green in Fig. 2a, and for some compounds it is hard to distinguish changes in relative abundance between chromatograms representing "3.4 days" and "9.8 days". Please consider changing the colors in Fig. 2a. Additionally, consider removing the "3.4 days" TIC from Figs. 2a and 2b – this figure seems to be a general, "big picture" type of plot, so this would simplify the chromatogram without changing the take-home points.

20. **Figure 3 and related text**: Please add a subpanel plotting the concentrations of organics and any relevant inorganic species (e.g. $K^+$) measured by AMS following OH exposure in the PAM reactor.

21. **Figures 4-5, Figures 6-7, Figures 11-12, Figures 13-14:** I find these figures to be complex and overwhelming. I found it difficult to quickly "match up" 37 sets of chromatograms and mass spectra (15 + 18 + 4) for each PMF factor across separate figures as is currently presented. In my opinion, reorganizing these figures to put the mass spectra next to their corresponding chromatograms would improve their clarity and usefulness. Here is one idea for consideration:

Figs. 4, 6, 11, 13: Put each factor "TIC" for 0, 3.4 days, 9.8 days on the same x-axis, (as was done in Fig. 2). Choose a three colors, one each for 0, 3.4, 9.8 days, that are shared across all factors. Then place the corresponding mass spectra shown in Figs. 5, 7, 12, 14 to the right of the TICs. This modification would:

(i)      reduce the number of "PMF figures" from 8 to 4

(ii)     remove the number of subpanels in Figs. 4, 6, 11, 13 by 3x

(iii)    allow enough room to put the mass spectra from Figs. 5, 7, 12, 14 to the right of their chromatograms

> (iv)    make it unnecessary to use unique colors for each factor in attempt to match up the chromatograms and mass spectra across figures.

If this is not agreeable, the authors might consider just labeling the factors in Figs. 4, 6, 11, 13 and moving the mass spectral figures (5, 7, 12, 14) to the Supplement. This would save space/publication costs in the main part of the manuscript without making it any more difficult to "match up" the chromatograms and spectra.

> **22. Figure S13**: The caption states 15 micrograms of levoglucosan and 5 micrograms of quinic acid were injected. That seems like a very large analyte mass for a single compound injection – is there any chance this is a typo, and that the injected quantities were actually 15 and 5 nanograms?

**References**

Isaacman, G., Kreisberg, N. M., Yee, L. D., Worton, D. R., Chan, A. W. H., Moss, J. A., Hering, S. V., and Goldstein, A. H.: Online derivatization for hourly measurements of gas- and particle-phase semi-volatile oxygenated organic compounds by thermal desorption aerosol gas chromatography (SV-TAG), *Atmos. Meas. Tech*., 7, 4417-4429, https://doi.org/10.5194/amt-7-4417-2014, 2014.

Lavrich, R.J., and Hays, M.D. Validation studies of thermal extraction-GC/MS applied to source emissions aerosols. 1. Semivolatile analyte-nonvolatile matrix interactions. *Anal Chem*., 79(10):3635-45, 2007.

A. T. Lambe, H. J. Chacon-Madrid, N. T. Nguyen, E. A. Weitkamp, N. M. Kreisberg, S. V. Hering, A. H. Goldstein, N. M. Donahue, and A. L. Robinson. Organic Aerosol Speciation: Intercomparison of Thermal Desorption Aerosol GC/MS (TAG) and Filter-Based Techniques. *Aerosol Science and Technology*; *44(2)*, 141-151, 2010.

A. T. Lambe, T. B. Onasch, D. R. Croasdale, J. P. Wright, A. T. Martin, J. P. Franklin, P. Massoli, J. H. Kroll, M. R. Canagaratna, D. R. Wornsop, and P. Davidovits. Transitions from functionalization to fragmentation reactions of secondary organic aerosol (SOA) generated from the laboratory OH oxidation of alkane precursors. *Environ. Sci. Tech.*; 46, 5430-5437, DOI: 10.1021/es300274t, 2012.

Ortega, A. M., Hayes, P. L., Peng, Z., Palm, B. B., Hu, W., Day, D. A., Li, R., Cubison, M. J., Brune, W. H., Graus, M., Warneke, C., Gilman, J. B., Kuster, W. C., de Gouw, J., Gutiérrez-Montes, C., and Jimenez, J. L.: Real-time measurements of secondary organic aerosol formation and aging from ambient air in an oxidation flow reactor in the Los Angeles area, *Atmos. Chem. Phys*., 16, 7411-7433, https://doi.org/10.5194/acp-16-7411-2016, 2016.

Smith, J. D., Kroll, J. H., Cappa, C. D., Che, D. L., Liu, C. L., Ahmed, M., Leone, S. R., Worsnop, D. R., and Wilson, K. R.: The heterogeneous reaction of hydroxyl radicals with sub-micron squalane particles: a model system for understanding the oxidative aging of ambient aerosols, *Atmos. Chem. Phys*., 9, 3209-3222, https://doi.org/10.5194/acp-9-3209-2009, 2009.

Kwok, E.S.C. and Atkinson, R. Estimation of hydroxyl radical rate constants for gas-phase organic compounds using a structure-reactivity relationship: an update. *Atmos. Environ*., 29, 14, 1685-1695, 1995.

Zhao, R., Lee, A. K. Y., Huang, L., Li, X., Yang, F., and Abbatt, J. P. D.: Photochemical processing of aqueous atmospheric brown carbon, *Atmos. Chem. Phys*., 15, 6087-6100, https://doi.org/10.5194/acp-15-6087-2015, 2015.

J.P.S. Wong, A. Nenes, and R.J. Weber. Changes in Light Absorptivity of Molecular Weight Separated Brown Carbon Due to Photolytic Aging. *Environ. Sci. Technol.*, 51 (15), DOI: 10.1021/acs.est.7b01739, 8414–8421, 2017.

A. Lauraguais, I. Bejan, I. Barnes, P. Wiesen, and C. Coeur, Rate Coefficients for the Gas-Phase Reactions of Hydroxyl Radicals with a Series of Methoxylated Aromatic Compounds, *J. Phys. Chem. A*, *119* (24), 6179-6187, DOI: 10.1021/acs.jpca.5b03232, 2015**.**

Renbaum, L. H. and Smith, G. D.: Artifacts in measuring aerosol uptake kinetics: the roles of time, concentration and adsorption, *Atmos. Chem. Phys*., 11, 6881-6893, https://doi.org/10.5194/acp-11-6881-2011, 2011.

J. Bai, X. Sun, C. Zhang, Y. Xu, and C. Qi. The OH-initiated atmospheric reaction mechanism and kinetics for levoglucosan emitted in biomass burning, *Chemosphere*, 93, 2004-2010, http://dx.doi.org/10.1016/j.chemosphere.2013.07.021, 2013.

---

## Author Comment (AC1) · 13 Oct 2017

The authors have contacted the handling editor to request a deadline extension for the final response. We apologize for any inconvenience.

---

## Author Comment (AC2) · 6 Nov 2017

RESPONSES TO REVIEWER 1

Reviewer comments are italicized; author responses are in normal font.

*Fortenberry et al. present a study that examines links between chemically speciated fresh and aged BBOA data to more bulk measurements of the AMS. They use PMF technique to pull out mass spectral trends in the speciated data and looked mass spectra from compounds that eluted during the thermal decomposition window of the TAG. Their results show 60 m/z, a traditionally used ion for the AMS to track biomass burning, depends on fuel type and aging of the aerosol particles. They also suggest that 44 m/z ion could be used as an estimate for aerosol particle's aging state. This manuscript is written clearly and contributes to the understanding of both the complexity of BBOA and interpreting results from the AMS. I recommend this paper be published in ACP with some minor revisions.*

We thank the reviewer for his/her comments and insight. We address comments individually below. Where appropriate, approximate line numbers corresponding to the edited (with markup) manuscript provided, along with line numbers relative to the section/paragraph number.

**Minor Comments:**

*Line 38: parallel structure, change to "and impacts" or something along those lines*

In line 36 (Section 1, Paragraph 1, Line 2), "…and can impact the global energy balance…" was changed to "… and impacts the global energy balance…"

*Line 39: awkward sentence starting with "Organic aerosol. . ." In addition, aerosol refers to both the particle and gas phase. When the authors mean particle phase, please change aerosol to aerosol particle.*

We reworded this sentence to improve clarity (lines 37-39; Section 1, Paragraph 1, Lines 7-9):

"Organic aerosol (OA) particles compose 20-90% of submicron PM ($PM_1$) and may consist of thousands of distinct organic compounds (Goldstein and Galbally, 2007; Ng et al., 2010; Zhang et al., 2007)."

In addition, we minimized our use of the word "aerosol" throughout the text, instead using "particles," "gases," and "emissions" where necessary.

*Line 164: Traditionally, the radical dot of OH is left off, though the authors are technically correct. The dot just looks a bit strange when used, for example, in line 580.*

To remain consistent with previous literature, we removed the radical dot from each reference to the hydroxyl radical.

*Line 166: How well-mixed in the PAM reactor? Could large concentration gradients in aerosol particles affect the observed results?*

Recent work demonstrates that the PAM reactor used in this study is approximately well mixed if sufficient time is given prior to sample collection to establish a well-mixed and near steady-state concentration throughout the combustion chamber and PAM chamber (Mitroo, 2017). We therefore do not expect concentration gradients within the PAM reactor to significantly impact observed results.

This question has been addressed in-text with the addition of the following sentence in lines 219-223 (Section 2.3, Paragraph 7):

"Flow field simulations and chemical tracer tests have demonstrated that the PAM reactor used in this study is approximately well mixed if sufficient time (at least 15 minutes) is given prior to sample collection to establish a well-mixed and near steady-state concentration throughout the combustion chamber and PAM chamber (Mitroo, 2017). The TAG therefore consistently collected 30 minutes after the biomass heat pulse to minimize particle concentration gradients within the reactor."

*Line 180: add fuels after leaf*

In line 188 (Section 2.3, Paragraph 2, Line 9), "For both heartwood and leaf, …" was changed to "For both heartwood and leaf fuels, …"

*Line 210: What is the rate at which the cells are heated to 310ËŽC? There is some discussion that heating rate will affect which compounds desorb vs. thermal decompose. The authors do illustrate the ramping time in Figure 1, but it would be helpful to have it written down in the text.*

In line 258 (Section 2.4.1, Paragraph 1, Line 2), a typical collection and thermal desorption heating rate of $50°C$ $min^{-1}$ has been added to the text per the reviewer's suggestion.

*Line 222: combustion chamber were clean*

Here (now line 278; Section 2.4.1, Paragraph 3, Line 4), we clarified the language by replacing "… to ensure that the emissions and combustion chamber was clean prior to …" with "… to ensure that both the emissions chamber and the PAM reactor were clean prior to …"

*Line 469: have specific standards been observed to decompose at these ions in this thermal decomposition window?*

We thank the reviewer for this question. We have not observed these ions in any recent standard work, though this is an area of active research.

Developing satisfactory analytical standards for the TAG decomposition window has been particularly challenging. While we can tentatively identify the fragments eluting in the decomposition window using available mass spectral identification tools, we often cannot infer the source of the fragments, since they are products of compound thermal decomposition rather than volatilization. With that, many of the compounds undergoing decomposition during sample desorption are potentially too involatile for typical GC-MS analysis.

We have expanded our conclusion to include a discussion of the challenges of interpreting thermal decomposition window data (now lines 766-774; Section 4, Paragraph 6):

"The utility of the thermal decomposition window is limited by a lack of adequate analytical standards, particularly for organic components. Although ammonium sulfate and ammonium nitrate standards have been used to quantify sulfate and nitrate particles in previous work (Williams et al., 2016), the development of satisfactory standards for decomposing organics remains difficult for several reasons. While fragments eluting in the decomposition window may be tentatively identified using available mass spectral identification tools, we often cannot infer the source of the fragments, since they are products of compound thermal decomposition rather than volatilization. Many of the compounds undergoing decomposition during sample desorption are therefore too thermally labile for typical GC-MS analysis, and the original molecular structure remains undetermined. Despite these challenges, analytical standards composed of complex organic mixtures are currently under development to aid interpretation of decomposition window results based on molecular functionality."

*Line 534 (though may happen earlier): No comma after Kessler. This comma is not needed as Kessler and others (in English) requires no comma even when used in line. Please remove comma in previous+subsequent usages.*

We thank the reviewer for pointing out this mistake. The unnecessary commas have been eliminated from in-line references throughout the text.

*Figure 2 (a): The three types of green are very difficult to distinguish. I understand the authors were aiming for green=leaf and warm colors for heartwood, but greens all look the same. Maybe cool colors for leaf and warm colors for heartwood?*

The colors in Figure 2(a) have been changed to teal, purple, and pink to improve figure readability. Subsequent leaf BBOA plots have been changed to match this color scheme.

*Figure 3 (b): Why are syringol, syringaldehyde, and vanillin dotted lines? Is it because they are observed to increase with aging time? If so, please mention that in the caption.*

These compounds were indicated with the dotted lines to visually distinguish that unlike the rest of the compounds, they do not directly decrease abundance with equivalent aging. The following sentence has been added at the end of the figure caption:

"Compounds that decrease in abundance are indicated with solid lines, while compounds that deviate from this trend are displayed with dotted lines."

*Figure 8 (a): The greens are hard to distinguish.*

As with figure 2(a), the colors were changed to teal, purple, and pink to improve figure readability.

*Figure 10 (a): The green color gradient is not great and implies there was a near continuous gradient of samples collected for that range of aging time. There were only three sampled times, so a gradient seems a bit misleading. Also, the colored right-hand and top axes are a bit confusing because the reader is trying to match the green points to the green axis instead of TAG to green. Also, the caption says the dotted lines are guidelines for where the points tend to be concentrated. This doesn't seem to be the case for the green AMS points. The blue dotted line for these points would look more like a rectangle covering the bottom half of the graph then a downward pointing cone. Is there greater meaning behind this cone?*

We agree with the reviewer that the continuous color gradient may be misleading. In both figure 10(a) and 10(b), we have changed the legend to reflect the three distinct aging times for each fuel type. We have also changed the color scheme in figure 10(a) to match the colors used in figure 2(a).

The triangle formed by the dotted lines provides a visual guideline for the evolution of OA chemical composition in f43/f44 space. The apex of the triangle formed by the lines indicates the direction of increasing photochemical oxidation. These triangles have been used as visual aids in previous presentations of AMS f43 and f44 data (Ng et al., 2010) and are provided by default by the AMS Squirrel analysis software. We agree that in the case of our leaf BBOA AMS data, the points do not fit well within the confines of the dotted lines, so we have modified the position of the triangle to better fit our AMS data points. In addition, we have clarified the functionality of the triangle in the figure caption (what is now Figure 8) by modifying the last sentence:

"The triangles formed by the blue dotted lines provide visual guidelines for the evolution of OA chemical composition across $f_{44}$ vs $f_{43}$ space; the apex of the triangle indicates the direction of photochemical oxidation for AMS measurements (Ng et al., 2010)"

*Figure 15 (A): same comment about the green.*

As with figure 2(a), the colors were changed to teal, purple, and pink to improve figure readability.

*Even more minor comments for the SI:*

*Figure S4: the TAG collection 1 blue shaded region does not look to start at 30 minutes after start of heat pulse as the caption indicates.*

Now Figure S5, the location of the shaded region has been fixed to start at 30 minutes after the heat pulse.

*Table S5-6: The raw SIC integration numbers have too many significant digits (and is difficult to read). Maybe consider limiting it to 2-3 with scientific notation.*

The raw SIC integrations have been fixed as suggested: the values now have 3 significant digits and are presented in scientific notation.

**Literature Cited:**

Goldstein, A. H. and Galbally, I. E.: Known and Unexplored Organic Constituents in the Earth's Atmosphere, Environ. Sci. Technol., 41(5), 1514–1521, doi:10.1021/es072476p, 2007.

Mitroo, D.: Applications and Flow Visualization of a Potential Aerosol Mass Reactor, PhD Thesis, Washington University in St. Louis, St. Louis, MO., 2017.

Ng, N. L., Canagaratna, M. R., Zhang, Q., Jimenez, J. L., Tian, J., Ulbrich, I. M., Kroll, J. H., Docherty, K. S., Chhabra, P. S., Bahreini, R., Murphy, S. M., Seinfeld, J. H., Hildebrandt, L., Donahue, N. M., DeCarlo, P. F., Lanz, V. A., Prévôt, A. S. H., Dinar, E., Rudich, Y. and Worsnop, D. R.: Organic aerosol components observed in Northern Hemispheric datasets from Aerosol Mass Spectrometry, Atmos Chem Phys, 10(10), 4625–4641, doi:10.5194/acp-10-4625-2010, 2010.

Williams, B. J., Zhang, Y., Zuo, X., Martinez, R. E., Walker, M. J., Kreisberg, N. M., Goldstein, A. H., Docherty, K. S. and Jimenez, J. L.: Organic and inorganic decomposition products from the thermal desorption of atmospheric particles, Atmos Meas Tech, 9(4), 1569–1586, doi:10.5194/amt-9-1569-2016, 2016.

Zhang, Q., Jimenez, J. L., Canagaratna, M. R., Allan, J. D., Coe, H., Ulbrich, I., Alfarra, M. R., Takami, A., Middlebrook, A. M., Sun, Y. L., Dzepina, K., Dunlea, E., Docherty, K., DeCarlo, P. F., Salcedo, D., Onasch, T., Jayne, J. T., Miyoshi, T., Shimono, A., Hatakeyama, S., Takegawa, N., Kondo, Y., Schneider, J., Drewnick, F., Borrmann, S., Weimer, S., Demerjian, K., Williams, P., Bower, K., Bahreini, R., Cottrell, L., Griffin, R. J., Rautiainen, J., Sun, J. Y., Zhang, Y. M. and Worsnop, D. R.: Ubiquity and dominance of oxygenated species in organic aerosols in anthropogenically-influenced Northern Hemisphere midlatitudes, Geophys. Res. Lett., 34(13), L13801, doi:10.1029/2007GL029979, 2007.

---

## Author Comment (AC3) · 6 Nov 2017

RESPONSES TO REVIEWER 2

Reviewer comments are italicized; author responses are in normal font.

*This paper describes experiments and analysis examining the composition of fresh and aged aerosol emissions from volatilization/combustion of oak wood and leaf matter using powerful and novel analytical methods. The paper describes repeated experiments heating the biomass in a combustion chamber and sampling it via a Potential Aerosol Mass (PAM) reactor into a Thermal desorption Aerosol Gas Chromatograph – Mass Spectrometer (TAG), High-Res AMS and a SMPS. Emissions were sampled unaged and under two different PAM aging conditions. TAG data were analyzed in a number of ways, including for specific eluted compounds, positive matrix factorization of chromatograms, and analysis of chemical fragments from thermal decomposition of particles. These analyses are compared to more 'standard' analyses of bulk aerosol composition from the AMS.*

*The suite of different approaches taken to analyze these data leads to a number of interesting and potentially important conclusions that will be of interest to the broad community interested in emissions from biomass burning and how they evolve in the atmosphere. For example, the emission of aliphatic aldehydes from leaf coating volatilization is nicely supported. The emission and evolution of components contributing mass at m/z60 in both TAG and AMS spectra receives special attention and evidence for contribution from components formed during oxidation to mass at m/z 60 is given. It does an especially nice job of spanning levels of chemical detail from compound-specific determination using the TAG, to PMF of tag chromatograms to point to compound classes, to the AMS measurements of bulk fragments.*

*Overall, this is a very nicely written and clear paper that makes an important contribution to understanding of a complex and important source of atmospheric aerosol. Therefore, I find it to be suitable for publication in ACP once my concerns are addressed. I have identified a number of points that, when addressed, will help the paper better fit into the existing literature on the topic.*

We thank the reviewer for his/her insight. We address each comment individually below (the reviewer's major and minor points have been numbered for ease of reference). Where appropriate, approximate line numbers corresponding to the edited (with markup) manuscript provided, along with line numbers relative to the section/paragraph number.

**Major points:**

**[1]** *While the analytical methods applied here are unique and provide strong insights, I have a concern about how these results can be compared to other 'forms' of biomass combustion, and so I think that more effort should be made to qualify/compare the types of emissions that were sampled. The emphasis in emission generation was clearly on repeatability and consistency, rather than on representativeness, which makes sense for these experiments. However, it would be helpful to put the OA studied here a bit more clearly in the context of 'biomass burning OA (BBOA)', which typically refers to ambient observations of biomass burning emissions emitted from a range of different fuels/types/combustion conditions. In this case, the nature of the 'combustion' that was the source of the sampled aerosols is someone unclear to me. Very small portions of biomass (0.2-0.5 g) were 'combusted' in the chamber, but it is not clear to me if flame was involved, or strictly smoldering, and so how the results might be compared to what might come from a fire. For example, flame typically produces black carbon, was any generated here? The experiments are called 'devolatilization and combustion', but is there any way to classify this combustion more broadly or put it in the context of biomass burning more generally? If not, can anything be said about the representativeness of the emissions from this setup relative to other studies?*

The authors agree this point is important to clarify for future comparisons of lab and field observations. To provide better context for the type of emissions sampled here, we expanded the description of the devolatilization and combustion process in the Materials and Methods section. Specifically, while we had previously specified that smoldering combustion was observed only in the final minute of the heat ramp, we now also clarify that no flaming combustion occurred during any of the experiments (lines 153-154; Section 2.2, Paragraph 2, Lines 5-6)

We also provide a more thorough discussion of the influence of combustion conditions on the relevance of our measurements to real world systems. We provide more detailed discussion in the Conclusion section of the main text (lines 737-748; Section 4, Paragraph 3):

"Based on previous studies, combustion conditions are expected to significantly impact the chemical composition of both primary and secondary BBOA (Ortega et al., 2013; Reece et al., 2017; Weimer et al., 2008; see "AMS Chemical Characterization" in Supplemental Information). The resistive heating technique applied in these experiments allows for the isolation of devolatilization (pre-combustion) and low-temperature (≤300°C) smoldering conditions, which is difficult to achieve in combustion chambers that require ignition of a flame. For example, Tian et al. designed a chamber that allows the user to control the relative contributions of smoldering and flaming combustion, though smoldering combustion is only achieved in this chamber following the introduction of a flame to the biomass fuel (Tian et al., 2015). The devolatilization and combustion procedure presented here is thus advantageous for investigating aerosol from small masses of biomass fuel under tightly controlled conditions. However, these results alone are likely not representative of many real-world fire systems, where smoldering combustion often occurs alongside flaming combustion. Our results may therefore serve to complement field measurements, where either smoldering or flaming combustion may dominate, as well as laboratory studies where combustion conditions are controlled."

In addition, we have added a new section in Supplemental Information ("AMS Chemical Characterization"; see response to Major Comment #2) in which we compare AMS chemical composition to relevant data reported in literature.

**[2]** *The relative change in f60 for the study of Ortega et al. is shown in Fig. 16, but the chemical character of the OA is not compared to that measured in that or other studies, even of the same type of fuel. Several studies have shown that combustion phase/type can have a substantial effect on OA emission properties (Weimer et al. 2008; Reece et al. 2017) and as you noted, observations of SOA production in lab and field studies have been found to be highly variable and distinct. Therefore, to the extent that you can include information about your combustion and the basic characteristics of your emissions, it will enable comparison with existing measurements and analysis.*

In addition to expanding the description of the devolatilization/combustion characteristics (see major comment #1), we have reviewed the recommended sources and now discuss them in our expanded analysis. To more thoroughly compare the chemical character of the OA produced here to previous measurements, we have added a section to the Supplemental Information titled "AMS Chemical Characterization." This section includes the following figures:

- An average AMS mass spectrum for each fuel at each level of oxidation, as well as difference mass spectra (Figure S13)
- A van Krevelen plot for both heartwood and leaf BBOA (Figure S14)
- Total organics, potassium, and sulfate concentrations (Figure S15)

In this section, we discuss the impact of combustion techniques and conditions on the chemical composition of both aged and unaged oak BBOA, contextualizing our results with information from previous studies. We refer to this section in the manuscript in lines 336-339 (Section 3.1, Paragraph 2):

"According to AMS mass spectra, the BBOA measured in these experiments is chemically consistent with BBOA from similar oak fuel sources, though with key differences related to combustion conditions (Cubison et al., 2011; Ortega et al., 2013; Reece et al., 2017; Weimer et al., 2008). Detailed analysis and contextualization of the AMS chemical composition data is provided in Supplemental Information (Section: AMS Chemical Characterization)."

**[3]** *In a similar vein, one of the motivations discussed for the use of the PAM was to understand SOA production, but this is never discussed in the paper, though some evidence is presented in Table S2 that there is SOA production for the wood experiments but not for leaf experiments. These outcomes are of interest in the context of the variability in SOA production discussed above, but also because they may influence interpretation of the 'relative to unaged' presentation of data that is used in a number of figures (e.g. Fig. 3, 16). For example, are changes in fragments/compounds due to 'dilution' of primary OA by SOA, or strictly due to gas-phase or heterogeneous oxidation? This is mentioned in the paper's final paragraph, but it seems at least some further evidence/data could be presented.*

Decoupling effects of SOA formation from other processes occurring in the PAM reactor remains challenging and is the subject of ongoing study. However, we have added discussion of the potential relative contributions of other mechanisms (e.g. gas-particle partitioning and heterogeneous vs homogeneous reactions) for certain key species. We address each of these changes below:

- In lines 390-401 (Section 3.2.1, Paragraph 3), we use gas-phase reaction rate constants from literature (Kwok and Atkinson, 1995) to justify the assumption that gas-phase reactions contribute little to overall trends in oak leaf tracer compounds.
- Similarly, in lines 407-411 (Section 3.2.1, Paragraph 4, Lines 6-10), we use the literature-reported sinapaldehyde/OH gas phase reaction rate constant (Lauraguais et al., 2015) to justify the assumption that sinapaldehyde decay occurs primarily in the particle phase.
- We calculated approximate particle-phase partitioning fractions (Table S12) for levoglucosan in both heartwood and leaf BBOA and include an expanded discussion of levoglucosan phase partitioning in lines 680-694 (Section 3.4, Paragraph 8).
- We evaluated phase partitioning for syringol, syringaldehyde, and vanillin, species in the heartwood BBOA that increase in abundance with photochemical aging (e.g. those that may be formed through secondary processes). These calculations are discussed in lines 415-431 (Section 3.2.1, Paragraphs 5-6).

**[4]** *The authors rightly point out that the operation of the PAM during experiments was not fully constrained by the $SO_2$ calibration of integrated OH exposure, but then in the paper use quite tightly constrained values (3.4 and 9.8 days) of equivalent oxidation to describe the aging under the two operation conditions. The fact that there are repeated experiments and repeatable results is great (and difficult to do for biomass burning) and suggests that aging within an experiment tyep should be consistent. However, your 'sensitivity' analysis (Table S1) shows that actual OH exposure estimates for your experiments may vary by a factor of 5 to 10 given the assumed range of external OH reactivities. Therefore, it seems strange to specify your aging conditions to such a precise degree. I would feel more comfortable if a range of days were reported or if you can find a way to estimate OH reactivity during your experiments (e.g. using published VOC profiles and a tracer ratio?) to better constrain this. At the very least, uncertainty in this value should be clearly stated when it is discussed (e.g. in the context of Table 1), so that the values given are not over-interpreted.*

This is an excellent comment, and the authors have considered thoroughly the best way to handle this point. To better constrain the equivalent aging times characteristic of our system, we conducted additional experiments to estimate total gas-phase $OHR_{ext}$. A full description of these experiments is provided in Supplemental Information ("Methods: PAM Calibrations and Equivalent Aging Estimations, "Estimation of External OH Reactivity ($OHR_{ext}$)"). In the main text, we discuss these experiments briefly in lines 191-218 (Section 2.3, Paragraphs 3-6).

In these experiments the burn procedure was repeated for both leaf and heartwood fuels, and CO was measured using a trace-level CO monitor. Aerosol was sampled alternately through the PAM reactor and through a bypass line to obtain CO measurements for aged and unaged emissions. During these experiments, the PAM light settings corresponded to approximately 3 equivalent days of aging according to the most recent offline calibration.

With this method, we found that in both types of BBOA, aged and unaged CO concentrations exhibited little variation, giving $OHR_{ext}$ values of 0.56 and 0.52 s$^{-1}$, respectively. To estimate total $OHR_{ext}$, we took the reviewer's suggestion and estimated typical emissions by scaling published VOC profiles from laboratory studies of oak forest emissions (Burling et al., 2010) to our measured CO concentrations. Using rate constants from the NIST Chemical Kinetics Database, we obtained $OHR_{ext}$ for each relevant species, and, taking the sum of all calculated species-specific $OHR_{ext}$, obtain a total $OHR_{ext}$ of 2.21 s$^{-1}$ and 2.17 s$^{-1}$. We therefore use 2.2 s$^{-1}$ in subsequent estimations of $OH_{exp}$ and equivalent aging time ranges.

In addition, to better inform our use of the Oxidation Flow Reactor Exposure Estimator tool, we measured reactor-produced $O_3$ with no external $O_3$ addition and report the measurements in the supplemental information (Table S1). Based on these data, and assuming an $OHR_{ext}$ of 2.2 s$^{-1}$, we obtain a lower limit for equivalent aging times at each level of oxidation, which we provide in Table 1. We retain the $OH_{exp}$ values calculated from the offline $SO_2$ calibration as the upper limits for each oxidation condition, but we reduce the significant figures in the corresponding equivalent aging times from two to one.

**Specific points**

**[5]** *Page 4; Line 138 - Initially it was unclear to me whether this heading referred to separate experiments or one type. As noted above, more of an effort should be made to describe/qualify the approach taken and how the resultant emissions compare to what might come from a fire. In addition, it would make sense to be clear and consistent when using 'BBOA' in the context of your experiments.*

To better qualify our experimental setup and approach, we have added expanded discussion of the combustion characteristics of our procedure, contextualizing our results with data from previous studies (see Major Comment #1)

*Page 5; Line 165 - Was level of external O3 injection always the same?*

Yes, the level of external O3 injection was consistent. In all experiments, we passed 0.4 L min$^{-1}$ of oxygen through the lamps, which were held at a constant intensity (lines 171; Section 2.3, Paragraph 1, Line 7).

*Page 6; Line 194-195 - As noted above, this uncertainty should be reflected in estimated atmospheric ages used throughout paper.*

We thank the reviewer for this suggestion and believe we have addressed his/her concerns above (see Major Comment #4.)

*Page 7; Line 234-235 - Also related to combustion emission properties. Why are SMPS volumes used and not AMS OA concentrations? For example, if there is a contribution from BC, this will both effect the grdetermination of OA mass by adding to volume, and also potentially affecting DMA sizing. This may not be an issue, but could at least compare AMS OA to SMPS volume?*

We used SMPS volumes rather than AMS total OA concentrations because we did not obtain satisfactory AMS data during the first set of TAG experiments (lines 509-521; Section 3.3, Paragraph 2). We do agree that normalizing to AMS total organic concentrations would otherwise be more appropriate, and we plan to do so where possible in all future experiments.

We have provided a table in supplemental information (Table S3) with total organic concentrations alongside co-measured SMPS volume concentrations.

*Page 9; Line 298 - Need to be clear that this is referring to relative abundance - important if SOA production is 'diluting' primary species.*

As suggested, we have modified the sentence to specify that this is a relative abundance (lines 376-377; Section 3.2.1, Paragraph 2, Lines 4-5):

"Nearly all compounds identified after 35 minutes decrease in relative abundance with photochemical aging."

*Page 9; Line 296 - I noted this included in Supplemental tables, but it might be helpful to translate to effective saturation concentration.*

As requested, we have added this information into the main text (now lines 368-371; Section 3.2.1, Paragraph 1, Lines 9-10):

"Based on even-numbered alkane standard injections, compounds eluting after minute 35 exhibit approximate saturation vapor pressures not exceeding that of docosane (approximately $2.73 \times 10^{-5}$ torr at 25ºC), which corresponds approximately to $\log_{10}(C^*) = 2.76$"

We also include saturation concentration values in our discussion of oak heartwood compound volatilities (lines 403-405; Section 3.2.1, Paragraph 4, Lines 2-4):

"Based on even alkane standard injections, compounds eluting within this time window exhibit approximate vapor pressures within $4.52 \times 10^{-3}$-$2.73 \times 10^{-5}$ torr at 25ºC ($\log_{10}(C^*) \approx 4.85$-$2.76$; Table S4 in Supplemental Information; ACD/Labs, 2017)."

*Page 9; Line 299 - Where possible (e.g. Fig 3), would be best to include error bars to show inter-test variability. You have done this in some places, but would be good to see it here.*

We thank the reviewer for the suggestion. We initially tried including error bars in this figure, but found that it made the figure difficult to read. For this reason, we included raw chromatographic abundances and errors in the supplemental information (Tables S6 and S7). We now refer the reader to these tables in the caption of figure 3.

*Page 10; Line 323-324 - These don't seem to be fully depleted -seems to be 50-100% of relative abundance at the start?*

We have changed "fully depleted" to "depleted" in this sentence (line 417; Section 3.2.1, Paragraph 5, Line 3).

*Page 12; Line 412-413 - A useful comparison to quantify inter-test variability might be to do this calculation on repeated experiments at same loading. E.g., what are dot products between repeated tests at same conditions that are averaged together for other analyses?*

We thank the reviewer for the suggestion. We have repeated the procedure and provide the results in supplemental information (Table S9).

*Page 13; Line 425 - Isn't really clear if this is indicating an increase in the presence of material containing mz44 that can thermally decompose or an increase in thermal decomposition?*

While this is an interesting question, the temperature cycle is consistent from run to run and any change in this signal would be due to a combination of the type of material present and the associated thermal decomposition potential of that material. It has been shown in past work that this signal correlates best with oxygenated OA concentrations (Williams et al. 2016). With the lack of adequate standards, we do not know the relative decomposition potential of all types of OA, and include this point in our discussion of the need for standard calibrations in the Conclusions section of the manuscript (lines 766-774; Section 4, Paragraph 6).

The text has been updated to reflect this point (lines 534-538; Section 3.3.1, Paragraph 1, Lines 12-16):

"For both types of BBOA, the decomposition *m/z* 44 integrated signal increases overall from 0 days to 6-10 days of equivalent aging, indicating an increase in OA material that can thermally decompose with increased PAM oxidation. This trend is consistent with relative increased decomposition of highly oxidized aerosol within the PAM reactor, as was also indicated in previous ambient aerosol observations (Williams et al., 2016)."

*Page 13; Line 431 - As noted above, to be most useful, this should be placed in the context of other BBOA measured by AMS. How do these numbers compare to those measured in other studies - e.g. Ortega et al, 2013, Reece et al, 2017*

We thank the reviewer for this suggestion and believe that we have addressed his/her concerns (see Major Comment #3).

*Page 13; Line 433-436 - Significant figures not justified (or, really, linear regression advised) for 3 data points unless there is a very strong argument for there being a linear relationship*

In general, we agree with the reviewer, and we reduce the number of significant figures in our regressions to 1. We retain the linear regression because we feel that it serves as a useful comparison between trends in leaf and heartwood BBOA *m/z* 44 signals.

*Page 13; Line 439 - It seems as or more plausible that fragmentation leads to move volatile species that aren't captured by the TAG?*

Implicit in this explanation is that the TAG does not capture the highly volatile products of fragmentation. We have modified the sentence to clarify this point (lines 547-551; Section 3.3.1, Paragraph 2, Lines 6-10):

"The non-linear trend in TAG decomposition $m/z$ 44 for leaf BBOA may indicate a shift in the dominant oxidation mechanisms between moderate and high levels of OH within the PAM chamber; at the highest $OH_{exp}$, primary gas and/or particle-phase components may undergo increased fragmentation, leading to a net decrease in production of the aged OA that thermally decomposes during TAG analysis, along with an increase in highly volatile fragmentation products that are not captured by the TAG."

*Page 15; Line 504-505 - This is a good point, but here the distinction may be as much type of emission/combustion as type of biomass, as it appears that at least some OA is from volatilized leaf coating so is not 'burned' (for leaf e.g. Fig. S7)*

We have modified this point to acknowledge the role that combustion characteristics play in measured $m/z$ 60 (lines 621-623; Section 3.4, Paragraph 2, Lines 10-12):

"Additionally, the presence of $m/z$ 60 is likely dependent on the combustion characteristics, as combustion processes can influence the emission and phase of different compounds."

*Page 15; Line 511-513 - If possible, it would be helpful to quantify (even approximately) the relative amount of material contributing $m/z$ 60 in the compound window vs decomposition window. I take it there is more in the former? The AMS will presumably see a weighted average of the two?*

We thank the reviewer for this suggestion. We have quantified relative $m/z$ 60 abundances as percentages for both compound and decomposition window signals and provide them in a separate supplemental table (Table S11). Based on these calculations, the AMS likely does measure an approximately weighted average of the two signals, which is displayed in figure 12 (was figure 16 previously).

Minor points

*Page 7; Line 219 - data were, not data was*

"Data was" has been changed to "data were" in this sentence (line 275; Section 2.4.1, Paragraph 3, Line 1) and in subsequent occurrences.

*Page 14; Line 461 - I think I know what 'triplicate averages' is meant to say, but can be said more clearly.*

We have modified this sentence for clarity (Lines 524-526; Section 3.3.1, Paragraph 1, Lines 2-4):

"At each oxidation condition, SICs from the triplicate chromatograms were blank subtracted, normalized to maximum volume concentrations, and averaged to obtain the displayed trace."

*Page 15; Line 508 - I think 'distinct' would work better than 'unique'.*

As suggested, "unique" has been replaced with "distinct" in this sentence (line 627; Section 3.4, Paragraph 3, Line 4).

*Page 15; Line 506-507 - Would be good to point to Fig. 16 here.*

We now reference figures 11 and 12 (figure 12 was figure 16 previously) in line 626 (Section 3.4, Paragraph 3, Line 3).

*Page 17; Line 578-579 - Not sure if a species can be called a 'tracer' (for a primary source) if it is increasing w/ atmospheric processing. At the very least, it's not a tracer of a unique source.*

We have replaced "tracers" with "components present in freshly-emitted BBOA" in this sentence (line 723; Section 4, Paragraph 1, Line 6).

Literature Cited:

Burling, I. R., Yokelson, R. J., Griffith, D. W. T., Johnson, T. J., Veres, P., Roberts, J. M., Warneke, C., Urbanski, S. P., Reardon, J., Weise, D. R., Hao, W. M. and de Gouw, J.: Laboratory measurements of trace gas emissions from biomass burning of fuel types from the southeastern and southwestern United States, Atmos Chem Phys, 10(22), 11115–11130, doi:10.5194/acp-10-11115-2010, 2010.

Cubison, M. J., Ortega, A. M., Hayes, P. L., Farmer, D. K., Day, D., Lechner, M. J., Brune, W. H., Apel, E., Diskin, G. S., Fisher, J. A., Fuelberg, H. E., Hecobian, A., Knapp, D. J., Mikoviny, T., Riemer, D., Sachse, G. W., Sessions, W., Weber, R. J., Weinheimer, A. J., Wisthaler, A. and Jimenez, J. L.: Effects of aging on organic aerosol from open biomass burning smoke in aircraft and laboratory studies, Atmos Chem Phys, 11(23), 12049–12064, doi:10.5194/acp-11-12049-2011, 2011.

Kwok, E. S. C. and Atkinson, R.: Estimation of hydroxyl radical reaction rate constants for gas-phase organic compounds using a structure-reactivity relationship: An update, Atmos. Environ., 29(14), 1685–1695, doi:10.1016/1352-2310(95)00069-B, 1995.

Lauraguais, A., Bejan, I., Barnes, I., Wiesen, P. and Coeur, C.: Rate Coefficients for the Gas-Phase Reactions of Hydroxyl Radicals with a Series of Methoxylated Aromatic Compounds, J. Phys. Chem. A, 119(24), 6179–6187, doi:10.1021/acs.jpca.5b03232, 2015.

Ortega, A. M., Day, D. A., Cubison, M. J., Brune, W. H., Bon, D., de Gouw, J. A. and Jimenez, J. L.: Secondary organic aerosol formation and primary organic aerosol oxidation from biomass-burning smoke in a flow reactor during FLAME-3, Atmos Chem Phys, 13(22), 11551–11571, doi:10.5194/acp-13-11551-2013, 2013.

Reece, S. M., Sinha, A. and Grieshop, A. P.: Primary and Photochemically Aged Aerosol Emissions from Biomass Cookstoves: Chemical and Physical Characterization, Environ. Sci. Technol., 51(16), 9379–9390, doi:10.1021/acs.est.7b01881, 2017.

Tian, J., Chow, J., Cao, J., Han, Y., Ni, H., Chen, L.-W. A., Wang, X., Huang, R., Moosmüller, H. and Watson, J.: A Biomass Combustion Chamber: Design, Evaluation, and a Case Study of Wheat Straw Combustion Emission Tests, Aerosol Air Qual. Res., 15(5), 2104–2114, 2015.

Weimer, S., Alfarra, M. R., Schreiber, D., Mohr, M., Prévôt, A. S. H. and Baltensperger, U.: Organic aerosol mass spectral signatures from wood-burning emissions: Influence of burning conditions and wood type, J. Geophys. Res. Atmospheres, 113(D10), D10304, doi:10.1029/2007JD009309, 2008.

---

## Author Comment (AC5) · 6 Nov 2017

RESPONSES TO REVIEWER 3

Reviewer comments are italicized; author responses follow in normal font.

*Fortenberry et al. present chemical composition measurements of photochemically aged laboratory biomass burning organic aerosols (BBOA). BBOA was generated from the combustion of oak leaves and oak wood samples in a burn chamber, then exposed to OH radicals in a Potential Aerosol Mass oxidation flow reactor. Ensemble aerosol mass spectra were obtained with an AMS, and GC-MS samples were obtained with a TAG. The authors used factor analysis to identify characteristic groups of GC effluent signals that behaved differently as a function of OH exposure. In my opinion, the manuscript presents an interesting experiment and application of the measurement techniques that were used to mimic aging of BBOA surrogates. Publication in ACP may be appropriate after consideration of my comments below.*

We thank the reviewer for his/her insight and address each comment individually below. Where appropriate, approximate line numbers corresponding to the edited (with markup) manuscript provided, along with line numbers relative to the section/paragraph number.

*General/Major Comments*

1. *Given the goal of using TAG measurements to interpret ensemble/bulk techniques such as the AMS, and given the large number of oxygenated/polar compounds present in BBOA (and oxidized BBOA), it wasn't clear to me why the authors chose not to incorporate the online derivatization technique used in previous TAG measurements (Isaacman et al., 2014), which reports "complete derivatization of [...] alkanoic acids, polyols, diacids, sugars, and multifunctional compounds." In principle, derivatization should offer the following advantages:*
    a. *improved recovery of methoxyphenols, levoglucosan, and other sugars and primary species that are measured in this work, along with potentially less significant matrix effects (see Comment #2).*
    b. *recovery of oxidation products formed following OH exposure in the PAM reactor (e.g. dicarboxylic acids) that were not resolved here.*
    c. *evaluation/supplementation of the thermal decomposition window because the TAG recovery and resolution of highly polar compounds is still low, as implied by discussion in L441-L454 (see Comment #16) and L493-L494. The authors should explain in the manuscript why they chose not to incorporate/adapt the TAG derivatization technique published by Isaacman et al.*

    Although online derivatization would provide several advantages for measuring both primary and secondary BBOA components, the derivatization technique used in Isaacman et al., 2014 was not used in these experiments for multiple reasons. First, this derivatization technique was developed for a metal filter collection cell and has not been successfully adapted for the impaction CTD cell featured on our system. In fact, the thermal decomposition window is not available when derivatizing since the derivatization agent needs to be purged from the cell and derivatized molecules are refocused on a secondary trap. Through this process, any decomposing material is also purged. Additionally, derivatization may complicate the identification of unknown compounds by altering mass spectral fragmentation patterns. Finally, not all compounds derivatize with 100% efficiency, further complicating quantification efforts. Further study may incorporate evaluation of the metal filter cell and online derivatized measurements to complement decomposition window analysis.

    As requested, we briefly address the lack of derivatization in the manuscript in lines 267-274 (Section 2.4.1, Paragraph 2):

    "The TAG system developed by Isaacman et al. features an online derivatization technique designed to improve analysis of oxidized species, including methoxyphenols, levoglucosan, and other compounds unique to BBOA (Isaacman et al., 2014). Although this technique presents multiple analytical advantages, it was developed for a metal filter collection cell and is not suitable for the impactor-style CTD cell used in these experiments. We chose to use the impactor-style CTD cell to allow analysis of the thermal decomposition

window, since other collection cells purge this material when transferring to a secondary trap. Additionally, we were interested to identify new molecular marker compounds that could be associated with these source types. We therefore performed all experiments without sample derivatization prior to chromatographic analysis."

2. *The TAG recovery of the selected tracers is potentially influenced by BBOA matrix effects, which could be either positive or negative in magnitude. Using a different TD-GC/MS system, Lavrich and Hays et al. (2007) showed that the thermal extraction of large PAHs from a soot matrix was hindered. Using a TAG system, Lambe et al. (2010) showed that the recovery of a C30D62 alkane internal standard in a lubricating oil matrix increased by a factor of 2-3 as a function of matrix loading. Matrix effects may be even more significant for the polar analytes measured in BBOA (e.g. methoxyphenols and sugars). Without application of representative internal standards for at least a subset of experiments, in my opinion the authors cannot unambiguously rule out the contribution of matrix effects. For example, in Fig. 3b, the authors show an increase in the abundance of vanillin, syringol, and syringaldehyde when the OH exposure in the PAM reactor is increased to 2 3.4 days. In the manuscript, a plausible formation mechanism for vanillin was provided (Figure S8). The increase in vanillin, other methoxyphenols, and other tracers (including the integrated m/z = 44 SIC) that display similar behavior could also be due to higher concentrations of desorbed primary or secondary organic aerosols that adsorbed onto active sites in the TAG sample transfer path, e.g. the effect observed in Lambe et al. (2010). At the least, the discussion should be revised to acknowledge that the above scenarios can plausibly explain the observed trends regarding increase and decrease in abundance as a function of OH exposure. A more convincing response – which may prove the above hypotheses incorrect – would be to repeat one or two of the combustion experiments, while manually spiking each collected TAG sample with an appropriate set of isotopically labeled standards. For example, there are sugars that are readily available with a range of levels of deuterium-substitution, and I also found suppliers of vanillin-5-d1 and isovanillin-2,5,6-d3.*

We thank the reviewer for this insight. While we acknowledge that the impacts of matrix effects cannot be unambiguously ruled out for our experiments, we do not think that the effects observed by Lambe et al. cited by the reviewer can adequately explain our findings.

Lambe et al. observed a two-fold increase in $C_{30}D_{62}$ TAG responses with motor oil co-injected over a range of 0-60 µg, possibly due to greater competition for active sites in the TAG sampling system with greater masses of organic matter (i.e. the motor oil). They also found that uncertainties in TAG responses were large throughout these experiments, and that the effect varied depending on the tracer compound (e.g. the size of the tracer molecule) tested. From SMPS estimated maximum mass concentrations (assuming a typical BBOA density of 1.2 g m$^{-3}$; e.g. Li et al., 2015), we estimate that for each experiment, the TAG collected total particle masses ranging from 6-16 µg for leaf BBOA and 22-36 µg for wood BBOA.

Based on the high replicability of our TAG measurements between triplicate experiments within a fuel type and the small ranges of collected masses (relative to 0-60 ug motor oil), we do not expect the effect reported in Lambe et al. to explain the trends observed in our TAG data, especially the eight-fold increase in vanillin abundance. Additionally, based on the results from the previous work cited by the reviewer, we expect matrix effects to be more significant for components with very high molecular masses, e.g. large PAHs and long-chain alkanes, and less so for smaller compounds like syringol, syringaldehyde, and vanillin.

However, we do agree that the influence of matrix effects should be investigated further to rule out bias in future experiments. Unfortunately, due to time and resource constraints, we were not able to conduct the suggested tracer tests, but these will be incorporated into future combustion experiments. We have therefore added a brief discussion of potential matrix effects to the manuscript (lines 749-759; Section 4, Paragraph 4):

"Future work will focus on characterizing sources of bias to improve quantification of material in both the TAG compound and decomposition window. For example, particle matrix effects, whereby certain compounds exhibit enhanced or diminished recovery due to the presence of a particle matrix, have been reported to influence compound responses in previous work with the TAG and other thermal desorption GC systems, particularly for large molecular weight compounds (Lambe et al., 2009; Lavrich and Hays, 2007). Lambe et al. quantified this effect for the TAG by co-injecting a constant $C_{30}$ deuterated alkane standard with 0-60 µg motor oil and found that the presence of the motor oil matrix enhanced recovery of the standard by a factor of 2-3 (Lambe et al., 2009). In these experiments, the TAG collected estimated ranges of 6-16 µg particles for leaf BBOA and 22-36 µg particles for heartwood BBOA. Based on these mass ranges, we do not expect these matrix effects to contribute significantly to our results, especially for the lower molecular weight compounds. However, future work will incorporate an evaluation of matrix effects to minimize bias in TAG measurements."

3. *Aerosol loadings corresponding to primary BBOA, oxidized BBOA, and/or SOA formed from oxidation of VOCs/IVOCs in the PAM reactor are not presented. In my opinion this data should be added to provide information about (1) the magnitude of SOA formation and corresponding SOA-to-BBOA ratio (2) phase partitioning of the selected biomass burning tracers. For example:*

    a. *C23, C25, C29 alkane signals decrease ~60%, ~70%, and ~75% following 9.8 days aging time (Fig. 3a). At an OH exposure of ~1.1E12 molec/cm3 /sec (8.5 days), Smith et al. (2009) observed ~70% decay of squalane particles subjected to heterogenous oxidation by OH. Thus, the observed C23, C25, & C29 decay rates are broadly consistent with heterogenous oxidation in the condensed phase. On the other hand, if the same compounds were oxidized in the gas phase, the observed decay rates should be much faster because the reaction is no longer rate-limited by diffusion of OH to the particle surface. Applying estimated gasphase OH rate constants of 2.9E-11, 3.2E-11, and 3.8E-11 cm3 /molec/sec, for C23, C25, C29 alkanes (Kwok and Atkinson, 1995) suggests that ~100% of the alkanes should be reacted at only 3.4 days' OH exposure if the reaction occurs in the gas phase. Information about the experimental partitioning of this tracers would provide context for interpreting the observed decay rates.*

We are grateful to the reviewer for providing this insight. We agree with the reviewer's analysis of C23, C25, and C29 partitioning, though using the information provided in Kwok and Atkinson, 1995, we obtained slightly different gas-phase OH reaction constants. Still, we found the overall conclusion to be the same: if these compounds were reacting in the gas phase only, they would be entirely depleted at 3 days of equivalent aging. Using parameters from Kwok and Atkinson's work, we also performed a similar analysis to evaluate the gas-phase kinetics of the long-chain aldehydes identified in the leaf BBOA. These findings have been summarized in the manuscript in lines 390-401 (Section 3.2.1, Paragraph 3):

"Literature information available for hydrocarbon particle- and gas-phase OH kinetics indicates that the trends observed in leaf BBOA alkanes and aldehydes with $OH_{exp}$ are consistent with heterogeneous OH oxidation. For example, Smith et al. report approximately 70% decay of squalane (a $C_{30}$ branched alkane) particles when exposed to an $OH_{exp}$ of $1.1 \times 10^{12}$ molec cm$^{-3}$ s$^{-1}$ (8.5 days of equivalent aging; Smith et al., 2009), a figure approximately consistent with the observed $C_{29}$ alkane decay of 75% at 6-10 days of equivalent aging. Additionally, based on parameters provided by Kwok and Atkinson, gas-phase OH reaction rate constants at 298K are estimated to be $2.5 \times 10^{-11}$, $2.7 \times 10^{-11}$, and $3.1 \times 10^{-11}$ cm$^3$ molec$^{-1}$ s$^{-1}$ for $C_{23}$, $C_{25}$, and $C_{29}$ alkanes, respectively (Kwok and Atkinson, 1995). Taking these rate constants into account, if purely gas-phase chemistry is assumed, all three alkanes would react nearly 100% before 1-3 days of equivalent aging. A similar analysis on relevant aldehydes gave estimated gas-rate constants of $2.5 \times 10^{-11}$, $2.8 \times 10^{-11}$, and $3.0 \times 10^{-11}$ cm$^3$ molec$^{-1}$ s$^{-1}$ for $C_{24}$, $C_{26}$, and $C_{28}$ aldehydes, respectively (Kwok and Atkinson, 1995), which in all cases would lead to complete depletion by 3.4 days of equivalent aging if gas-phase chemistry is assumed."

b.  *Levoglucosan signal decreases ~80% following 9.8 days aging time (Fig. 16). The authors reference literature rate constants of 3.09E-13 cm3 /molec/sec and 1.1E- 11 cm3 /molec/sec. The levoglucosan decay rate reported in this paper is somewhere in between the referenced literature values. Is it possible that some of the discrepancy is related to phase partitioning? This is alluded to near the end of the paper (L558-L571), but it wasn't clear to me why the authors didn't explore this further by calculating the levoglucosan phase partitioning in the oak leaf and 3 oak wood experiments and comparing to phase partitioning in the literature studies.*

We thank the reviewer for this insight. Per the reviewer's suggestion, particle-phase fractions for levoglucosan were calculated based on AMS total organic concentrations ($C_{OA}$, ug m$^{-3}$) and levoglucosan effective saturation concentrations ($C_{LG}$, ug m$^{-3}$) based on previous work (Donahue et al., 2006). The equation used to calculate partitioning is now included as equation 1 in the main text. We discuss these calculations and the relative contribution of phase partitioning to our results in lines 680-694:

"While levoglucosan decays rapidly in the leaf BBOA with increasing $OH_{exp}$, levoglucosan in the wood BBOA is depleted more slowly. Levoglucosan is classified as semivolatile (at 25°C, $p_L°$ ~ $1.81\times10^{-7}$ torr; ACD/Labs, 2017) and is therefore expected to partition between the gas and particle phases. To approximate phase partitioning, particle-phase fractions for levoglucosan ($\xi_{LG}$) were calculated based on AMS total organic concentrations and effective saturation concentrations ($C_{LG}{}^{*}$, μg m$^{-3}$) using equation 1. The resulting values and relevant parameters are reported in Table S12. For each fuel, little variance is expected in levoglucosan particle-phase fraction between oxidation conditions, so we conclude that phase partitioning is unlikely to be driving trends in levoglucosan abundances observed in these experiments. Based on the partitioning approximations, the leaf BBOA is expected to contain a higher percentage of levoglucosan in the particle phase than the heartwood BBOA (91.1 ± 1.65% vs 77.8% ± 2.26%), though in both cases, gas-phase levoglucosan concentrations are likely to remain low. The prevalence of levoglucosan in the particle phase during photochemical aging is consistent with previous laboratory measurements of aged levoglucosan particles (Kessler et al., 2010). Considering that heartwood BBOA exhibited lower total organic concentrations than the leaf BBOA, the slower depletion of levoglucosan in the heartwood samples is perhaps consistent with OH suppression effects, wherein OH experiences increased reactivity with gas-phase species at the particle surface."

c.  *Increased condensed-phase partitioning of vanillin and other methoxyphenols following potentially significant SOA formation after ~3.4 days aging time (~4.4e11 molec/cm3*sec) might explain their increase in concentration from 0 to 3.4 days' oxidation. At this approximate OH exposure, the "peak" SOA yield from oxidation of a specific precursor has been observed in previous studies, e.g. Lambe et al. (2012), Ortega et al. (2016). Although vanillin is relatively volatile, without knowing the aerosol loadings and ensuing partitioning, one can hypothesize plausible scenarios to explain some or all of the effect observed in Figure 3b. I encourage the authors to expand their discussion to analyze the observed tracer decay rates in the context of the expected phase partitioning. They already report calculated C*'s, which, together with the aerosol loadings provided by AMS, facilitate this discussion. While I don't view it as the authors' responsibility to resolve the discrepancy in reported levoglucosan decay rates, it would certainly increase the impact of the paper if a plausible explanation is possible (L533-L549).*

In addition to expanding discussion on levoglucosan phase partitioning (see above comment), we evaluated phase partitioning of relevant methoxyphenols (syringol, syringaldehyde, and vanillin) using AMS total organic concentrations and effective saturation concentrations. In doing so, we determined that under standard conditions, all three methoxyphenols are expected to exist almost exclusively in the gas phase (Table S12). We therefore agree with the reviewer's suggestion that increased SOA formation with oxidation may be driving these compounds into the particle phase.

The following discussion has been added to the main text (lines 420-431; Section 3.2.1, Paragraphs 6-7):

"To examine the potential impacts of phase partitioning for these compounds, particle-phase fractions for syringol, syringaldehyde, and vanillin ($\xi_i$) were calculated based on AMS total organic concentrations ($C_{OA}$, µg m$^{-3}$) and effective saturation concentrations ($C_i^*$, µg m$^{-3}$) using the basic partitioning equation (Donahue et al., 2006):

$$\xi_i = \left(1 + \frac{C_i^*}{C_{OA}}\right)^{-1} \tag{1}$$

Resulting particle-phase fractions are tabulated in supplemental information (Table S12). Based on these approximations, syringol, syringaldehyde, and vanillin are expected to partition primarily to the gas phase. For these compounds, the increase in abundances at low-mid levels of oxidation could therefore result from increased SOA formation driving these compounds into the particle phase. This observation is consistent with previous measurements where maximum SOA concentrations were observed at similar levels of OH$_{exp}$ for aerosol generated from oxidation of a single precursor (Lambe et al., 2012; Ortega et al., 2016)."

4. *Photobleaching of biomass burning particles has reported in previous literatures studies, e.g. Zhao et al., ACP, 2015; Wong et al., ES&T, 2017. The authors should discuss the potential role of 254 nm photolysis in these experiments, especially in regard to degradation of condensed-phase aromatic species that strongly absorb 254 nm radiation and react relatively slowly with OH due to diffusion limitations. Were control experiments conducted with 254 nm radiation (no 185 nm radiation) and no ozone addition to investigate whether photolysis induces changes in BBOA composition?*

We did not perform control experiments with only 254 nm radiation, but will do so in future experiments. We agree with the reviewer that more attention should be given to the potential contribution of 254 photolysis and diffusional effects. We therefore provide further discussion to contextualize our results (lines 224-230; Section 2.3, Paragraph 8):

"Photobleaching of BBOA, particularly at 254 nm, has been reported in previous literature (e.g. Sumlin et al., 2017; Wong et al., 2017; Zhao et al., 2015) and therefore should be considered when estimating oxidative aging. With the spreadsheet provided by Peng et al., we estimate 254 and 185 nm exposure ratios (ratio of photon flux, photons cm$^{-2}$, to OH$_{exp}$; Peng et al., 2016) to be $1.2\times10^5$ cm s$^{-1}$ and $8.1\times10^2$ cm s$^{-1}$, respectively, at a measured internally-generated O$_3$ concentration of 1.7 ppm (at the highest PAM UV lamp intensity), a water mixing ratio of 1% (RH = 30%), and assuming a maximum OHR$_{ext}$ value of 1 (Peng et al., 2016). Using Figures 1 and 2 of Peng et al., 2016 to interpret these values, we find that photolysis at both 185 nm and 254 nm is likely less than 10% in both cases."

5. *To the extent possible, I recommend that the authors make additional effort to simplify, consolidate, and streamline the results that are presented, so that the reader is not overwhelmed – especially with the PMF results (see Comment #21).*

We thank the reviewer for the suggestions for improvement. To improve the readability of our results, we made many of the changes suggested by the reviewer in the technical/minor comments, which we address individually.

Technical/Minor Comments

6. *L139: Out of curiosity, what factor(s) led to the use of oak leaves and oak wood as opposed to, for example, a soft wood fuel that might have generated a much different range of tracers? Please briefly explain why the chosen systems were studied.*

We chose oak leaves and wood because these fuels are of interest to us in Missouri, which is characterized by oak deciduous forests, and the different fuel fractions represent different types of wildfire or controlled combustion. Ongoing experiments seek to characterize BBOA from various other relevant fuels.

We address our chosen system in lines 139-142 (Section 2.2, Paragraph 1, Lines 1-4):

"White oak (*Q. alba*) heartwood and leaves were chosen for these studies due to their high abundance in Missouri and the southeastern U.S. While comparing different tree species is also of interest, two different plant fractions of the same species are studied here to investigate different types of wildfire or controlled combustion processes, some of which may only impact leaf litter-fall and others would have wood available as a fuel."

7. *L163: Clarify that the chromate coating increases the electrical conductivity of the chamber, which decreases charge buildup, and consequently loss of charged particles to the walls of the reactor.*

We have modified the description of the chromate coating to provide the appropriate clarification (lines 166-168; Section 2.2, Paragraph 1, Lines 2-4):

"The reactor consists of a 13 L cylindrical aluminum chamber coated internally with Iridite 14-2 (MacDermid, Inc., Waterbury, CT), a chromate conversion film designed to decrease charge buildup and thereby inhibit losses of charged particles to the walls of the reactor."

8. *L164-L166: State here the range of ozone mixing ratios that were added to the reactor via the ozone chamber, and the range of ozone mixing ratios that were generated inside the reactor via 185 nm irradiance of O2.*

As requested, we have included the externally added ozone mixing ratio (4 ppm, line 172; Section 2.3, Paragraph 1, Line 8) and the internally produced ozone mixing ratios (0.3-1.7 ppm, line 212; Section 2.3, Paragraph 6, Line 1). Internally produced ozone is also tabulated in supplemental information (Table S1).

9. L172: Here, and elsewhere, please be more precise with statements such as "The role of RH in OH· formation…". Changing [H2O] does change the rate of OH formation, and from the text, it appears that the authors did manipulate [H2O]. Changing RH by itself, however – for example, changing the temperature inside the reactor – does not change the rate of OH production.

We altered the wording as suggested to provide clarity (lines 177-179; Section 2.3, Paragraph 1, Lines 13-15):

"The reactor water concentration, and therefore RH, was altered by controlling $N_2$ flow through a Nafion membrane humidifier (Perma Pure LLC, Lakewood, NJ). The role of water concentration in OH formation is discussed in detail in Supplemental Information (Method: PAM Calibrations and Figure S3)."

10. *L183-L195: It wasn't clear why the authors didn't simply add SO2 during a "representative" combustion experiment to conduct an online OH exposure calibration in the presence of (I)VOCs that might have suppressed OH. I would certainly encourage this, if practical, as this approach should introduce less uncertainty than attempting to apply the OH exposure estimator when the OH reactivity of the biomass smoke emissions is not known.*

We did not add $SO_2$ during a representative experiment for several reasons. First, we had concerns about the impacts of $SO_2$ on the chemical pathways occurring during atmospheric aging. Previous work demonstrates that addition of $SO_2$ can accelerate heterogeneous reactions of OH with organic aerosol by reacting with peroxy radicals to produce alkoxy radicals, propagating a chain reaction (Richards-Henderson et al., 2016). Additionally, long-chain aliphatics, particularly alkenes and fatty acids, have been observed to react with SO2 to create long-chain organosulfate molecules (Passananti et al., 2016). Second, the sulfuric acid created

in the PAM reactor tends to damage inert coatings on the equipment used in these experiments (Williams et al., 2016).

With additional CO measurements (described in Supplemental Information: Methods: PAM Calibrations and Equivalent Aging Estimations, "Estimation of External OH Reactivity ($OHR_{ext}$)"), we have attempted to better constrain the calibration. However, in future experiments, we hope to directly measure both total VOCs and concentrations of specific VOCs to obtain a more quantitative calibration.

11. *L282: I suggest replacing "determined" with "inferred" or similar.*

   As suggested, "determined" has been replaced with "inferred" in this sentence (now line 356; Section 3.2, Paragraph 2, Line 8).

12. *L309: This wording is confusing. Were oak leaves placed in a solvent to extract compounds on the surface of the leaves, and was this extract then injected into the TAG CTD? If so, please rewrite the sentence to clarify. What solvent(s) were used?*

   The wording here was altered for clarity (lines 385-389; Section 3.2.1, Paragraph 2, Lines 13-17):

   "To confirm the presence of aldehydes in the leaf waxes, solvent extractions were performed on oak leaves and were manually injected onto the TAG CTD cell (Method: Oak Leaf Solvent Extractions and Figure S8 in Supplemental Information). Analysis of these extractions confirm that the aldehydes are present in the leaf wax prior to devolatilization and combustion."

   All details regarding oak leaf solvent extractions are provided in Supplemental Information in the section titled "Method: Oak leaf solvent extractions."

13. *L316-L318: It's true that sinapaldehyde signal decays more quickly than other tracers (e.g. alkanes), but ~70% decay over 3.4 days' aging is still slow in the context of gas phase oxidation rates – this corresponds to an effective rate constant of ~2.7E-12 cm3 /molec/sec, whereas, for example, the gas-phase OH rate constant of syringol is 8.5E- 11 cm3 /molec/sec (Lauraguais et al., 2015).*

   We have modified the manuscript to include relative information on the reaction rate constants of sinapaldehyde (lines 407-411; Section 3.2.1, Paragraph 4, Lines 6-10):

   "Of the compounds examined, sinapaldehyde decays most rapidly in the PAM reactor, with the normalized average integrated peak area decreasing by approximately 70% from 0 days to 2-3 days of equivalent aging (Figure 3b). Based on a rapid gas-phase OH reaction rate constant of $2.7 \times 10^{-12}$ $cm^3$ $molec^{-1}$ $s^{-1}$ (Lauraguais et al., 2015), the observed sinapaldehyde decay is likely occurring in the particle phase."

14. *L387-L395: This paragraph seems out of place here, I would consider paraphrasing and moving to Conclusions.*

   As suggested, this paragraph has been moved to the Conclusions section (now lines 728-736; Section 4, Paragraph 2).

15. *L417: What is the signal-to-noise ratio for the m/z = 44 decomposition SICs? I understand that the SIC's presented are background corrected – how large are gas-phase CO2 + backgrounds compared to the background + sample m/z = 44 SIC's? This might be useful information to add to the Supplement.*

   We thank the reviewer for this suggestion. We have incorporated raw m/z 44 background signals and example decomposition m/z 44 SICs for each fuel type into a new supplemental figure (Figure S16).

16. *L441-L454 and Figure 10: Implicit in this discussion is the observation that TAG recovery of highly oxidized/oxygenated species is low (even with inclusion of the thermal decomposition window). One or two sentences should be added that states this explicitly. Another point that should be made is that this attempt at a direct f43 and f44 comparison assumes AMS flash vaporization at T = 600 deg C and TAG thermal decomposition at T < 310 deg C produce the same m/z = 43 and m/z = 44 ion signals. It's not clear to me that this assumption is justified, but at the least, this assumption should also be stated explicitly.*

The purpose of this figure is to suggest that inclusion of the thermal decomposition window facilitates more thorough TAG analysis of oxidized OA. We provide AMS f43 and f44 for comparison because this is a well-established technique for interpreting OA oxidative evolution.

Implicit in the interpretation of this figure is that the m/z 43 and m/z 44 detected by the TAG and the AMS are similar enough to merit some form of comparison, at least in how the signals trend with oxidative aging. However, we wish to clarify that this comparison does not entail that f43 and f44 measurements are the same between the two instruments (hence the distinct axes for both TAG and AMS f43 and f44).

The reviewer's comments have been considered, and lines 564-570 (Section 3.3.1, Paragraph 3, Lines 11-17) now read:

"In general, the TAG fractions tend to fall to the left of AMS $f_{44}$ vs $f_{43}$ data points, indicating that the TAG excels at throughput of less-oxygenated hydrocarbon OA and struggles with throughput of oxidized species in the compound window. However, the increase in TAG $f_{44}$ with inclusion of decomposition window material shows a clearer oxidation trend that is in greater agreement with the AMS oxidation trend. This interpretation implies that the $m/z$ 43 and $m/z$ 44 signals obtained in the TAG decomposition window from sample thermal desorption at 310°C is similar in nature to material flash-vaporized at 600°C in the AMS."

17. *L483-L489: Consider also moving this to Conclusions.*

This paragraph has been removed in favor of a more thorough discussion of thermal decomposition window analysis in the conclusion section (lines 766-774; Section 4, Paragraph 6).

18. *L539-L549: In my opinion, Lai et al.'s explanation for discrepancy in levoglucosan oxidation kinetics requires two unlikely scenarios: 5 a. using mz144 rather than mz162 would bias kLG ~ 30x too low -- Fortenberry et al.'s measurements are not subject to mass spectra interference either, and their levoglucosan decay rate is much closer to Kessler et al. than than Hennigan/Lai et al. (L564). A calculated levoglucosan + OH rate constant of 2.21E-13 cm3 /molec/sec (Bai et al., 2013), which is based on a theoretical study, may help put the different results in context. OR b. oxidation kinetics of OH + levoglucosan (or other model organics) are not firstorder with respect to OH. Previous studies suggest otherwise (e.g. Renbaum and Smith, 2011). I don't think it benefits the discussion in this paper to cite someone else's (in my opinion) incomplete explanation. I would consider removing it.*

We thank the reviewer for this insight. In the heartwood BBOA, the kinetics do agree best with those presented by Kessler et al., but in the leaf BBOA, the kinetics are most similar to those of Hennigan et al./Lai et al. Because we do not believe we have an adequate explanation for this discrepancy, the goal of this discussion was to investigate all potential explanations currently available in the literature. Therefore, we retain the explanation provided in Lai et al. because we believe it provides additional context for discrepancies in literature-reported levoglucosan kinetics, though we clarify that our chromatographic methods are not subject to the same mass spectral interferences (lines 673-674; Section , Paragraph 7, Lines 14-15):

"However, our chromatographic methods are not subject to this mass spectral interference, and in the case of the heartwood BBOA, the TAG-measured levoglucosan decay matches the decay predicted by Kessler et al."

19. *Figure 2 and related text: It is hard to distinguish the multiple shades of green in Fig. 2a, and for some compounds it is hard to distinguish changes in relative abundance between chromatograms representing "3.4 days" and "9.8 days". Please consider changing the colors in Fig. 2a. Additionally, consider removing the "3.4 days" TIC from Figs. 2a and 2b – this figure seems to be a general, "big picture" type of plot, so this would simplify the chromatogram without changing the take-home points.*

The color scheme has been changed to teal, purple, and pink in all leaf BBOA figures. Because we want to demonstrate visually that a select few compounds may increase in abundance with photochemical aging, we retain the trace for low-mid level of oxidation in the figure. However, we believe the change in color scheme greatly improves figure readability for the leaf BBOA data.

20. *Figure 3 and related text: Please add a subpanel plotting the concentrations of organics and any relevant inorganic species (e.g. $K+$ ) measured by AMS following OH exposure in the PAM reactor.*

Since the focus of this figure was intended to be solely on trends in TAG species, and because AMS total organics are presented in later figures (e.g. Figure 12), we did not include these species in Figure 3a and 3b. However, we have incorporated total organics, potassium ($K+$), and sulfate ($SO4+$) concentrations, which were the most abundant species in our samples, in a supplemental plot (Figure S15).

21. *Figures 4-5, Figures 6-7, Figures 11-12, Figures 13-14: I find these figures to be complex and overwhelming. I found it difficult to quickly "match up" 37 sets of chromatograms and mass spectra (15 + 18 + 4) for each PMF factor across separate figures as is currently presented. In my opinion, reorganizing these figures to put the mass spectra next to their corresponding chromatograms would improve their clarity and usefulness.*

*Here is one idea for consideration:*

*Figs. 4, 6, 11, 13: Put each factor "TIC" for 0, 3.4 days, 9.8 days on the same x-axis, (as was done in Fig. 2). Choose a three colors, one each for 0, 3.4, 9.8 days, that are shared across all factors. Then place the corresponding mass spectra shown in Figs. 5, 7, 12, 14 to the right of the TICs. This modification would: (i) reduce the number of "PMF figures" from 8 to 4 (ii) remove the number of subpanels in Figs. 4, 6, 11, 13 by 3x (iii) allow enough room to put the mass spectra from Figs. 5, 7, 12, 14 to the right of their chromatograms 6 (iv) make it unnecessary to use unique colors for each factor in attempt to match up the chromatograms and mass spectra across figures. If this is not agreeable, the authors might consider just labeling the factors in Figs. 4, 6, 11, 13 and moving the mass spectral figures (5, 7, 12, 14) to the Supplement. This would save space/publication costs in the main part of the manuscript without making it any more difficult to "match up" the chromatograms and spectra.*

To improve readability, we modified all PMF figures according to the reviewers first suggestion: time series and mass spectra for each factor are now presented side-by-side. This change has allowed for consolidation of PMF mass spectra and time series figures into one figure (Figures 4, 5, 9, and 10).

22. *Figure S13: The caption states 15 micrograms of levoglucosan and 5 micrograms of quinic acid were injected. That seems like a very large analyte mass for a single compound injection – is there any chance this is a typo, and that the injected quantities were actually 15 and 5 nanograms?*

This mass was not a typo. We experienced difficult single-component throughput of the injected levoglucosan and quinic acid standards, necessitating much larger injected masses. The mass injected here is much larger than found in the samples of interest, and the signal is larger than would be necessary as seen with peak "fronting" that appears in these chromatograms, a sign that excess mass was injected.  While significantly

lower concentrations can be observed, poor transfer of highly oxidized standard compounds from the CTD cell to the column has been reported as a persistent problem in previous TAG literature (Williams et al., 2006), spurring the development of *in situ* derivatization methods (Isaacman et al., 2014), as discussed previously. We expect that, as reported for previous TAG measurements, a particle matrix improves throughput of these oxidized compounds compared to liquid standard injections (Lambe et al., 2009). Thus, while the standards presented here are sufficient for identification of compound retention time, they are not adequate for mass calibration of levoglucosan or quinic acid. Ongoing work focuses on better characterizing particle matrix effects for BBOA compounds of interest and improving TAG mass calibration methods.

**Literature Cited:**

Donahue, N. M., Robinson, A. L., Stanier, C. O. and Pandis, S. N.: Coupled Partitioning, Dilution, and Chemical Aging of Semivolatile Organics, Environ. Sci. Technol., 40(8), 2635–2643, doi:10.1021/es052297c, 2006.

Isaacman, G., Kreisberg, N. M., Yee, L. D., Worton, D. R., Chan, A. W. H., Moss, J. A., Hering, S. V. and Goldstein, A. H.: Online derivatization for hourly measurements of gas- and particle-phase semi-volatile oxygenated organic compounds by thermal desorption aerosol gas chromatography (SV-TAG), Atmos Meas Tech, 7(12), 4417–4429, doi:10.5194/amt-7-4417-2014, 2014.

Kessler, S. H., Smith, J. D., Che, D. L., Worsnop, D. R., Wilson, K. R. and Kroll, J. H.: Chemical Sinks of Organic Aerosol: Kinetics and Products of the Heterogeneous Oxidation of Erythritol and Levoglucosan, Environ. Sci. Technol., 44(18), 7005–7010, doi:10.1021/es101465m, 2010.

Kwok, E. S. C. and Atkinson, R.: Estimation of hydroxyl radical reaction rate constants for gas-phase organic compounds using a structure-reactivity relationship: An update, Atmos. Environ., 29(14), 1685–1695, doi:10.1016/1352-2310(95)00069-B, 1995.

Lambe, A. T., Logue, J. M., Kreisberg, N. M., Hering, S. V., Worton, D. R., Goldstein, A. H., Donahue, N. M. and Robinson, A. L.: Apportioning black carbon to sources using highly time-resolved ambient measurements of organic molecular markers in Pittsburgh, Atmos. Environ., 43(25), 3941–3950, doi:10.1016/j.atmosenv.2009.04.057, 2009.

Lavrich, R. J. and Hays, M. D.: Validation studies of thermal extraction-GC/MS applied to source emissions aerosols. 1. Semivolatile analyte-nonvolatile matrix interactions, Anal. Chem., 79(10), 3635–3645, doi:10.1021/ac0623282, 2007.

Li, C., Ma, Z., Chen, J., Wang, X., Ye, X., Wang, L., Yang, X., Kan, H., Donaldson, D. J. and Mellouki, A.: Evolution of biomass burning smoke particles in the dark, Atmos. Environ., 120(Supplement C), 244–252, doi:10.1016/j.atmosenv.2015.09.003, 2015.

Passananti, M., Kong, L., Shang, J., Dupart, Y., Perrier, S., Chen, J., Donaldson, D. J. and George, C.: Organosulfate Formation through the Heterogeneous Reaction of Sulfur Dioxide with Unsaturated Fatty Acids and Long-Chain Alkenes, Angew. Chem. Int. Ed., 55(35), 10336–10339, doi:10.1002/anie.201605266, 2016.

Peng, Z., Day, D. A., Ortega, A. M., Palm, B. B., Hu, W., Stark, H., Li, R., Tsigaridis, K., Brune, W. H. and Jimenez, J. L.: Non-OH chemistry in oxidation flow reactors for the study of atmospheric chemistry systematically examined by modeling, Atmos Chem Phys, 16(7), 4283–4305, doi:10.5194/acp-16-4283-2016, 2016.

Richards-Henderson, N. K., Goldstein, A. H. and Wilson, K. R.: Sulfur Dioxide Accelerates the Heterogeneous Oxidation Rate of Organic Aerosol by Hydroxyl Radicals, Environ. Sci. Technol., 50(7), 3554–3561, doi:10.1021/acs.est.5b05369, 2016.

Smith, J. D., Kroll, J. H., Cappa, C. D., Che, D. L., Liu, C. L., Ahmed, M., Leone, S. R., Worsnop, D. R. and Wilson, K. R.: The heterogeneous reaction of hydroxyl radicals with sub-micron squalane particles: a model system

for understanding the oxidative aging of ambient aerosols, Atmos Chem Phys, 9(9), 3209–3222, doi:10.5194/acp-9-3209-2009, 2009.

Sumlin, B. J., Pandey, A., Walker, M. J., Pattison, R. S., Williams, B. J. and Chakrabarty, R. K.: Atmospheric Photooxidation Diminishes Light Absorption by Primary Brown Carbon Aerosol from Biomass Burning, Environ. Sci. Technol. Lett., doi:10.1021/acs.estlett.7b00393, 2017.

Williams, B. J., Goldstein, A. H., Kreisberg, N. M. and Hering, S. V.: An In-Situ Instrument for Speciated Organic Composition of Atmospheric Aerosols: Thermal Desorption Aerosol GC/MS-FID (TAG), Aerosol Sci. Technol., 40(8), 627–638, doi:10.1080/02786820600754631, 2006.

Williams, B. J., Zhang, Y., Zuo, X., Martinez, R. E., Walker, M. J., Kreisberg, N. M., Goldstein, A. H., Docherty, K. S. and Jimenez, J. L.: Organic and inorganic decomposition products from the thermal desorption of atmospheric particles, Atmos Meas Tech, 9(4), 1569–1586, doi:10.5194/amt-9-1569-2016, 2016.

Wong, J. P. S., Nenes, A. and Weber, R. J.: Changes in Light Absorptivity of Molecular Weight Separated Brown Carbon Due to Photolytic Aging, Environ. Sci. Technol., 51(15), 8414–8421, doi:10.1021/acs.est.7b01739, 2017.

Zhao, R., Lee, A. K. Y., Huang, L., Li, X., Yang, F. and Abbatt, J. P. D.: Photochemical processing of aqueous atmospheric brown carbon, Atmos Chem Phys, 15(11), 6087–6100, doi:10.5194/acp-15-6087-2015, 2015.

---

## Editor Decision (ED1)

Dear authors. Please fix the following minor issues (the ones you agree with) before the final publication.

L136: There is an issue with the section heading

L154, L186, L257, L699: missing space after the period

L314: Figures S13 and S14 are mentioned before earlier SI figures. Along the same lines, Table S9 is mentioned before tables S4, S5, S6, S7 on page 10. I am not sure whether it is easy to fix the order of appearance with the current organization of the SI section. I encourage you to think how you can reorganize the SI section to avoid (or at least minimize) jumping up and down through it while reading the manuscript.

L320: missing period and space after the subsection title

L337: It is not clear to me how this can even be done: "the compound structure was inferred by retention time and manually evaluating possible fragmentation patterns". A more specific description would help. Have you tried running the molecules you came up with as standards?

Figure 3: the text in the legend is too small.

Figures 4, 5, 9, 10: some of the text label are too small and will be hard to see in the final version

Most of the figures: the image quality appears to be too low in the PDF file. I would increase the dpi in the final images produced for the final paper.

Points raised by the reviewer 3: I tend to agree with the comment by reviewer 3 that matrix effects could affect the interpretation of the results, and that using internal standards added to the sample would help avoid such matrix effects. While it is not going to be possible to address this in this paper, I think it would be good if you make it happen for your future papers of this sort.

Supporting Information (SI) section:

Figures S6/S7 and Tables S4/S5:

- Structure 19 is unlikely, allenes are quite uncommon in the atmosphere. Perhaps it is an alkyne?
- Structure 23 is also unlikely, its saturation vapor pressure is way too low for this compound or its decomposition products to be detectable by GC or TAG without derivatization. This could be a library mismatch.
- Does it make sense to split one figure (images of your molecules) in two S6 and S7? Same question for the table? I understand that they stretch over 2 pages; you can say "Figure S6 continued" in the caption

Figure S9: the step with the four-membered ring is probably incorrect. The classical RO2 -> RO -> CC bond fission reaction sequence is much more likely.

Figure S14: was there an issue with H/C analysis? How is it possible to end up with unphysical H/C around 0 or above 3? I would add a clarification on the atomic ratio measurements.

Figure S15: calling sulfate (SO4+) is going to be misleading. Even though SO4+ may be the ion measured by AMS (I do not actually know how AMS does it), the calibrated axis actually refers to SO4(2-). I would fix it.

Table S2: too many significant figures in the reported results. The general rule is that the error bar should have 1 or at most 2 significant digits. Therefore, 314.20 +/- 78.59 should be 314 +/- 79, etc.

Tables S10, S11 – the same issue with significant digits.